# Measurement-Consistent Langevin Corrector: A Remedy for Latent Diffusion Inverse Solvers

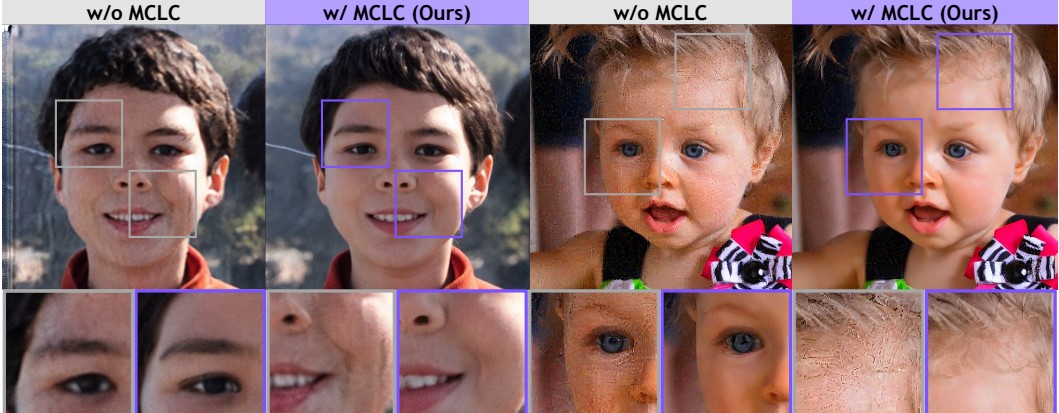

Figure 1: **Reconstructed results without and with our proposed Measurement-Consistent Langevin Corrector (MCLC).** While naive solvers (Rout et al., 2023; Song et al., 2024) fail to recover images faithfully, our plug-and-play MCLC successfully corrects latent diffusion inverse solvers by reducing the gap to the true reverse sampling process. MCLC is simple yet effective in stabilizing latent diffusion inverse solvers, thereby mitigating artifacts and improving quality.

## Abstract

With recent advances in generative models, diffusion models have emerged as powerful priors for solving inverse problems in each domain. Since Latent Diffusion Models (LDMs) provide generic priors, several studies have explored their potential as domain-agnostic zero-shot inverse solvers. Despite these efforts, existing latent diffusion inverse solvers suffer from their instability, exhibiting undesirable artifacts and degraded quality. In this work, we first identify the instability as a discrepancy between the solver's and true reverse diffusion dynamics, and show that reducing this gap stabilizes the solver. Building on this, we introduce *Measurement-Consistent Langevin Corrector (MCLC)*, a theoretically grounded plug-and-play correction module that remedies the LDM-based inverse solvers through measurement-consistent Langevin updates. Compared to prior approaches that rely on linear manifold assumptions, which often do not hold in latent space, MCLC operates without this assumption, leading to more stable and reliable behavior. We experimentally demonstrate the effectiveness of MCLC and its compatibility with existing solvers across diverse image restoration tasks. Additionally, we analyze blob artifacts and offer insights into their underlying causes. We highlight that MCLC is a key step toward more robust zero-shot inverse problem solvers.

## 1 Introduction

In many scientific and engineering problems, we have access only to limited observations obtained through a forward system; thus, recovering the underlying signal from these measurements is a long-standing challenge, known as the *inverse problem* (Groetsch & Groetsch, 1993). Most inverse problems, such as image restoration (Kawar et al., 2022; Chung et al., 2023; Zhu et al., 2023), medical imaging (Song et al., 2022; Chung & Ye, 2022; Cha et al., 2021), and astrophotography (Sun &

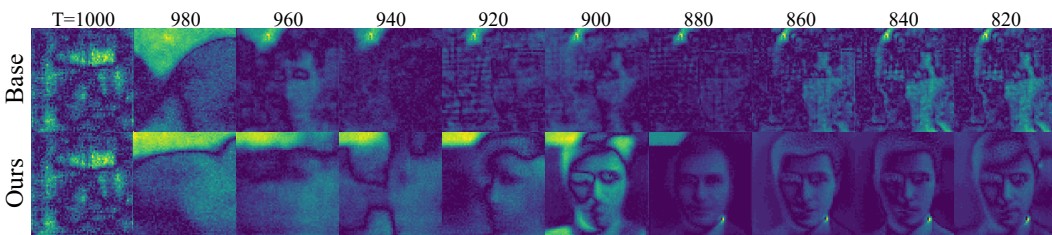

Figure 2: **Reverse diffusion dynamics of PSLD.** We visualize the reverse sampling trajectory of $z_{0|t}$ for the latent diffusion inverse solver (PSLD (Rout et al., 2023)). The naive dynamics of solver exhibits undesirable artifacts (first row), whereas the solver corrected with MCLC yields cleaner and more structured latents (second row). For clarity, only the fourth channel is visualized.

Bouman, 2021; Zhang et al., 2025b; Zheng et al., 2025), aim to recover an unknown underlying signal from partial and noisy measurements. According to Hadamard (1902), such problems are defined as ill-posed problems, which require prior knowledge of the original signal domain.

Hand-crafted priors, including total variation (TV), Tikhonov regularizations, or Deep Image Prior, were designed to impose manually specified constraints that natural images are assumed to satisfy (Romano et al., 2017; Ulyanov et al., 2018). Although these priors have been widely used, they are limited in their expressiveness and often fail to capture the complexity of real signals. Data-driven neural prior has emerged as an alternative that learns signal priors from data using neural networks. As one of the most promising approaches, generative modeling has gained attention for learning data-driven priors. In particular, pretrained diffusion models serve as strong learned priors, showing remarkable performance in solving inverse problems (Chung et al., 2023; Song et al., 2021; Kawar et al., 2022). However, raw-signal (*e.g.*, pixel) based diffusion models yield domain-specific priors, restricting diffusion inverse solvers to domain-specialized solutions.

Latent Diffusion Models (LDMs) (Rombach et al., 2022) * could achieve scalability while capturing broad natural image statistics, thereby serving as domain-agnostic generic priors. This potential for generalization to complex signals allows generic learned priors to be extended to solving inverse problems in a zero-shot manner, beyond domain-specific settings (*e.g.*, face images). Despite this promise as zero-shot inverse solvers, using LDMs for inverse problems still falls short due to the solver instability, which leads to artifacts and degraded reconstruction quality (Rout et al., 2023; 2024; Song et al., 2024; Zhang et al., 2025a) (see Fig. 1). This limitation is known to stem from backpropagation through the latent decoder, which induces problematic latent updates (Chung et al., 2024; Raphaeli et al., 2025) (see the first row of Fig. 2). To alleviate this, latent diffusion inverse solvers (Rout et al., 2023; Chung et al., 2024) have incorporated various forms of regularization. Another predominant approach is to extend manifold-preserving methods (Zirvi et al., 2025; He et al., 2024) to LDMs under the linear manifold assumption, which often fails to hold in latent space (Song et al., 2024). Despite these efforts, we observe that a substantial gap remains between the true reverse diffusion dynamics and the dynamics followed by latent diffusion inverse solvers (refer to Fig. 3).

In this work, we first identify the instability of LDM-based inverse solvers as a gap between the solver's and the true reverse diffusion dynamics, and show that alleviating this gap stabilizes the solvers. Building on this insight, we propose **M**easurement-**C**onsistent **L**angevin **C**orrector (MCLC), a novel and theoretically justified plug-and-play method that reduces the gap by correcting latent diffusion inverse solver dynamics toward the true reverse diffusion dynamics while preserving measurement consistency. We leverage Langevin dynamics as a principled corrector that pulls the updated latent closer to the stationary distribution at each timestep. Importantly, we constrain the Langevin updates to the orthogonal complement of the measurement-consistent gradient so that Langevin updates preserve measurement consistency. In doing so, MCLC stabilizes the LDM-based inverse solvers and consequently remedies artifacts and degraded quality , without compromising the performance of latent diffusion inverse solvers. MCLC takes a more general perspective that does not rely on the linear manifold assumption, which is commonly adopted in prior approaches but may not hold in latent space. This leads to more stable and reliable behavior in latent diffusion inverse solvers, improving overall performance for base and competing methods across a range of inverse problems. In addition, we analyze artifacts in latent space and provide valuable insights for future studies.

---

*In this paper, LDM denotes a general image prior (*e.g.*, SD v1.5:huggingface.co/botp/stable-diffusion-v1-5), not domain-specific ones (*e.g.*, FFHQ-LDM)

## 2 BACKGROUND AND RELATED WORK

### 2.1 DIFFUSION MODELS

Diffusion models learn data distribution $p(\boldsymbol{x})$ by modeling score vector field, *i.e.*, $\nabla_{\boldsymbol{x}} \log p(\boldsymbol{x})$ (Song et al., 2021). Since the computing true score of the data distribution is intractable, a forward diffusion process is introduced that gradually perturbs the data into a Gaussian distribution (Ho et al., 2020). This process is formulated as stochastic differential equations (SDE), defining a family of marginal distributions at each timestep $\{q_t\}_{t \in [0,1]}$. The training objective, denoising score matching, is defined such that the reverse process matches the family of distributions induced by the forward process. Formally, the model is trained to minimize the expected KL divergence across timesteps: $\mathbb{E}_{t \sim \mathcal{U}[0,1]}[\mathcal{D}_{\mathrm{KL}}(q_t \| p_t)]$, which is equivalent to learning the score function with a neural score network parameterization $\boldsymbol{s}_\theta$, *i.e.*, $\mathbb{E}_{t \sim \mathcal{U}[0,1], x \sim q_t}[\|\nabla_x \log p_t(\boldsymbol{x}) - \boldsymbol{s}_\theta(x, t)\|_2^2]$. With the learned score network, sampling is performed by solving the reverse-time SDE (or probability flow ODE), which follows the predicted score trajectory to transform Gaussian noise into data samples (Ho et al., 2020).

### 2.2 DIFFUSION INVERSE SOLVERS

Inverse problems aim to estimate the underlying signal $\boldsymbol{x}$ from measurements $\boldsymbol{y}$, formulated as:

$$\boldsymbol{y} = \mathcal{A}(\boldsymbol{x}) + \boldsymbol{n}, \tag{1}$$

where $\mathcal{A}$ is a forward measurement operator and $\boldsymbol{n}$ denotes measurement noise. From a Bayesian perspective, the posterior distribution can be expressed as $p(\boldsymbol{x}|\boldsymbol{y}) \propto p(\boldsymbol{y}|\boldsymbol{x}) \, p(\boldsymbol{x})$, where $p(\boldsymbol{y}|\boldsymbol{x})$ is the likelihood defined by the measurement model in Eq. (1) and $p(\boldsymbol{x})$ represents a prior on the signal.

Since diffusion models have shown remarkable performance in modeling the prior $p(\boldsymbol{x})$, various diffusion inverse solvers have been studied (Kawar et al., 2022; Chung et al., 2022; 2023; Wang et al., 2023). Among existing approaches, gradient-based diffusion inverse solvers (Chung et al., 2023; Rout et al., 2023) have been widely adopted as they require only the differentiability of the measurement operator, which allows broad applications beyond linear problems. Specifically, in the Bayesian formulation, the inverse problem can be solved by using the gradient of the log-posterior, which can be decomposed into a likelihood term and a prior term:

$$\nabla_{\boldsymbol{x}_t} \log p(\boldsymbol{x}_t|\boldsymbol{y}) = \nabla_{\boldsymbol{x}_t} \log p(\boldsymbol{y}|\boldsymbol{x}_t) + \nabla_{\boldsymbol{x}_t} \log p(\boldsymbol{x}_t). \tag{2}$$

The prior term is modeled by diffusion models, whereas obtaining the log-gradient of the noisy likelihood $\nabla_{\boldsymbol{x}_t} \log p(\boldsymbol{y}|\boldsymbol{x}_t)$ is generally intractable. Diffusion Posterior Sampling (DPS; Chung et al., 2023) addressed this intractability by approximating $p(\boldsymbol{y}|\boldsymbol{x}_t)$ as $p(\boldsymbol{y}|\boldsymbol{x}_0 = \mathbb{E}[\boldsymbol{x}_0|\boldsymbol{x}_t])$. Then, by performing gradient ascent with the diffusion prior and approximated likelihood, the inverse problem is solved within the reverse sampling process.

Diffusion inverse solvers have been mainly studied in the raw signal domain, and are often built upon the manifold hypothesis and linear manifold assumption. Under this assumption, the gradient used for the measurement-consistency step may push the updated states away from the desired diffusion manifold at each timestep $t$ (Chung et al., 2022; 2023; He et al., 2024; Zirvi et al., 2025). Since this drift leads to reduced fidelity, several works have proposed manifold-preserving approaches based on the linear manifold assumption: He et al. (2024) use a pretrained autoencoder for manifold projection, and Zirvi et al. (2025) project the gradient, both aiming to prevent updates from leaving the diffusion manifold. Although these approaches have been extended to latent spaces, the underlying linear manifold assumption may not generally hold true (Song et al., 2024).

### 2.3 LATENT DIFFUSION INVERSE SOLVERS

Although diffusion models have become leading generative models, scaling them to large datasets in pixel space is computationally prohibitive. Latent Diffusion Models (LDMs) (Rombach et al., 2022) address this issue by operating in the latent space of a pretrained autoencoder (Kingma & Welling, 2014), enabling efficient large-scale generative modeling and serving as versatile priors. In this context, a line of works has extended diffusion-based inverse solvers to LDMs towards zero-shot inverse problem solvers (Rout et al., 2023; 2024; Song et al., 2024; Zhang et al., 2025a; Kim et al., 2025b). While these approaches broaden the applicability and generalizability of inverse solvers,

instability remains an inherent challenge, leading to artifacts and degraded quality (Chung et al., 2024; Raphaeli et al., 2025). Previous studies (Song et al., 2024; Zirvi et al., 2025) attributed these issues to decoder backpropagation, which exacerbates off-manifold deviations from the desired manifold.

A few notable approaches (Zirvi et al., 2025; He et al., 2024) have addressed this challenge by extending manifold-preserving methods to LDMs. While these methods mitigate the problem, the linear manifold assumption generally fails to hold (Song et al., 2024), so these methods still face artifacts and limited performance. More recently, Raphaeli et al. (2025) instead avoids the need to backpropagate through the decoder by training task-specific degradation operators that operate directly in the latent space. However, this design undermines measurement consistency–a fundamental goal of inverse problems–and relies on domain-specific components that limit generalizability. MCLC operates without the linear manifold assumption while preserving measurement consistency, yielding more stable behavior in latent space. This approach substantially mitigates these limitations and effectively remedies latent diffusion inverse solvers.

## 3 MEASUREMENT-CONSISTENT LANGEVIN CORRECTOR (MCLC)

In this section, we introduce *Measurement-Consistent Langevin Corrector* (MCLC), which leverages Langevin dynamics in latent space in a post-update manner, to correct deviations from the desired distribution at each timestep $t$. To preserve measurement consistency in correcting steps, the Langevin update is restricted to the subspace orthogonal to the measurement-consistent direction. This approach reveals the potential of existing LDM-based inverse solvers by stabilizing them while preserving measurement consistency–a key aspect of baseline solver performance and inverse problem objectives.

To clarify the instability observed in prior works, we examine the reverse diffusion dynamics of latent diffusion inverse solvers. As shown in Fig. 2, the naive solver yields unstable reverse dynamics, which may lead to artifacts and degraded reconstructions. To demonstrate this observation more concretely, we quantify the gap by measuring the Kullback–Leibler (KL) divergence against the true reverse diffusion dynamics. Figure 3 demonstrates that the naive solver dynamics exhibits a significant gap, indicating clear divergence from the stationary distribution at each timestep. Detailed experimental settings of Fig. 3 are provided in the Appendix. C.1. Based on this, we assume that reverse dynamics of latent diffusion inverse solvers deviate from the stationary distribution at each timestep.

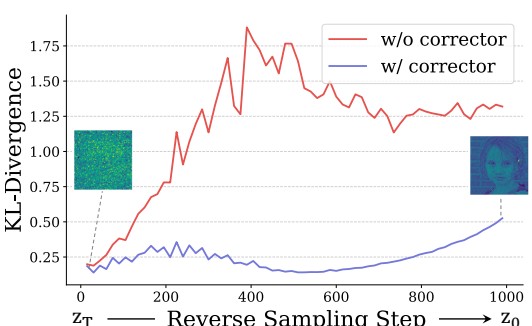

Figure 3: **KL divergence between the time-evolving distribution of solver and the true reverse diffusion distribution across timesteps.** The clear gap (red line) supports our assumption, and MCLC effectively narrows it (purple line).

**Assumption 1** (Deviation from $p_t$). *The measurement-guided, time-evolving distribution $q_t^{\#}$, which may yield artifacts and degraded quality, deviates from $p_t$ at timestep $t$. Formally,*

$$\mathcal{D}_{\mathrm{KL}}(q_t^{\#} \| p_t) \geq \gamma_t, \quad \text{for some } \gamma_t > 0, \tag{3}$$

*where $p_t$ denotes the marginal distribution of the reverse diffusion process at time $t$, which serves as the stationary distribution of the Langevin corrector dynamics at each timestep.*

Based on this assumption, the following proposition establishes that applying Langevin dynamics after the measurement-consistency step facilitates the convergence toward the stationary distribution (Vempala & Wibisono, 2019). This stabilizes the dynamics of latent diffusion inverse solvers, which in turn mitigates artifacts and improves reconstruction quality. The proof of Proposition 1 can be found in Appendix. A.

**Proposition 1** (Langevin Corrector). *Fix a timestep $t$ and let $p_t$ be a target distribution. Consider the continuous corrector process $\{Z_t^c\}_{c \geq 0}$ initialized with $Z_t^0 \sim q_t^{\#}$. The process evolves according to the Langevin dynamics with frozen target $p_t$: $dZ_t^c = \nabla \log p_t(Z_t^c)dc + \sqrt{2}dW_c$. Let $q_t^c$ denote the distribution of $Z_c$. Then, the KL divergence monotonically decreases along the process, unless $q_t^c = p_t$, in which case equality holds:*

$$\mathcal{D}_{\mathrm{KL}}(q_t^c \| p_t) \leq \mathcal{D}_{\mathrm{KL}}(q_t^{\#} \| p_t), \quad \forall c \geq 0. \tag{4}$$

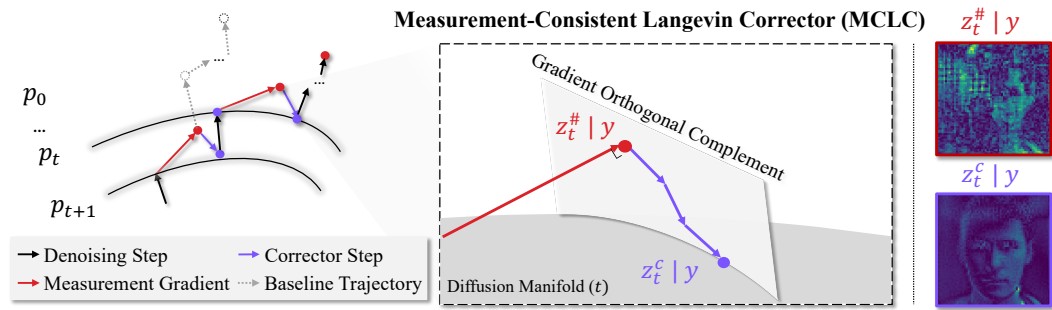

Figure 4: **Illustration of MCLC.** After the measurement consistency step, MCLC is applied to mitigate the off-stationarity. MCLC performs Langevin updates on the subspace orthogonal to the measurement gradient, thereby preserving measurement consistency during the correction process. Our proposed corrector effectively alleviates the problematic latent updates.

As prior studies (Dalalyan, 2017; Durmus & Moulines, 2019) have shown, discretization of the Langevin process preserves the property of decreasing KL divergence up to discretization error, provided the step size is sufficiently small. In this work, we implement the corrector using the Euler–Maruyama discretization of Langevin SDE, with an update step given by:

$$z_t^c \leftarrow z_t^\# + \eta_t \, \nabla \log p_t(z_t^\#) + \sqrt{2\eta_t} \, \epsilon, \qquad \text{where } \epsilon \sim \mathcal{N}(0, I). \tag{5}$$

**Remark 1** (Vanilla corrector may disturb measurement consistency). *While the Langevin corrector is effective in reducing off-stationarity, the vanilla Langevin update may disturb measurement consistency $r(z_t) \coloneqq L(z_t, y)$ imposed by the LDM inverse solver. A first-order Taylor expansion of $r$ after the Langevin update is given by:*

$$r(z_t + \Delta z_t) \approx r(z_t) + \nabla_{z_t} r(z_t) \, \Delta z_t, \tag{6}$$

*where $\Delta z_t = \eta_t \, \nabla \log p_t(z_t^\#) + \sqrt{2\eta_t} \, \epsilon$ denotes Langevin step. Even when higher-order terms are neglected, the measurement consistency is perturbed since the first-order term $\nabla_{z_t} r(z_t) \, \Delta z_t \neq 0$ in general, that is, $\mathbb{E}[r(z_t + \Delta z_t)] \neq r(z_t)$.*

Although the instability of LDM-based inverse solvers remains a persistent challenge to be addressed, ensuring measurement fidelity is essential for faithful signal reconstruction in inverse problems. However, as noted in Remark 1, even neglecting higher-order terms, the vanilla Langevin update generally perturbs measurement consistency. This motivates us to propose *Measurement-Consistent Langevin Corrector* (MCLC), which applies an orthogonal projection at each Langevin update step onto the current measurement-consistent subspace. The MCLC update takes the following form:

$$z_t^c \leftarrow z_t^\# + \eta_t \cdot P_{\perp g_t} \, s_\theta(z_t^\#, t) + \sqrt{2\eta_t} \cdot P_{\perp g_t}(\epsilon), \qquad \text{where } g_t \coloneqq \frac{\nabla_{z_t} r(z_t)}{\|\nabla_{z_t} r(z_t)\|}. \tag{7}$$

Here, $P_{\perp g} = (I - gg^T)$ denotes projection of $v$ onto the orthogonal complement of $g$.

MCLC restricts the correction step to the orthogonal complement of the measurement-consistent gradient. Consequently, it preserves measurement consistency up to the first-order Taylor expansion (*i.e.*, $\nabla_{z_t} r(z_t) \Delta z_t = 0$). Even if higher-order terms are taken into account, MCLC still guarantees that the perturbation $\Delta z_t$ after the Langevin update can be bounded in terms of the step size.

**Theorem 1.** *The projected Langevin update onto the orthogonal complement of the measurement gradient decreases the KL divergence while preserving measurement consistency up to a controlled bound. Formally, if the update satisfies*

$$\mathbb{E}[\|\Delta z_t\|^2] \leq k < 1, \tag{8}$$

*then the expected perturbation of measurement consistency follows:*

$$\mathbb{E}[\Delta r] \leq Ck + O(k), \tag{9}$$

*for some constant $C > 0$ depending on the local smoothness of $r$.*

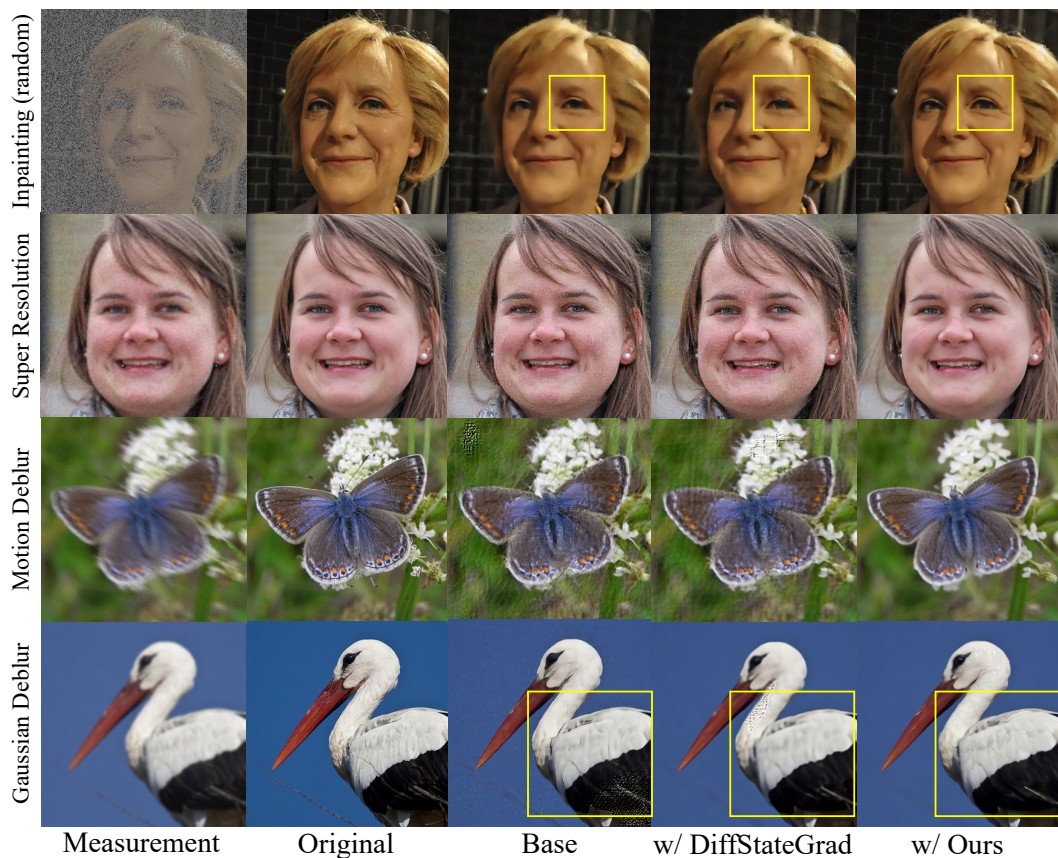

Figure 5: **Qualitative comparison of base latent diffusion inverse solvers and their plug-in versions with DiffStateGrad (Zirvi et al., 2025) and MCLC (ours)**. Proposed MCLC effectively alleviates artifacts and enhances reconstruction quality. The baseline used for visualization is ReSample (top two rows) and PSLD (bottom two rows).

In particular, as $k$ is controlled by the step size $\eta_t$, $\mathbb{E}[\Delta r]$ can be bounded at each timestep by selecting $\eta_t$ appropriately, thereby preserving measurement consistency while reducing the KL divergence. Theorem 1 suggests that latents deviated from $p_t$ can be pushed back toward the stationary distribution in terms of the KL divergence, while preserving measurement consistency within a controlled error. Detailed proofs are given in Appendix. A. Recall Assumption 1, which regards problematic latent updates that may lead to undesirable solutions as off-stationary deviations (*i.e.*, instability). Under this characterization, MCLC serves as a correction scheme that counteracts such deviations, stabilizing the LDM-based inverse solvers while maintaining measurement fidelity. This results in a substantial reduction of artifacts and improved reconstruction quality across LDM-based inverse solvers. The conceptual overview of MCLC is illustrated in Fig. 4. The algorithm is detailed in Algorithm 1.

## 4 EXPERIMENTS

**Experimental setup.** We evaluate our method by plugging it into existing LDM-based inverse solvers, including Latent DPS (LDPS) (Chung et al., 2023), PSLD (Rout et al., 2023), ReSample (Song et al., 2024), and LatentDAPS (Zhang et al., 2025a). As noted in Sec. 1, since we focus on the potential of zero-shot inverse solvers, all experiments adopt Stable Diffusion v1.5 (SD v1.5) as the underlying latent diffusion model, rather than domain-specific LDMs. For reproducibility, further details—including the integrated algorithms (*i.e.*, each solver combined with our corrector), task- and solver-specific hyperparameters of corrector, as well as the configurations of solvers, samplers, and other settings—are provided in the Appendix. C.

We benchmark the method across both linear and nonlinear inverse problems using two image datasets, FFHQ (Karras et al., 2019) and ImageNet (Deng et al., 2009). Following the LatentDAPS

Table 1: **Quantitative comparison for linear and nonlinear tasks on FFHQ and ImageNet.** MCLC improves overall performance across diverse base methods, demonstrating compatibility while achieving impressive performance compared to each base method and DiffStateGrad.

| Task | Base | Method | FFHQ | | | | ImageNet | | | |
|---|---|---|---|---|---|---|---|---|---|---|
| | | | PSNR ($\uparrow$) | LPIPS ($\downarrow$) | FID ($\downarrow$) | P-FID ($\downarrow$) | PSNR ($\uparrow$) | LPIPS ($\downarrow$) | FID ($\downarrow$) | P-FID ($\downarrow$) |
| Gaussian Deblur | LDPS | Base | 27.61 | 0.349 | 100.10 | 93.55 | 25.04 | 0.407 | 120.79 | 108.52 |
| | | DiffState | 27.59 | 0.348 | 100.82 | 94.14 | 25.00 | 0.409 | 122.14 | 106.84 |
| | | Ours | **28.14** | **0.303** | **80.83** | **54.74** | **25.84** | **0.395** | **103.87** | **93.28** |
| | PSLD | Base | 27.84 | 0.314 | 89.18 | 90.54 | 25.52 | **0.371** | 104.86 | 108.76 |
| | | DiffState | 27.89 | 0.311 | 86.73 | 87.90 | 25.47 | 0.377 | 106.90 | 109.92 |
| | | Ours | **27.97** | **0.286** | **66.28** | **59.13** | **25.89** | 0.380 | **92.74** | **95.01** |
| | ReSample | Base | 26.44 | 0.368 | 75.17 | 148.11 | 24.15 | 0.404 | 83.90 | 135.07 |
| | | DiffState | 26.05 | 0.396 | **74.03** | 140.76 | 24.12 | 0.417 | **79.57** | 133.22 |
| | | Ours | **27.25** | **0.353** | 78.38 | **106.16** | **25.19** | **0.378** | 81.71 | **123.00** |
| | LatentDAPS | Base | 27.51 | **0.348** | **99.53** | **120.56** | 25.41 | 0.375 | **112.54** | 111.22 |
| | | DiffState | **27.52** | 0.349 | 106.04 | 122.03 | **25.47** | **0.374** | 113.57 | **110.81** |
| | | Ours | 27.42 | 0.349 | 100.58 | 123.06 | 25.42 | 0.376 | 116.35 | 111.04 |
| Motion Deblur | LDPS | Base | 26.54 | 0.390 | 118.77 | 112.74 | 23.93 | 0.451 | 154.30 | 121.79 |
| | | DiffState | 26.56 | 0.387 | 118.71 | 110.60 | 24.08 | 0.447 | 154.48 | 122.01 |
| | | Ours | **27.45** | **0.318** | **82.94** | **55.55** | **24.79** | **0.424** | **119.65** | **97.68** |
| | PSLD | Base | 26.87 | 0.343 | 106.34 | 102.60 | 24.54 | 0.407 | 141.67 | 121.41 |
| | | DiffState | **26.88** | 0.340 | 107.95 | 102.28 | 24.60 | 0.401 | 138.91 | 121.98 |
| | | Ours | 26.86 | **0.308** | **74.64** | **60.05** | **24.94** | **0.387** | **99.21** | **93.49** |
| | ReSample | Base | 22.45 | 0.635 | 108.14 | 174.52 | 21.42 | **0.589** | 156.25 | 158.90 |
| | | DiffState | 23.16 | 0.623 | 104.42 | 133.68 | 21.58 | 0.633 | **101.84** | 129.92 |
| | | Ours | **24.24** | **0.588** | **102.02** | **118.87** | **22.33** | 0.616 | 102.81 | **118.81** |
| | LatentDAPS | Base | **24.83** | **0.491** | **160.50** | **198.21** | 23.06 | 0.503 | 183.22 | 142.50 |
| | | DiffState | 24.60 | 0.494 | 163.28 | 199.27 | 23.11 | **0.500** | 184.03 | **141.35** |
| | | Ours | 24.65 | 0.496 | 165.19 | 199.74 | **23.31** | 0.502 | **176.77** | 141.66 |
| Super Resolution (4×) | LDPS | Base | 28.47 | 0.301 | 78.08 | 69.66 | 26.22 | 0.401 | 118.33 | 107.19 |
| | | DiffState | **28.58** | 0.299 | 77.40 | 68.14 | **26.25** | 0.401 | 115.18 | 106.66 |
| | | Ours | 28.34 | **0.283** | **74.78** | **58.55** | 26.20 | **0.388** | **114.96** | **101.17** |
| | PSLD | Base | 27.69 | 0.265 | 63.95 | 63.47 | 25.21 | 0.373 | **88.92** | 95.21 |
| | | DiffState | **27.71** | **0.262** | 64.23 | 63.08 | **25.37** | **0.363** | 91.60 | **94.75** |
| | | Ours | 27.33 | 0.267 | **62.12** | **58.13** | 25.02 | 0.384 | 94.95 | 96.22 |
| | ReSample | Base | 26.40 | 0.347 | 70.16 | 133.15 | 23.37 | 0.435 | 72.22 | 134.39 |
| | | DiffState | 25.22 | 0.464 | 79.94 | 135.19 | 22.51 | 0.531 | 76.10 | 133.53 |
| | | Ours | **28.32** | **0.236** | **53.85** | **78.08** | **25.76** | **0.318** | **68.21** | **104.82** |
| | LatentDAPS | Base | **28.61** | **0.269** | 72.70 | **73.37** | 26.21 | **0.287** | **67.46** | **89.38** |
| | | DiffState | 28.58 | 0.270 | **72.08** | 73.92 | 26.22 | 0.288 | 71.16 | 90.32 |
| | | Ours | 28.50 | 0.270 | 75.25 | 73.79 | **26.25** | 0.290 | 70.86 | 89.78 |
| Inpainting (Random) | LDPS | Base | 31.22 | 0.171 | 48.88 | 83.30 | 28.78 | 0.238 | 57.12 | 119.25 |
| | | DiffState | 31.23 | 0.170 | 48.45 | 83.81 | 29.03 | **0.230** | 54.31 | 94.99 |
| | | Ours | **31.28** | **0.169** | **48.05** | **81.68** | **29.18** | 0.231 | **52.70** | **93.97** |
| | PSLD | Base | 30.14 | 0.222 | 58.84 | 79.59 | 27.85 | 0.300 | 70.51 | 96.30 |
| | | DiffState | 29.93 | 0.231 | 66.06 | 88.54 | 27.87 | 0.298 | 68.78 | 96.86 |
| | | Ours | **30.73** | **0.185** | **49.80** | **72.69** | **29.18** | **0.230** | **52.70** | **94.35** |
| | ReSample | Base | 27.27 | 0.374 | 103.17 | 133.80 | 24.84 | 0.489 | 132.87 | 151.96 |
| | | DiffState | 27.33 | 0.390 | 102.81 | 132.11 | 24.91 | 0.507 | 131.61 | 144.68 |
| | | Ours | **29.35** | **0.235** | **75.65** | **108.27** | **26.50** | **0.388** | **99.71** | **123.27** |
| | LatentDAPS | Base | 28.26 | **0.224** | 68.43 | **73.09** | 26.43 | 0.254 | 69.80 | 92.72 |
| | | DiffState | 28.27 | 0.225 | 65.99 | 73.10 | 26.35 | 0.256 | 69.89 | 92.17 |
| | | Ours | **28.29** | 0.225 | **64.49** | 73.70 | **26.51** | **0.253** | **68.84** | **90.99** |
| HDR | ReSample | Base | 25.75 | 0.197 | 80.44 | 85.46 | 24.91 | 0.218 | 74.43 | 91.78 |
| | | DiffState | **25.90** | 0.214 | 83.85 | 91.36 | **24.93** | **0.215** | 72.41 | 92.55 |
| | | Ours | 25.55 | **0.196** | **77.64** | **82.18** | 24.79 | 0.217 | **71.67** | **87.39** |
| | LatentDAPS | Base | 24.13 | 0.294 | **91.02** | 101.29 | **23.32** | 0.306 | 107.01 | 112.89 |
| | | DiffState | 24.12 | 0.295 | 91.57 | 100.64 | 23.31 | 0.306 | **104.67** | 112.96 |
| | | Ours | **24.52** | **0.293** | 91.12 | **99.03** | 23.25 | **0.305** | 106.91 | **112.76** |
| Nonlinear Deblur | ReSample | Base | 24.65 | **0.431** | 151.79 | 153.50 | 23.01 | 0.423 | 195.66 | 135.51 |
| | | DiffState | 24.61 | 0.432 | **142.54** | 154.07 | **23.10** | 0.424 | **182.52** | **135.05** |
| | | Ours | **24.84** | 0.449 | 148.42 | 159.76 | 22.96 | **0.421** | 185.63 | 136.69 |
| | LatentDAPS | Base | 24.48 | 0.481 | 152.40 | 152.80 | 22.58 | 0.515 | 186.81 | 148.47 |
| | | DiffState | **24.58** | 0.480 | 149.67 | 150.81 | **22.65** | 0.511 | **179.25** | **147.14** |
| | | Ours | 24.43 | **0.475** | **147.81** | **148.72** | 22.38 | **0.508** | 186.76 | 150.35 |

Table 2: **Quantitative results on AFHQ-val 1K using TReg** (Kim et al., 2025b). To further validate the compatibility of MCLC with recent advanced solvers, we report comparisons with and without MCLC using TReg, a recent latent diffusion inverse solver.

| | Gaussian Deblur | | | Super Resolution (16x) | | |
|---|---|---|---|---|---|---|
| | PSNR (↑) | LPIPS (↓) | FID (↓) | PSNR (↑) | LPIPS (↓) | FID (↓) |
| TReg | 20.84 | 0.476 | 37.12 | 18.39 | **0.633** | 44.91 |
| TReg w/ Ours | **21.33** | **0.456** | **27.62** | **19.15** | 0.646 | **33.86** |

Table 3: **Quantitative results on 100 FFHQ images using FlowChef** (Patel et al., 2025). To evaluate MCLC's compatibility with flow-based models, we report drop-in improvements on FlowChef, a recent flow-based inverse solver. We use Stable Diffusion v3 as the underlying flow model.

| | Motion Deblur | | | Super Resolution 12x (Bicubic) | | |
|---|---|---|---|---|---|---|
| Method | PSNR (↑) | LPIPS (↓) | FID (↓) | PSNR (↑) | LPIPS (↓) | FID (↓) |
| FlowChef | 22.58 | 0.519 | 185.44 | **25.30** | 0.501 | 174.51 |
| FlowChef w/ Ours | **26.01** | **0.353** | **100.40** | 24.85 | **0.424** | **130.70** |

data protocol (Zhang et al., 2025a), we selected 100 validation images from each dataset (*i.e.*, the first 100 images of each validation set). For linear tasks, we consider (1) Super Resolution with a downscaling factor of 4 using a bicubic resizer, (2) Gaussian Deblur with a $121 \times 121$ kernel and standard deviation $\sigma = 3.0$, (3) Motion Deblur with a $121 \times 121$ kernel and standard deviation $\sigma = 0.5$, and (4) inpainting with 70% random pixel dropout. For nonlinear tasks, we evaluate (5) High Dynamic Range (HDR) reconstruction and (6) Nonlinear Deblur. All experiments are conducted at a resolution of $512 \times 512$ with a Gaussian noise scale fixed to $\sigma = 0.03$, except for nonlinear deblurring, where the blur kernel is generated for $256 \times 256$. For the evaluation metric, we adopt PSNR, LPIPS, and FID following previous works. In addition, we introduce Patch-FID (P-FID) to more effectively quantify regional artifacts by comparing patch-wise statistics. To implement P-FID, we split each image into $3 \times 3$ patches and treat them as individual images to calculate the FID score.

**Experimental results.** We quantitatively demonstrate the benefits of integrating our corrector into existing LDM-based inverse solvers in Table 1. The corrector shows overall performance improvements without modifying the design of each solver. Compared to DiffState-Grad (Zirvi et al., 2025), which is also designed as a plug-and-play module and relies on the linear manifold assumption, our method outperforms it overall in latent diffusion. Notably, when integrated into basic solvers such as LDPS and PSLD, it elevates their performance to be comparable to, or even surpass, recent advanced solvers (*e.g.*, LatentDAPS), highlighting its ef-

| Method | Super Resolution (4×) | | | Gaussian Deblur | | |
|---|---|---|---|---|---|---|
| | PSNR | LPIPS | FID | PSNR | LPIPS | FID |
| MPGD | 27.25 | 0.280 | 77.90 | 28.69 | 0.260 | 75.52 |
| SILO | 25.91 | 0.251 | 70.11 | 25.79 | 0.276 | 76.18 |
| PSLD w/ Ours | 27.33 | 0.267 | 62.12 | 27.97 | 0.286 | 66.28 |
| ReSample w/ Ours | 28.32 | 0.236 | 53.85 | 27.25 | 0.353 | 78.38 |

Table 4: **Quantitative comparison with non-pluggable approaches.** MCLC, a fully pluggable method, achieves competitive overall performance while offering broad applicability.

fectiveness. Additionally, Fig. 5 shows that MCLC produces faithful reconstructions and provides more reliable solutions. We further show its stability in Sec. B.2. Since MCLC preserves the measurement consistency of the base solver, PSNR improvements differ across solvers, from modest to noticeably higher, according to the stabilization effect. At the same time, perceptual metrics such as LPIPS, FID, and P-FID show substantial gains, reflecting MCLC's strong effect in alleviating artifacts and enhancing reconstruction quality. For LatentDAPS, performance differences are marginal because its specific design breaks the reverse diffusion dynamics by re-initializing each iteration with annealed noise, whereas MCLC is intended to stabilize the reverse dynamics of LDM-based inverse solvers. Despite the limited compatibility that stems from the particular design of LatentDAPS, several tasks still exhibit gains. Additional discussion is provided in Appendix B.9. Importantly, this does not imply limited applicability of MCLC. In Table 2 and 3, MCLC shows clear and consistent gains when applied to a recent LDM-based (Kim et al., 2025b) and a flow-based solver (Patel et al., 2025), highlighting the future applicability of MCLC. See Appendix B.1 for additional results and details.

We also compare our method against non-pluggable approaches for artifact mitigation in LDMs, including MPGD (He et al., 2024), and SILO (Raphaeli et al., 2025), which employs a trained degradation operator to avoid decoder backpropagation. Table 4 demonstrates the effectiveness of

our method, even though it is employed as a plug-and-play module. In contrast, non-pluggable approaches exhibit relatively lower performance, often leaving artifacts and degraded quality or compromising the measurement consistency. Further discussions are provided in Appendix. B.4.

**Analysis on computational cost.** To quantify the additional computational cost of MCLC, we report the runtime and memory analyses across three solvers (LDPS, PSLD, and Resample). As shown in Fig. 6, the additional wall-clock time introduced by MCLC is modest for LDPS and PSLD (about 3%). For ReSample, the increase is more noticeable because the base solver already performs extensive inner gradient-descent loops for hard data consistency. Even in this case, the overall overhead remains manageable, and MCLC provides substantial improvements in reconstruction quality, as shown in Table 1. This efficiency stems from the fact that MCLC

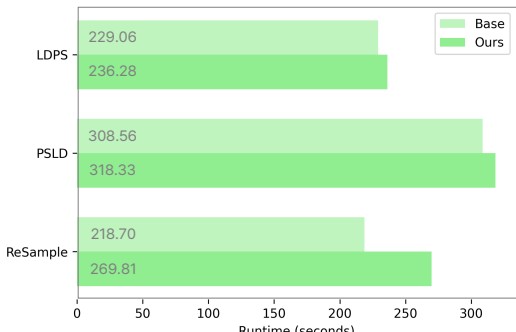

Figure 6: **Analysis of the additional runtime incurred when applying MCLC.**

requires only the LDM forward pass and simple algebraic operations, without any backward computation. While backward computation requires several times more computational cost for large prior models, MCLC avoids this backward process by reusing the gradient from the measurement-consistency step. As a result, MCLC incurs no additional memory overhead, as confirmed by the peak memory usage across all solvers. For all experiments, we use a single NVIDIA RTX A6000.

**Measurement-consistent correction scheme.** Our theoretical analysis reveals that instability can arise in the reverse dynamics of LDM-based inverse solvers, from which we propose Langevin correction as a tool for mitigating such instability. However, applying it directly can lead to suboptimal solutions: the update pulls the dynamics toward high-probability regions of the prior while disturbing measurement consistency, producing plausible but data-inconsistent solutions. As shown in Table 5, MCLC overcomes this issue by maintaining the measurement consistency while stabilizing the dynamics.

| Method | y-PSNR (↑) | PSNR (↑) | LPIPS (↓) | FID (↓) |
|---|---|---|---|---|
| Base | 34.30 | 28.47 | 0.301 | 78.08 |
| Ours (w/o MC) | 31.59 | 27.50 | **0.277** | 76.78 |
| Ours (w/ MC) | **33.23** | **28.34** | 0.283 | **74.78** |

Table 5: **Comparison between MCLC and LC.** 'w/ MC" indicates measurement-consistent correction and "w/o MC" indicates direct Langevin correction (LC). y-PSNR is the PSNR between predicted and ground-truth measurements. Results are on FFHQ 4× SR. using LDPS.

## 5 FURTHER ANALYSIS OF ARTIFACTS IN LATENT SPACE

In Sec. 3 and 4, we demonstrate that MCLC effectively mitigates most types of artifacts by considering them as instances of off-stationarity. Nevertheless, certain artifacts occur even when advanced solvers are employed. The artifacts, known as blob artifacts (Raphaeli et al., 2025), are characterized by localized distortions in the reconstructed results, as shown in Fig. 7. In this section, we analyze the special case and discuss possible ways to address them.

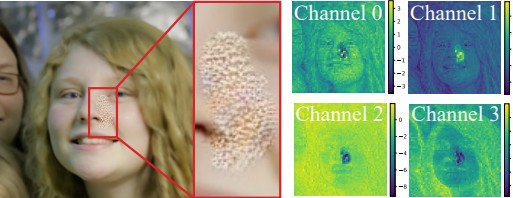

Figure 7: **Analysis on blob artifacts.** Blob artifacts in the decoded image arise when scaled-outliers exist in the latent.

**Where do these blob artifacts originate from?** Previous works (Song et al., 2024; Raphaeli et al., 2025) have mentioned that the artifact is caused by the decoder, which makes the gradient problematic. To clearly investigate the origin of these artifacts, we perform a more detailed analysis. As a first step, we examine whether the artifacts indeed originate from backpropagation through the decoder by analyzing the gradients $\frac{\partial L}{\partial x_{0|t}}$ and $\frac{\partial L}{\partial z_{0|t}} = J_{\mathcal{D}}^{\top} \frac{\partial L}{\partial x_{0|t}}$, where $J_{\mathcal{D}}$ denotes the Jacobian of the decoder. Then, we observe that backpropagation through the decoder makes the signal that is unrelated to the measurement gradient $\frac{\partial L}{\partial x_{0|t}}$ (see Appendix. B for details). This observation indicates that artifacts can indeed arise from decoder backpropagation. In addition, the blob artifacts tend to occur when *scaled-outliers* are present in the latent (see Fig. 7). We define the scaled-outlier as a localized latent

region whose values are substantially higher or lower than its surroundings, *i.e.*, deviations outside the typical latent range. This shows that the blob artifacts result from scaled-outliers.

**Clarification of Setups.** The latent *before* the measurement update is denoted by $z_{0|t}$ and the latent *after* applying the measurement gradient through the decoder's Jacobian $J_{\mathcal{D}}$ is denoted by $z_{0|t}(y)$. In the following, we analyze relationship between scaled-outliers and the decoder's Jacobian in detail.

**Why do scaled-outliers emerge?** Since scaled-outliers consistently appear in specific regions, we hypothesize that the decoder's Jacobian $J_{\mathcal{D}}$ selectively amplifies certain latent directions. To examine this, we analyze the principal eigenvector of $J_{\mathcal{D}} J_{\mathcal{D}}^{\top}$. Figure 8 shows that scaled-outlier regions in the updated latent $z_{0|t}(y)$ are strongly correlated with the regions amplified by the decoder's Jacobian $J_{\mathcal{D}}$. This reveals that scaled-outliers arise from Jacobian amplification.

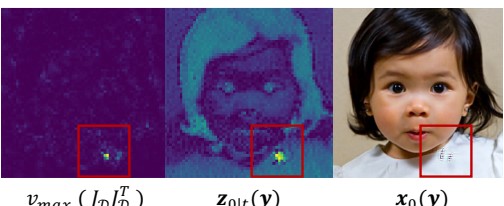

$v_{max}(J_{\mathcal{D}}J_{\mathcal{D}}^{T})$     $z_{0|t}(y)$     $x_0(y)$

Figure 8: **Analysis on scaled-outliers.** Scaled-outlier regions in $z_{0|t}(y)$ are aligned with regions amplified by the decoder's Jacobian $J_{\mathcal{D}}$.

**Why do such artifacts remain?** In principle, such artifacts should be heavily penalized by the loss function and thus eliminated, yet they persist. We find that when the latent $z_{0|t}$ already contains scaled-outlier regions before the update, the decoder's Jacobian amplifies the gradient in the surrounding area. To verify this effect, we artificially inject a $3 \times 3$ scaled-outlier latent patch into the center of the input latent $z_{0|t}$. As shown in Fig. 9, the decoder's Jacobian $J_{\mathcal{D}}$ exhibits strong amplification around the injected center region. In summary, when the input latent $z_{0|t}$ contains scaled-outliers, the decoder's Jacobian amplifies these, which then reappear in

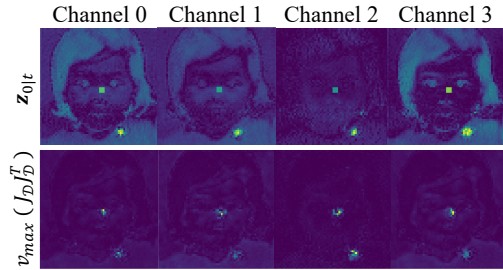

Channel 0   Channel 1   Channel 2   Channel 3

Figure 9: **Relation between scaled-outliers and** $J_{\mathcal{D}}$. By artificially injecting a scaled-outlier patch into $z_{0|t}$, we confirm that $J_{\mathcal{D}}$ amplifies such regions when the outliers are present in the input.

the updated latent $z_{0|t}(y)$. During the reverse sampling process, scaled-outliers are further magnified, which prevents their elimination and eventually manifests as blob artifacts.

**How can we handle it?** According to the analyses above, if the input latent contains no scaled-outliers, amplification does not occur. Interestingly, we find that the latent space of SD v1.5, which is widely used in latent diffusion inverse solvers, inherently contains scaled-outlier regions as confirmed by the encoding of original images (see Fig. 17). Although low-magnitude outliers do not manifest as visible artifacts when decoded, they can be amplified during the reverse sampling process of latent diffusion inverse solvers and eventually appear as visible artifacts. As shown in the last two rows of Fig. 5 and Sec. B.4, our proposed method suppresses this amplification, thereby removing the blob artifacts. However, since these outliers are inherent to the target latent distribution itself, they cannot be fully eliminated. A straightforward alternative is to adopt latent spaces that are free from such outliers, for example, those of SDXL (see Fig. 17). Another possible approach is to incorporate pixel-level optimization to avoid blob artifacts, as demonstrated in P2L (Chung et al., 2024).

# 6 CONCLUSION

In this work, we provide new theoretical insight and a principled correction scheme that improve the understanding and stability of latent diffusion inverse solvers. We identify their instability as a reverse-dynamics discrepancy from true diffusion and address it with the *Measurement-Consistent Langevin Corrector (MCLC)*, which corrects solver dynamics while maintaining measurement consistency. As a plug-and-play module, MCLC remedies LDM-based inverse solvers by stabilizing them without relying on the linear manifold assumption. This results in more faithful and stable solutions in LDM-based inverse solvers. We believe our findings offer meaningful conceptual advances and a theoretically grounded tool in LDM-based inverse problem solving, and hope our work inspires further research on stable zero-shot diffusion-based inverse solvers. Although MCLC provides meaningful progress toward a zero-shot inverse solver, choosing the corrector step size remains non-trivial, suggesting adaptive strategies as a promising future direction.

REPRODUCIBILITY STATEMENT

For reproducibility, we provide the details of algorithms and hyperparameters used in our experiments in Sec. C. The experimental settings for measuring KL divergence are described in Sec. C.1, and the full algorithms are presented in Sec. C.2. In addition, we include the complete configurations of each latent diffusion solver, along with the specifications of our corrector and its integration, in Sec. C.3, C.4, C.5, and C.6. We build on the open-source Stable Diffusion model (`https://huggingface.co/botp/stable-diffusion-v1-5`). All latent diffusion inverse solvers used in our work, including DPS(`https://github.com/DPS2022/diffusion-posterior-sampling`), PSLD(`https://github.com/LituRout/PSLD`), ReSample(`https://github.com/soominkwon/resample`), and DAPS(`https://github.com/zhangbingliang2019/DAPS`), as well as competing methods such as DiffStateGrad(`https://github.com/Anima-Lab/DiffStateGrad`), MPGD(`https://github.com/KellyYutongHe/mpgd_pytorch/`), and SILO(`https://github.com/ronraphaeli/SILO`), have publicly available implementations. If our paper is accepted, we will release our code to ensure full reproducibility.

ETHICS STATEMENT

Our experiments involve human face image data from the FFHQ dataset (Karras et al., 2019), which is a publicly available and widely used benchmark in inverse problems. FFHQ is released under the Creative Commons license, and our work does not involve any personally identifiable information or sensitive private data.

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

# APPENDIX

In this appendix, we first present the detailed proofs and derivations of the theoretical results (Appendix A). We then provide additional experiments and visualization results (Appendix B). Finally, we summarize the implementation details for reproducibility (Appendix C). The contents are organized as follows:

# CONTENTS

# A  PROOFS

**Notation.** We summarize the notation used in the proofs:

- $\nabla f$: gradient of a scalar function $f$, i.e., $(\partial_{x_1} f, \ldots, \partial_{x_d} f)^\top$.

- $\nabla \cdot F$: divergence of a vector field $F = (F_1, \ldots, F_d)$, i.e., $\sum_{i=1}^{d} \partial_{x_i} F_i$.

- $\Delta z_t$: increment (update change) of the variable $z_t$ in algorithmic updates, not to be confused with the Laplacian operator.

- $\mathrm{KL}(q \| p)$: Kullback–Leibler divergence, defined as $\int q(x) \log\left(\frac{q(x)}{p(x)}\right) dx$.

**Proposition 1** (Langevin Corrector). *Fix a timestep $t$ and let $p_t$ be a target distribution. Consider the continuous corrector process $\{Z_t^c\}_{c \geq 0}$ initialized with $Z_t^0 \sim q_t^{\#}$. The process evolves according to the Langevin dynamics with frozen target $p_t$: $dZ_t^c = \nabla \log p_t(Z_t^c) dc + \sqrt{2} dW_c$. Let $q_t^c$ denote the distribution of $Z_t^c$. Then, the KL divergence monotonically decreases along the process, unless $q_t^c = p_t$, in which case equality holds:*

$$\mathcal{D}_{\mathrm{KL}}(q_t^c \| p_t) \leq \mathcal{D}_{\mathrm{KL}}(q_t^{\#} \| p_t), \qquad \forall c \geq 0 \tag{10}$$

*Proof.* We recall that the Langevin Corrector dynamics is given by

$$dZ_c = \nabla \log p_t(Z_c) \, dc + \sqrt{2} \, dW_c, \qquad c \geq 0, \tag{11}$$

where $p_t$ is a fixed target distribution at timestep $t$. In this corrector process, keeping $t$ fixed, the KL divergence with respect to $p_t$ decreases monotonically (Durmus & Moulines, 2019; Vempala & Wibisono, 2019). For notational simplicity, when time $t$ is fixed in the corrector process, we write $Z_t^c$ as $Z_c$ below.

*Step 1. From SDE to Fokker–Planck PDE.* Let $\{X_t\}_{t \geq 0}$ be a stochastic process. At each time $t$, the random variable $X_t$ evolves according to the stochastic differential equation:

$$dX_t = a(X_t, t) \, dt + b(X_t, t) \, dW_t, \tag{12}$$

where $a(X_t, t)$ denotes the drift term, $b(X_t, t)$ the diffusion coefficient, and $W_t$ a standard Wiener process. The random variable $X_t$ has a distribution, denoted by $q_t$, which evolves over time. The time evolution of this distribution $q_t$ (*i.e.*, the law of $X_t$) is governed by the Fokker–Planck equation:

$$\frac{\partial q(x, t)}{\partial t} = -\nabla \cdot \big(a(x, t) \, q(x, t)\big) + \nabla \cdot \big(D(x, t) \, \nabla q(x, t)\big). \tag{13}$$

where $D(x, t) = \frac{1}{2} b(x, t) \, b(x, t)^{\top}$.

*Step 2. Langevin Corrector.* From Eq. (12), the Langevin Corrector dynamics can be written as the following SDE:

$$dZ_c = \nabla \log p_t(Z_c) \, dc + \sqrt{2} \, dW_c, \qquad c \geq 0, \tag{14}$$

with drift $a(x) = \nabla \log p_t(x)$ and isotropic diffusion $b(x) = \sqrt{2} \, I$, where $p_t$ is a fixed target distribution at timestep $t$. Let $q_t^c$ denote the distribution of $Z_c$, which evolves along the corrector process with the timestep $t$ fixed. Then, its evolution with respect to the corrector-time variable $c$ is also described by the Fokker–Planck equation:

$$\frac{\partial q_t^c}{\partial c} = \nabla \cdot \left( q_t^c \, \nabla \log \frac{q_t^c}{p_t} \right). \tag{15}$$

*Step 3. Evolution of the KL divergence along the corrector process.* Consider the KL divergence

$$\mathcal{F}[q_t^c] = \mathrm{KL}(q_t^c \| p_t) = \int q_t^c(x) \log \frac{q_t^c(x)}{p_t(x)} dx. \tag{16}$$

Differentiating with respect to $c$ yields,

$$\frac{d}{dc} \mathcal{F}[q_t^c] = \int \left( 1 + \log \frac{q_t^c}{p_t} \right) \frac{\partial q_t^c}{\partial c} \, dx \tag{17}$$

$$= \int \left( 1 + \log \frac{q_t^c}{p_t} \right) \nabla \cdot \left( q_t^c \nabla \log \frac{q_t^c}{p_t} \right) dx. \quad \text{(by Eq. (15))} \tag{18}$$

$$= -\int q_t^c(x) \left\| \nabla \log \frac{q_t^c(x)}{p_t(x)} \right\|^2 dx. \tag{19}$$

Then, following the Langevin Corrector dynamics, $\frac{d}{dc}\mathcal{F}[q_t^c]$ is written as Eq. (19). Since $q_t^c(x) \geq 0$ and $\left\|\nabla \log \frac{q_t^c(x)}{p_t(x)}\right\|^2 \geq 0$, the right-hand side is non-positive. Moreover, it equals zero if and only if $q_t^c = p_t$; otherwise it is strictly negative:

$$\frac{d}{dc}\mathrm{KL}(q_t^c \| p_t) \ \leq \ 0. \tag{20}$$

Therefore, the KL divergence between $q_t^c$ and the fixed target distribution $p_t$ monotonically decreases along the corrector process. In particular, the KL divergence of the corrected distribution is no larger than that of the problematic distribution obtained from the inverse solver update.

$$\mathrm{KL}(q_t^c \| p_t) \leq \mathrm{KL}(q_t^\# \| p_t), \qquad \forall c \geq 0. \tag{21}$$

This proves the proposition. $\square$

**Lemma 1.** *Let $U \sim \mathcal{N}(\mu, \Sigma)$ be a Gaussian random vector in $\mathbb{R}^d$, in the high-dimensional setting where $d$ is large. Then there exists a universal constant $\kappa > 0$ such that*

$$\mathbb{E}[\|U\|^3] \ \leq \ \kappa(\mathbb{E}[\|U\|^2])^{3/2}. \tag{22}$$

*Proof.* Let h: $\mathbb{R}_+ \to \mathbb{R}$ be twice differentiable, and suppose that $h''(x) \leq \Lambda$ for $x$ in the support of a random variable $Y$. Define

$$g(x) = h(x) - \tfrac{1}{2}\Lambda x^2. \tag{23}$$

Since $g''(x) = h''(x) - \Lambda \leq 0$, the function $g$ is concave. By Jensen's inequality,

$$\mathbb{E}[g(Y)] \leq g(\mathbb{E}[Y]). \tag{24}$$

Then,

$$\mathbb{E}[h(Y) - \tfrac{1}{2}\Lambda Y^2] \ \leq \ h(\mathbb{E}[Y]) - \tfrac{1}{2}\Lambda\mathbb{E}[Y]^2 \tag{25}$$

$$\mathbb{E}[h(Y)] - h(\mathbb{E}[Y]) \ \leq \ \underbrace{\tfrac{1}{2}\Lambda\,\mathbb{E}[Y^2] - \tfrac{1}{2}\Lambda\,\mathbb{E}[Y]^2}_{\tfrac{1}{2}\Lambda\,\mathrm{Var}(Y)}. \tag{26}$$

Let $Y = \|U\|^2$ and $h(x) = x^{3/2}$. Then,

$$h''(x) = \tfrac{3}{4}x^{-1/2}, \quad \text{so } \Lambda = O(x^{-1/2}). \tag{27}$$

Hence, by Eq. (26),

$$\mathbb{E}[h(Y)] - h(\mathbb{E}[Y]) \ \leq \ O\left(\frac{\mathrm{Var}(Y)}{\sqrt{\mathbb{E}[Y]}}\right) \tag{28}$$

$$= \ O\left(\frac{\mathrm{Var}(Y)}{\mathbb{E}[Y]^2}\mathbb{E}[Y]^{3/2}\right) \tag{29}$$

If the $U \in \mathbb{R}^d$ follows a Gaussian distribution in high dimensions, $\|U\|^2$ is concentrated around its mean, so that $\mathrm{Var}(Y)$ grows much more slowly than $\mathbb{E}[Y]^2$ (while $\mathrm{Var}(Y)$ grows only linearly with $d$, $(\mathbb{E}[Y])^2$ grows quadratically). Therefore, we may write

$$\mathbb{E}[h(Y)] - h(\mathbb{E}[Y]) \ \leq \ \delta\,\mathbb{E}[Y]^{3/2}, \qquad \text{for some small } \delta > 0, \tag{30}$$

where $\delta$ converges to 0 as the dimension $d \to \infty$.

Consequently,

$$\mathbb{E}[Y^{3/2}] \leq \underbrace{(1 + \delta)}_{:=\kappa}\,\mathbb{E}[Y]^{3/2} \tag{31}$$

Hence, there exists a universal constant $\kappa > 0$ such that

$$\mathbb{E}[\|U\|^3] \ \leq \ \kappa(\mathbb{E}[\|U\|^2])^{3/2}. \tag{32}$$

$\square$

**Theorem 1.** *The projected Langevin update onto the orthogonal complement of the measurement gradient decreases the KL divergence while preserving measurement consistency up to a controlled bound. Formally, if the update satisfies*

$$\mathbb{E}[\|\Delta \boldsymbol{z}_t\|^2] \leq k < 1, \tag{33}$$

*then the expected perturbation of measurement consistency follows:*

$$\mathbb{E}[\Delta r] \leq Ck + O(k), \tag{34}$$

*for some constant $C > 0$ depending on the local smoothness of $r$.*

*Proof.* The proof of Theorem 1 consists of two parts: (i) showing the decrease of the KL divergence, and (ii) establishing the measurement-consistency bound.

*Proof 1. KL divergence decrease.* As proved in Proposition 1, the Measurement-Consistent Langevin Corrector (MCLC) dynamics is written as:

$$dZ_c = P_{\perp \boldsymbol{g}} \nabla \log p_t(Z_c) \, dc + \sqrt{2} \, P_{\perp \boldsymbol{g}} \, dW_c, \tag{35}$$

where $P_{\perp \boldsymbol{g}} = I - \frac{gg^\top}{\|g\|^2}$ denotes the orthogonal projection onto the complement of the measurement gradient direction.

By the Fokker–Planck equation, the evolution of the distribution $q_t^c$ under the MCLC process is

$$\frac{\partial q_t^c}{\partial c} = \nabla \cdot \left( q_t^c \, P_{\perp \boldsymbol{g}} \nabla \log \frac{q_t^c}{p_t} \right). \tag{36}$$

Hence, the evolution of the KL divergence along the corrector process is

$$\frac{d}{dc} \mathcal{F}[q_t^c] = \int \left( 1 + \log \frac{q_t^c}{p_t} \right) \frac{\partial q_t^c}{\partial c} \, dx \tag{37}$$

$$= \int \left( 1 + \log \frac{q_t^c}{p_t} \right) \nabla \cdot \left( q_t^c \, P_{\perp \boldsymbol{g}} \nabla \log \frac{q_t^c}{p_t} \right) dx \quad \text{(by Eq. (36))} \tag{38}$$

$$= -\int q_t^c(x) \left\| P_{\perp \boldsymbol{g}} \nabla \log \frac{q_t^c(x)}{p_t(x)} \right\|^2 dx. \tag{39}$$

Then, the MCLC dynamics guarantees that the KL divergence monotonically decreases in the corrector step $c$, whenever the score difference $\nabla \log q_t^c - \nabla \log p_t$ has a non-zero component in the orthogonal complement of the measurement-consistent subspace:

$$\text{KL}(q_t^c \| p_t) \leq \text{KL}(q_t^\# \| p_t), \qquad \forall c \geq 0, \tag{40}$$

where, since $P_{\perp \boldsymbol{g}}$ is projection matrix, $\left\| P_{\perp \boldsymbol{g}} \nabla \log \frac{q_t^c(x)}{p_t(x)} \right\|^2 \geq 0$.

The score difference characterizes how the local geometry of the current corrected distribution $q_t^c$ deviates from that of the target distribution $p_t$. In other words, when the two distributions have divergence in the orthogonal complement of the measurement-consistent gradient, the KL divergence strictly decreases.

*Proof 2. Measurement consistency error bound.* Let $r : \mathbb{R}^d \to \mathbb{R}$ denote the measurement residual (*e.g.*, measurement-consistency loss). Let the MCLC step $\Delta z_t$:

$$\Delta z_{t,\perp} = \eta_t \, P_{\perp \boldsymbol{g}} s_t + \sqrt{2\eta_t} \, P_{\perp \boldsymbol{g}} \, \epsilon, \qquad \text{with } z_t \in \mathbb{R}^d, \ \epsilon \sim \mathcal{N}(0, I), \tag{41}$$

$$\Delta z_{t,\perp} = P_{\perp \boldsymbol{g}} \Big( \underbrace{\eta_t s_t + \sqrt{2\eta_t} \, \epsilon}_{:= \Delta z_t} \Big). \tag{42}$$

For convenience, denote the score as $s_t = \nabla \log p_t(z_t)$.

*Step 1. Bound of residual perturbation along the MCLC step.* By Taylor's expansion, the residual after one MCLC step can be expressed in terms of the residual before the step as:

$$\mathbb{E}[\, r(z_t + \Delta z_t)\,] = \mathbb{E}[\, r(z_t)\,] + \mathbb{E}[\nabla r(z_t)^\top \Delta z_t] + \tfrac{1}{2}\mathbb{E}[\Delta z_t^\top H \Delta z_t] + \mathbb{E}[O(r^3)], \tag{43}$$

where $H$ is the Hessian of $r(z_t)$, and $O(r^3)$ denotes the higher-order terms.

Then, the change in residual after one MCLC step is given by:

$$\underbrace{r(z_t + \Delta z_{t,\perp}) - r(z_t)}_{:=\Delta r(z_t)} = \underbrace{\nabla r(z_t)^\top \Delta z_{t,\perp}}_{=0} + \tfrac{1}{2} \Delta z_{t,\perp}^\top H \Delta z_{t,\perp} + O(r^3). \tag{44}$$

Using $\Delta z_t$, the $\Delta r(z_t)$ can be written as:

$$\Delta r(z_t) = \tfrac{1}{2} \Delta z_t^\top \underbrace{P_{g\perp}^\top H P_{g\perp}}_{:=H_{g\perp}} \Delta z_t + O(r^3). \tag{45}$$

$$= \tfrac{1}{2} \Delta z_t^\top H_{g\perp} \Delta z_t + O(r^3). \tag{46}$$

By assuming local Hessian Lipschitz continuity, the higher-order terms can be controlled as $O(\|\Delta z_t\|^3)$, that is bounded by Lipschitz bound $\frac{L}{6}\|\Delta z_t\|^3$ on the cubic term. In addition, since $H_{g\perp} = P_{\perp g}^\top H P_{\perp g}$ is symmetric, the second-order term can be bounded via the Rayleigh quotient:

$$\tfrac{1}{2}\lambda_{min,\perp}\|\Delta z_t\|^2 \leq \tfrac{1}{2} \Delta z_t^\top H_{g\perp} \Delta z_t \leq \tfrac{1}{2}\lambda_{max,\perp}\|\Delta z_t\|^2, \tag{47}$$

where $\lambda_{min,\perp}$ and $\lambda_{max,\perp}$ denote the minimum and maximum eigenvalues of $H_{g\perp}$, respectively.

Then, $\Delta r(z_t)$ is bounded as follows:

$$\Delta r(z_t) \leq \tfrac{1}{2}\lambda_{max,\perp}\|\Delta z_t\|^2 + O(\|\Delta z_t\|^3). \tag{48}$$

Because the random variable is contained in $\Delta z_t$, we derive the bound in expectation as:

$$\mathbb{E}[\Delta r(z_t)] \leq \tfrac{1}{2}\lambda_{max,\perp}\mathbb{E}[\|\Delta z_t\|^2] + O(\mathbb{E}[\|\Delta z_t\|^3]). \tag{49}$$

In our formulation, the random vector $U \in \mathbb{R}^d$ corresponds to a single step of Langevin dynamics, which yields a Gaussian distribution in the high-dimensional setting. By Lemma 1, the cubic term can therefore be controlled in terms of the second moment. Hence,

$$\mathbb{E}[\Delta r(z_t)] \leq \tfrac{1}{2}\lambda_{max,\perp}\mathbb{E}[\|\Delta z_t\|^2] + O\big(\mathbb{E}[\|\Delta z_t\|^2]^{3/2}\big). \tag{50}$$

Suppose that the second moment is controlled as

$$\mathbb{E}[\|\Delta z_t\|^2] \ \leq \ k \ < \ 1. \tag{51}$$

Then, by Eq. (50), the residual perturbation is bounded by

$$\mathbb{E}[\Delta r(z_t)] \leq \tfrac{1}{2}\lambda_{max,\perp}\mathbb{E}[\|\Delta z_t\|^2] + O\big(\mathbb{E}[\|\Delta z_t\|^2]^{3/2}\big) \leq Ck + O(k^{3/2}), \tag{52}$$

where the constant $C = \tfrac{1}{2}\lambda_{max,\perp}$ depends only on the local smoothness of $r$. Since $k \ < \ 1$, the cubic term bound is of order $O(k)$. Moreover, because $k = \mathbb{E}[\|\Delta z_t\|^2]$ is determined by $\eta_t$, we can choose $\eta_t$ that preserves measurement consistency at the current timestep while reducing the KL divergence.

*Step 2. The step size second moment.* We expand the second moment as

$$\mathbb{E}[\|\Delta z_t\|^2] = \mathbb{E}[\|\Delta z_t\|^2] \tag{53}$$

$$= \mathbb{E}[\|\eta_t s_t + \sqrt{2\eta_t}\,\epsilon\|^2] \tag{54}$$

$$= \mathbb{E}[\eta_t^2\|s_t\|^2 + 2\eta_t\|\epsilon\|^2] + \underbrace{\mathbb{E}[2\eta_t\sqrt{2\eta_t}s_t^\top\epsilon]}_{=0} \tag{55}$$

$$= \mathbb{E}[\eta_t^2\|s_t\|^2 + 2\eta_t\|\epsilon\|^2], \tag{56}$$

where $\mathbb{E}[e] = 0$.

Let the adaptive step size be parameterized as

$$\eta_t = \frac{\|\epsilon\|^2}{\|s_t\|^2}\lambda, \qquad \lambda > 0, \tag{57}$$

where $\lambda$ is a constant hyperparameter balancing the drift and diffusion terms as proposed in (Song et al., 2021). Under this parameterization, the second moment becomes:

$$\mathbb{E}[\|\Delta z_t\|^2] = \lambda^2 \, \mathbb{E}\left[\frac{\|\epsilon\|^4}{\|s_t\|^2}\right] + 2\lambda \, \mathbb{E}\left[\frac{\|\epsilon\|^4}{\|s_t\|^2}\right]. \tag{58}$$

Since $\epsilon \sim \mathcal{N}(0, I)$, we have $\mathbb{E}[\|\epsilon\|^4] = d\,(d+2)$. Moreover, by the concentration of measure in high dimension (Chung et al., 2023), the squared norm $\|\epsilon_\theta\|^2$ concentrates sharply around its mean $d$. For a well-trained diffusion model, the network prediction $\epsilon_\theta$ approximates $\epsilon$ in distribution. Therefore, we can assume $\mathbb{E}[\|\epsilon_\theta\|^2] = d$, with high probability due to the concentration effect.

Since the Stein score is defined as $s_t = -\frac{\epsilon_\theta}{\sigma_t}$, it follows that $\mathbb{E}[\|s_t\|^2] = \frac{d}{\sigma_t^2}$, where $\sigma_t$ is the variance schedule of the diffusion model. Hence,

$$\mathbb{E}\left[\frac{\|\epsilon\|^4}{\|s_t\|^2}\right] \approx \frac{\mathbb{E}[\|\epsilon\|^4]}{\|s_t\|^2} = (d+2)\sigma_t^2, \tag{59}$$

where the approximation holds with high probability by the concentration effect in high dimensions. Therefore, we obtain the compact form of the second moment:

$$\mathbb{E}[\|\Delta z_t\|^2] = (\lambda^2 + 2\lambda)\,(d+2)\,\sigma_t^2. \tag{60}$$

We aim to control the second moment such that

$$\mathbb{E}[\|\Delta z_t\|^2] \;\leq\; k \;<\; 1. \tag{61}$$

From the Eq. (60), this requirements holds if

$$\lambda^2 + 2\lambda \leq \frac{k}{(d+2)\sigma_t^2} \tag{62}$$

$$(\lambda + 1)^2 \leq 1 + \frac{k}{(d+2)\sigma_t^2} \tag{63}$$

$$\therefore \lambda \leq \sqrt{1 + \frac{k}{(d+2)\sigma_t^2}} - 1 \tag{64}$$

Since for any $v \geq 0$, it holds that $\sqrt{1+v} - 1 \leq \sqrt{v}$, a sufficient condition is

$$\lambda \leq \frac{1}{\sigma_t}\sqrt{\frac{k}{(d+2)}}. \tag{65}$$

Therefore, the sufficient condition $\lambda \leq \sqrt{\frac{k}{(d+2)}}$ guarantees that $\mathbb{E}[\|\Delta z_t\|^2] \;\leq\; k \;<\; 1$. $\qquad\square$

# B  ADDITIONAL RESULTS

In this section, we present additional results that could not be included in the main paper due to space limitations. In Sec. B.2, we show that our method improves the stability of solver dynamics through the success rate. In Sec. B.3 and Sec. B.4, we provide additional qualitative comparisons with competing methods. In Sec. B.5, we demonstrate that MCLC can also be applied to diffusion inverse solvers based on pixel diffusion models (PDMs). Finally, in Sec. B.10 and Sec. B.11, we present supplementary analyses that support our discussion in Sec. 5. In Sec. B.13, we explore further applications of our proposed corrector.

## B.1  QUANTITATIVE RESULTS: COMPATIBILITY WITH RECENT INVERSE SOLVERS

This section provides quantitative results to clarify the compatibility of our method with recent inverse problem solvers, including latent diffusion-based method TReg (Kim et al., 2025b) and latent flow matching–based method FlowChef (Patel et al., 2025). It highlights both the generality and future applicability of MCLC.

**Compatibility with TReg.** Table 2 shows the compatibility of our MCLC with the recent latent diffusion inverse solver, TReg (Kim et al., 2025b). For TReg, we strictly follow the experimental settings described in the paper (*e.g.*, task setup). Since several configuration details are not explicitly noted in the paper, we use the default AFHQ settings provided in the authors' official code (`https://github.com/TReg-inverse/TReg`) for the unspecified ones. Our reproduced performance is slightly worse than the reported numbers but remains close, and the relative trend is consistent. Within this reproduced setup, adding our MCLC module provides consistent improvements over the TReg baseline. In this experiment, we evaluate on the AFHQ-val 1K dataset (Choi et al., 2020). For all tasks, the measurement noise level is set to $\sigma_y = 0.01$. Super-resolution is performed with a 12× bicubic downsampling operator, and Gaussian deblurring uses a kernel size of 61 with intensity 5.0. We use the same MCLC hyperparameters (every 3 sampling steps, 3 correction iterations, and $\lambda = 0.2$) across all tasks, and additional algorithmic details and configurations are provided in Sec. C.

| Method | Gaussian Deblur | | | Super Resolution 12x (Avgpool) | | |
|---|---|---|---|---|---|---|
| | PSNR ($\uparrow$) | LPIPS ($\downarrow$) | FID ($\downarrow$) | PSNR ($\uparrow$) | LPIPS ($\downarrow$) | FID ($\downarrow$) |
| FlowChef | 23.76 | 0.364 | 106.76 | **25.26** | 0.480 | 181.15 |
| FlowChef w/ Ours | **28.52** | **0.288** | **77.36** | 24.81 | **0.393** | **125.57** |

Table 6: **Additional quantitative results on FFHQ using FlowChef.** Drop-in improvements on a flow-based solver, FlowChef Patel et al. (2025), are evaluated across super-resolution at 12× scale (average pooling) and Gaussian deblurring with kernel size 61 and intensity 3.0. , and motion deblurring with kernel size 61 and intensity 0.5.

**Compatibility with Flow-based Models.** Table 3 and 6 show further applicability of our MCLC with the recent flow-based generative model. In this experiment, we use Stable Diffusion v3 as the prior model and FlowChef (Patel et al., 2025) as the base solver. Across diverse tasks, MCLC consistently provides noticeable improvements. Although we demonstrate the applicability of MCLC to a flow-based model, we note that flow-based generative models differ from diffusion models in that they parameterize a velocity field rather than a score function. Accordingly, we estimate the score following the approach in (Kim et al., 2025a) to enable our correction scheme. A more tailored variant of MCLC for flow-based methods would be an interesting direction for future improvement. We use the FlowChef implementation from a subsequent work(`https://github.com/FlowDPS-Inverse/FlowDPS`) along with the recommended configurations. However, we find that the step size of 200 used for super-resolution in the paper Kim et al. (2025a) is unsuitable, so we tune it to 20 for all super-resolution tasks.

B.2 QUANTITATIVE RESULTS: EVALUATION OF STABILITY

To further assess stability, we present PSNR histograms similar to DiffStateGrad (Zirvi et al., 2025) in Fig. 10. These experiments demonstrate that our corrector enhances the reliability of base solvers by reducing failure cases and yielding overall performance gains, particularly in terms of mitigating artifacts and alleviating quality degradation. We report PSNR histograms for four linear tasks: Super Resolution, Inpainting (random), Motion Deblur, and Gaussian Deblur. As shown in the figure, MCLC consistently achieves performance improvements across all tasks.

B.3 QUALITATIVE RESULTS: DROP-IN IMPROVEMENT

We provide additional qualitative comparisons against reconstructions obtained with the naive latent diffusion solver (base), as well as the same base solver plugged in with DiffStateGrad or MCLC (ours) for each task. As shown in Fig. 21, 22, 23, 24, 25, and 26, MCLC substantially improves the performance of existing latent diffusion solvers and effectively alleviates visual artifacts.

B.4 QUALITATIVE RESULTS: COMPARISON WITH NON-PLUGGABLE APPROACHES

As noted in the main paper (Sec. 4), we further validate the effectiveness of our method by comparing it with non-pluggable artifact-removal approaches, MPGD (He et al., 2024) and SILO (Raphaeli et al., 2025). As shown in Fig. 11, MCLC achieves notable improvements in both measurement

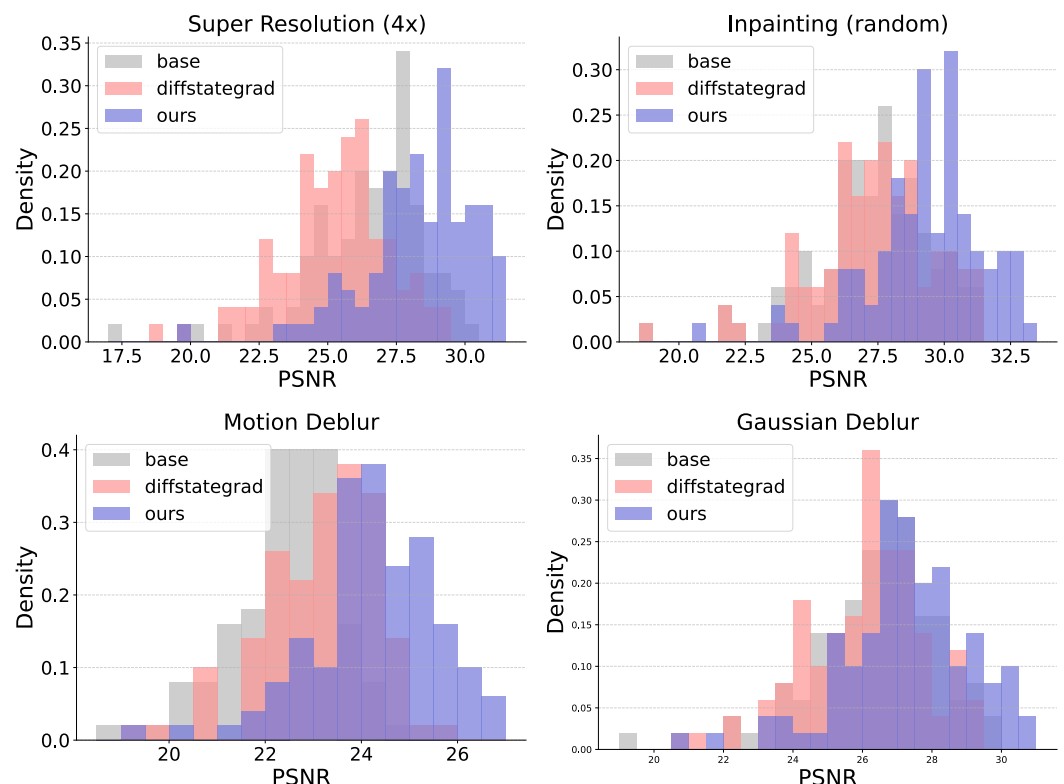

Figure 10: **Histogram visualization of linear tasks, based on PSNR.** Across all linear tasks, our method shows improvements in overall distribution compared to DiffStateGrad (Zirvi et al., 2025).

consistency and perceptual quality, even when compared to non-pluggable approaches designed for artifact removal.

MPGD provides slightly stronger measurement consistency; however, this improvement does not consistently translate into stability or overall quality, and noticeable artifacts remain. SILO achieves high perceptual quality by effectively removing artifacts; however, encoding measurements into the latent space introduces information loss, which leads to low measurement consistency. This limitation may undermine the fundamental goal of inverse problems, reconstructing the original signal consistent with observations. Moreover, since SILO trains its latent degradation operator in a domain-specific manner, it is difficult to generalize in a domain-agnostic setting, which restricts its applicability.

In contrast, MCLC not only adapts readily across domains in a plug-and-play manner without specialized designs but also ensures measurement consistency. This enables existing LDM-based inverse solvers to realize their potential by enhancing stability and quality without sacrificing fidelity or generalizability. Notably, MCLC yields significant gains even when combined with basic solvers such as LDPS and PSLD, highlighting its effectiveness. Considering its easily pluggable nature, MCLC can be combined with various baselines, leaving further room for performance improvement.

### B.5 QUANTITATIVE RESULTS: PIXEL DIFFUSION MODEL (PDM)

Since MCLC can be adaptable not only to Latent Diffusion Models(LDMs) but also to Pixel Diffusion Models(PDMs), we report the experimental results on DPS with and without MCLC. As shown in Table 7 and Fig. 12, our method achieves performance gain, with particularly notable improvements on the motion deblurring task. In this work, we focus on Latent Diffusion Models (LDMs) that provide generic priors. Unlike domain-specialized priors, which tend to produce fewer artifacts, LDMs such as Stable Diffusion suffer more severely from the gap to true reverse diffusion dynamics,

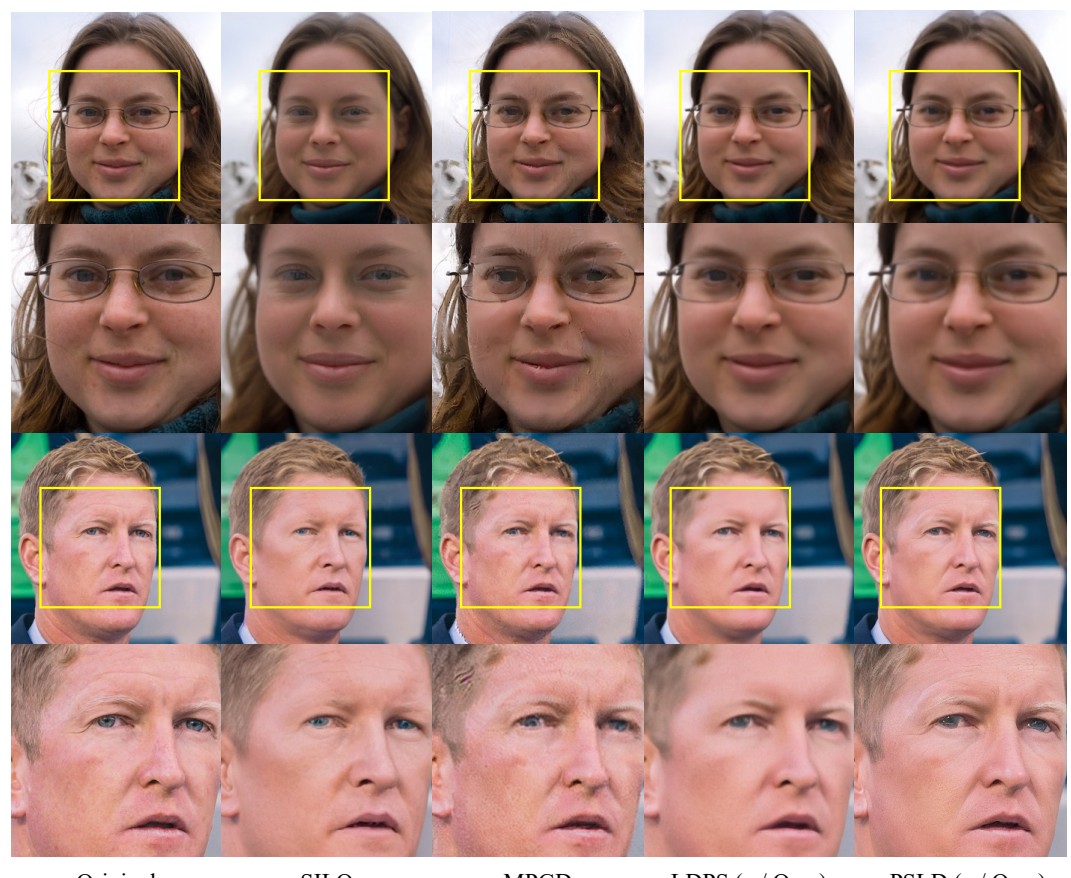

| Original | SILO | MPGD | LDPS (w/ Ours) | PSLD (w/ Ours) |

Figure 11: **Qualitative comparison with non-pluggable approaches.** MCLC mitigates artifacts and improves quality in a plug-and-play manner, surpassing non-pluggable approaches such as MPGD (He et al., 2024) and SILO (Raphaeli et al., 2025). SILO fails to reconstruct the glasses in the first row and often alters identity. MPGD exhibits noticeable overall artifacts.

often resulting in artifacts and degraded quality. This explains why the performance gain appears more substantial in the LDM setting.

| | Super Resolution (4x) | | | Motion Deblur | | | Gaussian Deblur | | | Nonlinear Deblur | | |
|---|---|---|---|---|---|---|---|---|---|---|---|---|
| | PSNR (↑) | LPIPS (↓) | FID (↓) | PSNR (↑) | LPIPS (↓) | FID (↓) | PSNR (↑) | LPIPS (↓) | FID (↓) | PSNR (↑) | LPIPS (↓) | FID (↓) |
| DPS | 25.62 | 0.263 | 78.60 | 24.31 | 0.282 | 81.14 | 25.06 | 0.257 | 75.73 | 22.97 | 0.369 | 108.97 |
| DPS w/ Ours | **25.63** | **0.262** | **77.92** | **26.11** | **0.248** | **70.60** | **25.08** | **0.256** | **74.77** | **23.11** | **0.362** | **104.83** |

Table 7: Comparison of DPS (Chung et al., 2023) with and without MCLC on FFHQ.

### B.6 QUANTITATIVE RESULTS: SINGLE DEFAULT HYPERPARAMETER SET

Table 8 demonstrates that MCLC achieves meaningful performance improvements even when using a single default hyperparameter set across all tasks. For each base solver, we apply single unified configuration: $k = 5$, $N_c = 1$, and $\lambda = 0.1$ for PSLD and LDPS, and for Resample we use the inpainting-style setting, *i.e.*, $k = 5$, $N_c = 3$, $\lambda = 0.15$, $N_c^{\text{DPS}} = 1$, and $\lambda^{\text{DPS}} = 0.05$. This unified choice is supported by our theoretical insight in Eq. (65), which indicates that an appropriate step size and a sufficient number of correction iterations allow MCLC to operate effectively without task-specific tuning. Nevertheless, additional tuning can further balance efficiency and performance under severe degradations, and we provide such tuned hyperparameter sets in Sec. C.

Table 8: **Quantitative results using a single default hyperparameter set.** Even with a unified hyperparameter configuration across tasks, MCLC provides consistent and meaningful performance improvements.

| Task | Base | Method | FFHQ | | | |
|------|------|--------|------|------|------|------|
| | | | PSNR (↑) | LPIPS (↓) | FID (↓) | P-FID (↓) |
| Gaussian Deblur | LDPS | Base | 27.61 | 0.349 | 100.10 | 93.55 |
| | | Ours (single default) | **27.99** | **0.334** | **85.84** | **71.62** |
| | | Ours (tuned) | 28.14 | 0.303 | 80.83 | 54.74 |
| | PSLD | Base | 27.84 | **0.314** | 89.18 | 90.54 |
| | | Ours (single default) | **27.94** | 0.317 | **79.81** | **69.39** |
| | | Ours (tuned) | 27.97 | 0.286 | 66.28 | 59.13 |
| | ReSample | Base | 26.44 | 0.368 | **75.17** | 148.11 |
| | | Ours (single default) | **27.33** | **0.355** | 80.90 | **96.17** |
| | | Ours (tuned) | 27.25 | 0.353 | 78.38 | 106.16 |
| Motion Deblur | LDPS | Base | 26.54 | 0.390 | 118.77 | 112.74 |
| | | Ours (single default) | **27.03** | **0.363** | **99.71** | **81.82** |
| | | Ours (tuned) | 27.45 | 0.318 | 82.94 | 55.55 |
| | PSLD | Base | 26.87 | **0.343** | 106.34 | 102.60 |
| | | Ours (single default) | **26.92** | 0.348 | **90.92** | **72.30** |
| | | Ours (tuned) | 26.86 | 0.308 | 74.64 | 60.05 |
| | ReSample | Base | 22.45 | 0.635 | 108.14 | 174.52 |
| | | Ours (single default) | **24.19** | **0.599** | **103.70** | **114.85** |
| | | Ours (tuned) | 24.24 | 0.588 | 102.02 | 118.87 |
| Super Resolution (4×) | LDPS | Base | **28.47** | **0.301** | 78.08 | 69.66 |
| | | Ours (single default) | 28.19 | 0.307 | **74.33** | **57.48** |
| | | Ours (tuned) | 28.34 | 0.283 | 74.78 | 58.55 |
| | PSLD | Base | **27.69** | 0.265 | 63.95 | 63.47 |
| | | Ours (single default) | 27.44 | **0.261** | **61.09** | **52.43** |
| | | Ours (tuned) | 27.33 | 0.267 | 62.12 | 58.13 |
| | ReSample | Base | 26.40 | 0.347 | 70.16 | 133.15 |
| | | Ours (single default) | **27.73** | **0.264** | **55.38** | **68.55** |
| | | Ours (tuned) | 28.32 | 0.236 | 53.85 | 78.08 |
| Inpainting (Random) | LDPS | Base | 31.22 | 0.171 | 48.88 | 83.30 |
| | | Ours (single default) | **31.31** | **0.167** | **47.76** | **79.23** |
| | | Ours (tuned) | 31.28 | 0.169 | 48.05 | 81.68 |
| | PSLD | Base | 30.14 | 0.222 | 58.84 | **79.59** |
| | | Ours (single default) | **31.30** | **0.167** | **48.04** | 79.74 |
| | | Ours (tuned) | 30.73 | 0.185 | 49.80 | 72.69 |
| | ReSample | Base | 27.27 | 0.374 | 103.17 | 133.80 |
| | | Ours (single default) | **28.75** | **0.296** | **85.90** | **103.25** |
| | | Ours (tuned) | 29.35 | 0.235 | 75.65 | 108.27 |

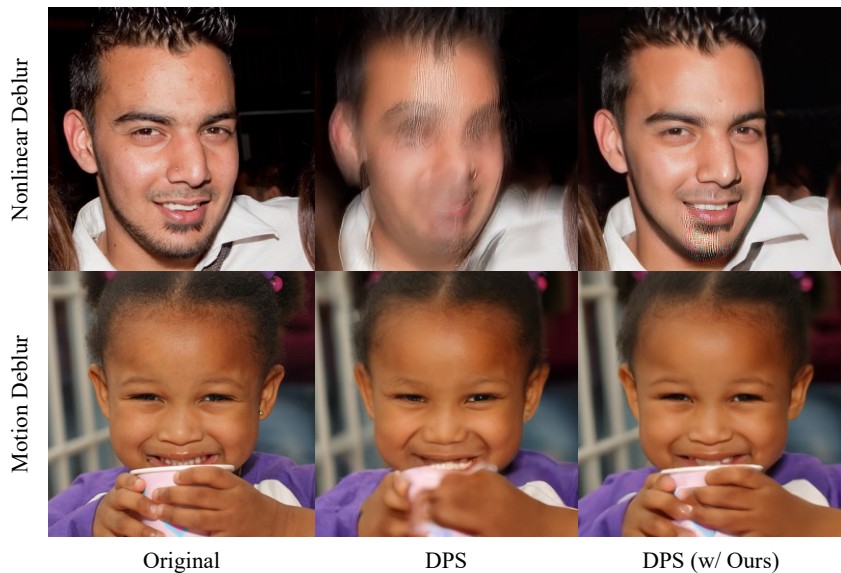

Figure 12: Qualitative result on pixel diffusion inverse solver (DPS (Chung et al., 2023)), with and without our method MCLC.

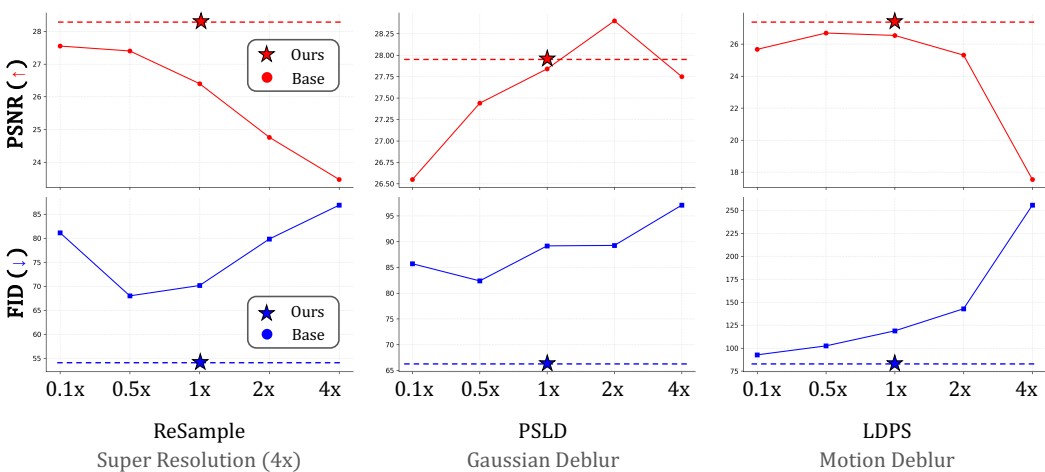

Figure 13: **PSNR and FID analyses across swept step-sizes.** The red and blue stars indicate the performance of MCLC applied to the default $1\times$ setting. MCLC pushes both metrics beyond the practical ceiling inferred by the sweep, demonstrating improvements that cannot be achieved through step-size tuning alone.

### B.7 QUANTITATIVE AND QUALITATIVE RESULTS: SOLVER PARAMETER SEARCH

To verify that the observed artifacts do not merely stem from under-tuned baselines, we conduct a parameter sweep over the most influential hyperparameter, the gradient step size. Starting from the configuration used in the main paper (noted as $1\times$), we evaluated $0.1\times, 0.5\times, 1\times, 2\times, 4\times$ for each solver and some tasks. As shown in Fig. 14, most settings still produce noticeable artifacts. Extremely small steps (e.g., $0.1\times$) suppress some artifacts but fail to provide sufficient data-fidelity updates, leading to degraded reconstructions.

Figure 13 shows the PSNR and FID curves across the swept step-size parameters. While the $1\times$ setting used in the main paper is not the exact optimum, it is generally close and remains a reasonably well-tuned choice across tasks. In this table, PSNR (red, higher is better) indicates reconstruction fidelity, and FID (blue, lower is better) indicates stability without artifacts and degraded results. The

sweep table demonstrates that some configurations come close to optimal, but the curves reveal a practical ceiling, meaning that it is difficult to find a single configuration that fully satisfies both reconstruction fidelity (PSNR) and perceptual quality (FID). Notably, applying MCLC to the default $1\times$ setting pushes both PSNR and FID beyond this apparent ceiling. This demonstrates that MCLC provides substantial gains beyond what can be achieved through hyperparameter tuning.

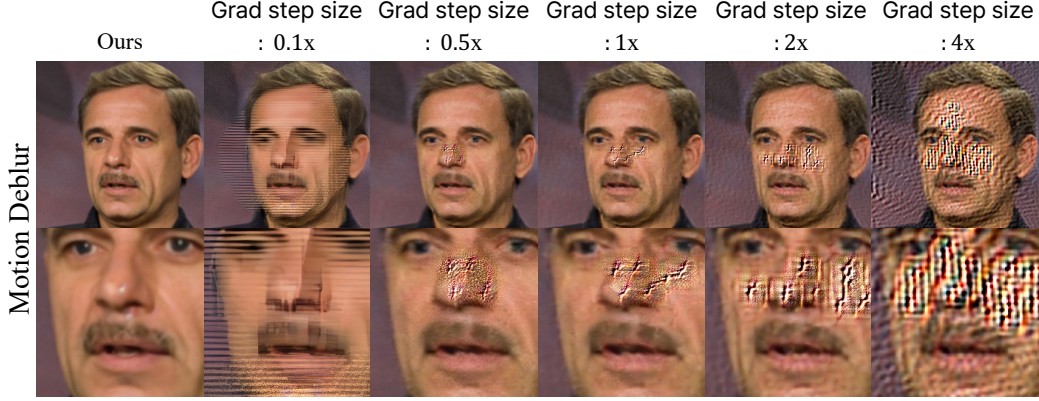

(a) LDPS

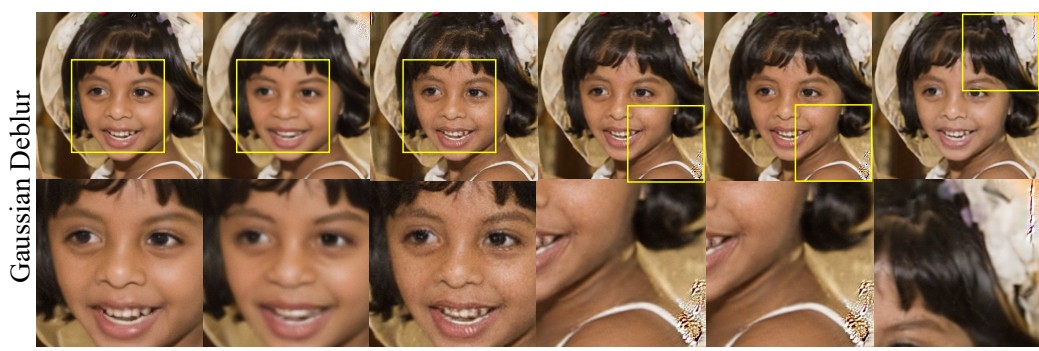

(b) PSLD

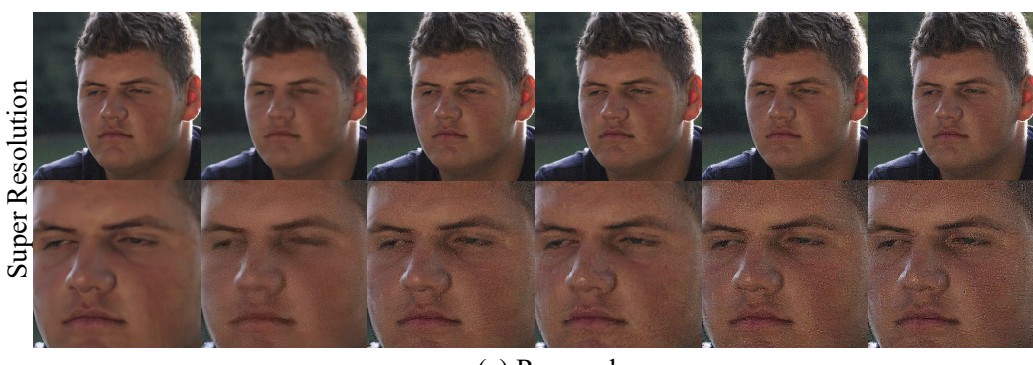

(c) Resample

Figure 14: **Qualitative results across swept gradient step sizes and with MCLC.** Across the hyperparameter sweep, artifacts or degraded reconstructions consistently appear in most settings. LDPS exhibits clear regional distortions, while Resample produces dotted or jittering noise patterns across different settings.

## B.8 QUALITATIVE RESULTS: BLOB ARTIFCATS MITIGATION

As noted earlier, blob artifacts arise when latent values become excessively amplified, that is, when they are pushed far outside the feasible latent range. While MCLC suppresses this artifact by pulling the latent values back toward stable regions, this corrective influence can be weaker than the amplification, which explains why certain artifacts may not be fully removed. Nevertheless, MCLC sufficiently suppresses the blob artifacts. To demonstrate how effectively MCLC mitigates this phenomenon, we provide additional qualitative results in Fig. 15. These examples show that MCLC significantly reduces the magnitude of out-of-range latent values and noticeably suppresses the resulting blob artifacts in the decoded images.

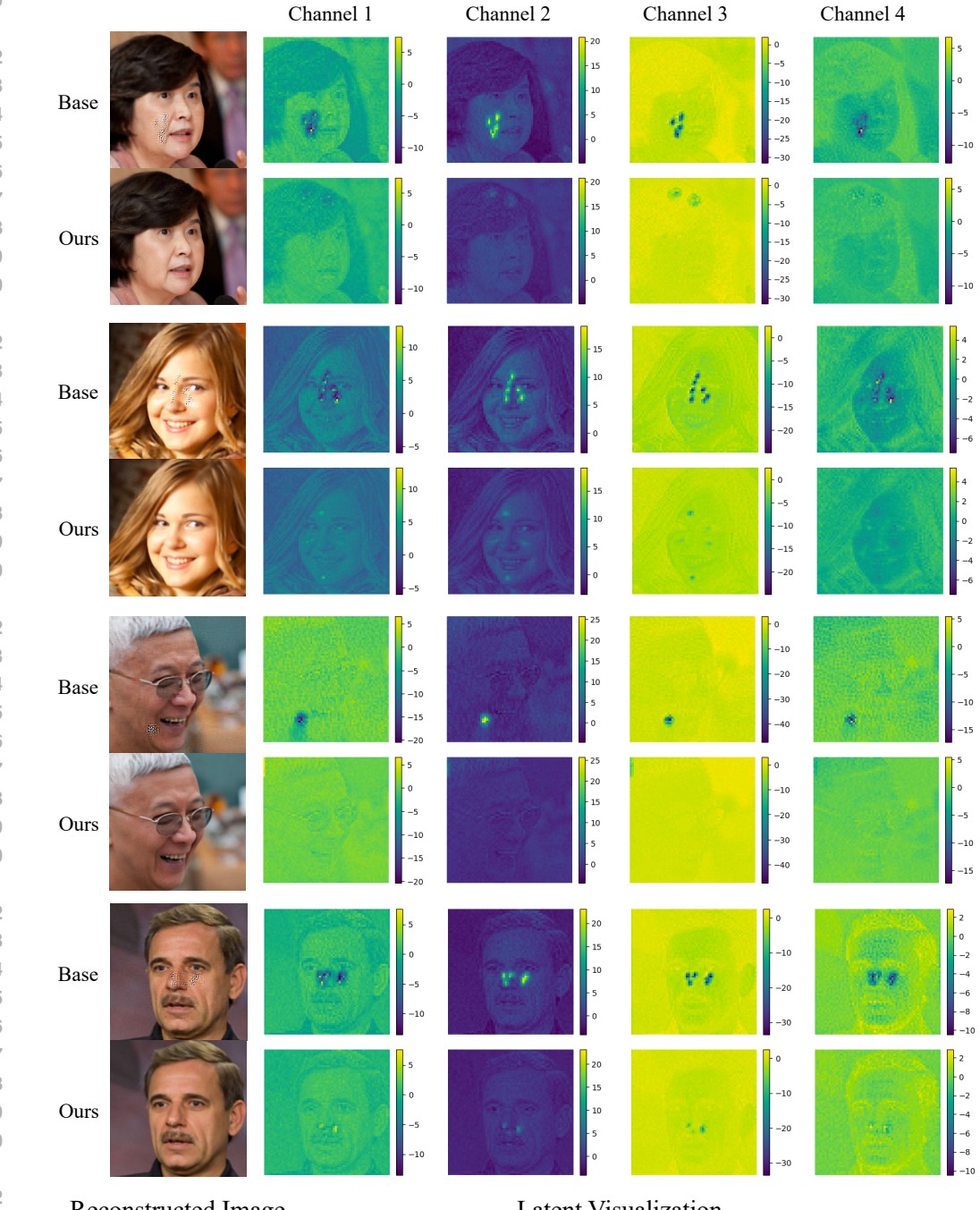

Figure 15: **Visualization of MCLC's suppression of blob artifacts.** The base solver is LDPS and the task is Gaussian deblurring.

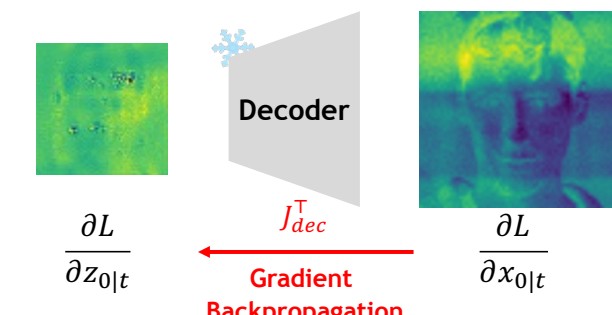

Figure 16: When the measurement-consistent loss $L$ is backpropagated through the decoder, redundant signals emerge in the gradient directions, leading to problematic latent updates.

### B.9 ANALYSIS: COMPATIBILITY WITH LATENTDAPS

As discussed in the main paper (Sec. 4), we provide a more detailed explanation of why Latent-DAPS (Zhang et al., 2025a) is less compatible with our proposed MCLC. Briefly, this could be attributed to its solving procedure which does not inherit reverse diffusion dynamics; instead, each iteration is initialized from a newly predicted $z_0$ with annealed noise, decoupling consecutive updates. As a result, the stabilizing effect of our corrector appears limited in this case.

To clarify, we first review how LatentDAPS operates. LatentDAPS decouples consecutive steps in the diffusion sampling trajectory. Rather than performing standard reverse sampling, it directly predicts $z_{0|t}$ with ODE solver and updates it using the log-posterior gradient $\nabla_{z_0} \log p(z_{0|t}|y)$, after which annealed noise is added to proceed to the next step. In this way, the dependency between successive steps is broken, allowing the method to explore a larger solution space.

However, our proposed MCLC is designed to reduce the gap between the true reverse diffusion dynamics and the dynamics of the latent diffusion inverse solver, thereby making latent diffusion solvers more stable. Yet, because LatentDAPS does not follow reverse dynamics by decoupling consecutive steps and repeatedly reinitializing with annealed noise, the stabilizing effect of MCLC accumulates less effectively in this setting. For this reason, Table 1 shows that MCLC is less effective with LatentDAPS compared to other solvers, where it delivers substantial performance gains. Although the powerful recent solver LatentDAPS is less compatible with MCLC, the value of our approach remains clear: it closes the gap toward the true reverse process of diffusion model without relying on the linear manifold assumption, and most solvers are still built on reverse diffusion sampling combined with a measurement-consistency step.

### B.10 ANALYSIS: PROBLEMATIC GRADIENT

As noted in Sec. 5, we confirm that the decoder itself produces problematic gradients. To investigate this, we decompose the gradient $\frac{\partial L}{\partial z_{0|t}}$ into two components $J_{\mathcal{D}}^{\top} \frac{\partial L}{\partial x_{0|t}}$ and compare their characteristics to examine the effect of the decoder Jacobian. Specifically, we analyze the gradients $\frac{\partial L}{\partial x_{0|t}}$ in pixel space and $\frac{\partial L}{\partial z_{0|t}}$ in latent space. As shown in Fig. 16, the decoder Jacobian introduces signals in directions redundant to the measurement-consistent pixel-level gradients, thereby distorting the latent gradients.

### B.11 ANALYSIS: SCALED-OUTLIERS IN LATENT SPACE

In Sec. 5, we analyze the scaled-outlier in the latent space of VAE. We present a visualization of the encoded latents from multiple RGB images in Fig. 17. The result indicates that the encoded latents from Stable Diffusion v1.5 already contain scaled-outlier regions across channels and images, whereas those from Stable Diffusion XL do not. This finding supports that blob artifacts arise not only from the VAE decoder, but also from the pre-trained VAE itself. Therefore, the blob artifacts could be mitigated by replacing the base diffusion model with an enhanced diffusion model that can reduce scaled-outliers.

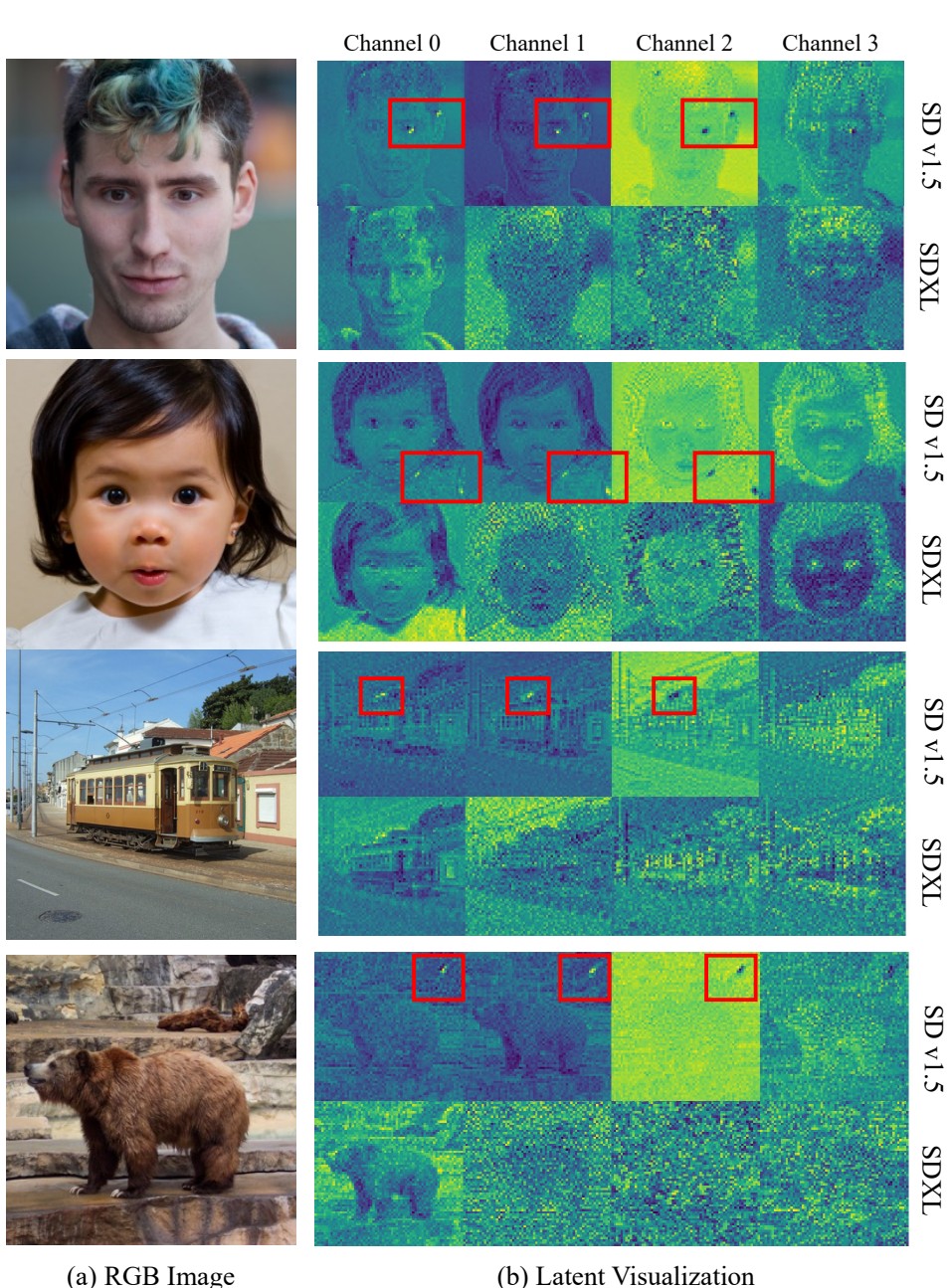

(a) RGB Image      (b) Latent Visualization

Figure 17: Visualization of latent spaces in SD v1.5 and SDXL. SD v1.5, the commonly used latent diffusion model for inverse problems, exhibits scaled outliers in its latent space. The scaled-outliers are amplified through the decoder, which may result in undesirable artifacts. Unlike SD v1.5, the latent space of SDXL shows no such outliers, displaying only a slightly noisy appearance.

## B.12 ANALYSIS: DO THE ARTIFACTS STEM FROM THE PRIOR?

To clarify whether the observed artifacts originate solely from the pretrained LDM prior itself, we provide two supporting analyses. First, we show that these artifacts persist when replacing Stable Diffusion v1.5 with other pretrained LDMs (e.g., Stable Diffusion v2.1, RealisticVision v5.1): despite using different priors, the same LDM-based inverse solver (PSLD) continues to exhibit the artifacts, suggesting that the issue is tied to the inverse-solving dynamics rather than to a particular prior (see Fig. 18). Moreover, applying our MCLC, which increases prior fidelity, suppresses these artifacts rather than producing new ones, indicating that the prior itself cannot be the sole cause (see Table 9).

| | Gaussian Deblur | | | | Super Resolution (4x) | | | |
| --- | --- | --- | --- | --- | --- | --- | --- | --- |
| | PSNR (↑) | LPIPS (↓) | FID (↓) | P-FID (↓) | PSNR (↑) | LPIPS (↓) | FID (↓) | P-FID (↓) |
| RV v5.1 (base) | 26.69 | 0.380 | 110.38 | 78.25 | **27.96** | 0.305 | 71.55 | 54.91 |
| RV v5.1 (ours) | **26.75** | **0.334** | **95.46** | **56.66** | 27.82 | **0.295** | **68.2** | **50.10** |
| SD v2.1 (base) | 26.22 | 0.403 | 116.49 | 120.94 | **28.70** | 0.244 | 60.34 | 49.99 |
| SD v2.1 (ours) | **26.69** | **0.335** | **91.13** | **52.76** | 28.64 | 0.246 | **60.33** | **48.30** |

Table 9: **Quantitative results on other LDMs.** These results show that the instability is also observed when using different pretrained LDM priors, and that it can be mitigated by applying MCLC.

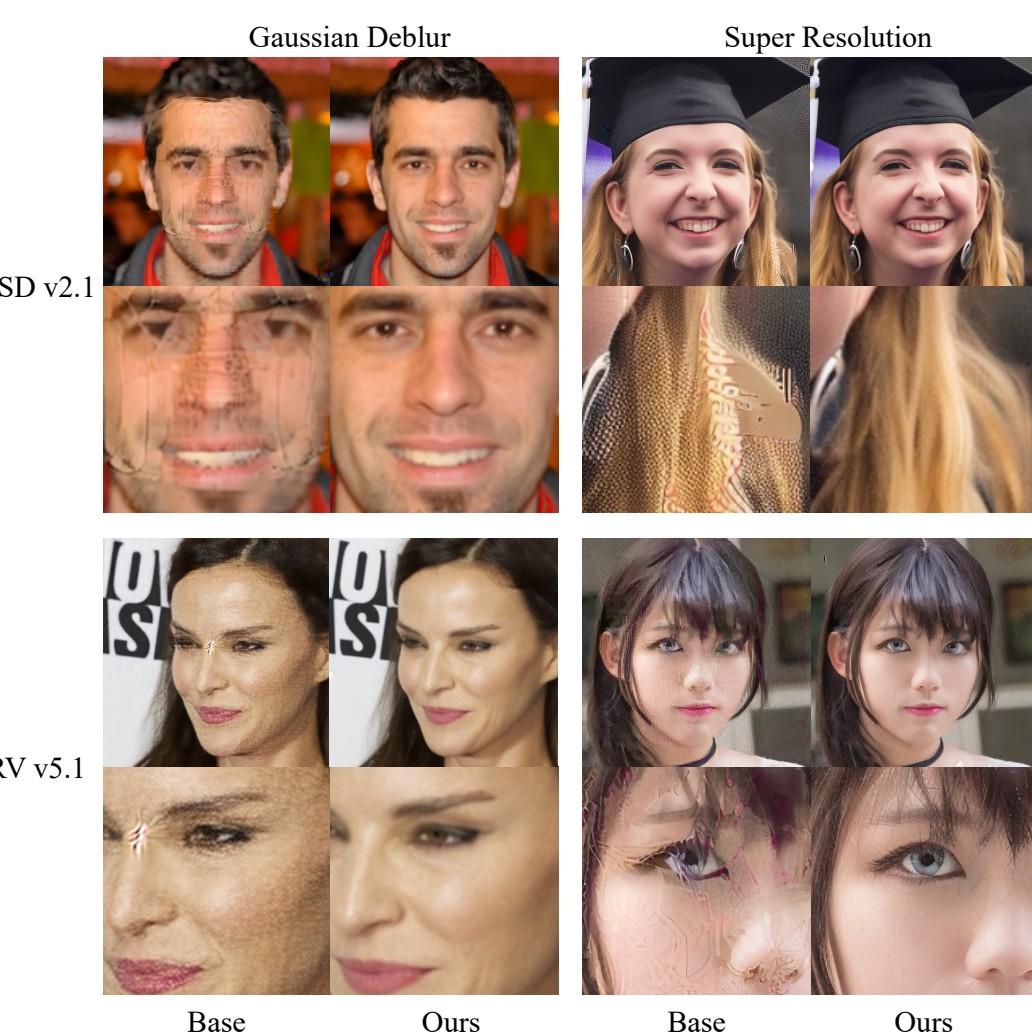

Figure 18: **Artifacts across different LDM priors.** Artifacts persist even when changing the prior, and MCLC effectively mitigates them.

Second, we note that pure generation from the LDM does not produce such artifacts. To provide an explainable demonstration, we compare two reconstructions of the same sample: (a) applying DDIM inversion followed by the standard generative reverse process, and (b) solving the inverse problem using its measurement obtained through the measurement operator. Both procedures use the same pretrained prior and start from the same sample, yet artifacts appear only in the inverse-solved output (see Fig. 19). This demonstrates that the artifacts are far more likely to arise from the inverse-problem dynamics rather than from the pretrained prior itself.

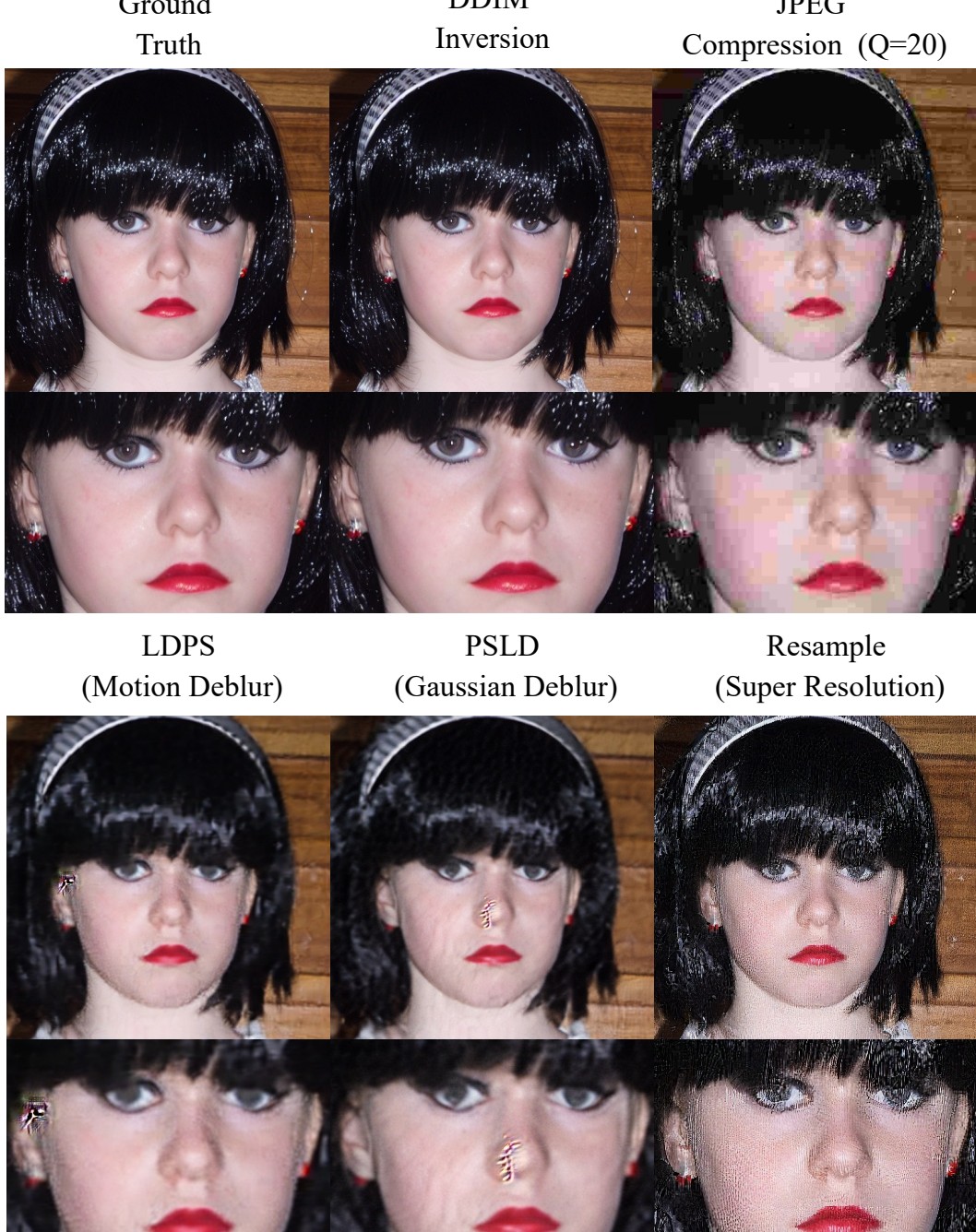

Figure 19: **Reconstruction comparison.** DDIM-inversion reconstruction shows no artifacts, whereas inverse-solver reconstructions exhibit clear artifacts, indicating that the generative prior alone does not introduce them.

## B.13 APPLICATION BEYOND INVERSE PROBLEM

**Prompt:** Walker hound, Walker foxhound on snow

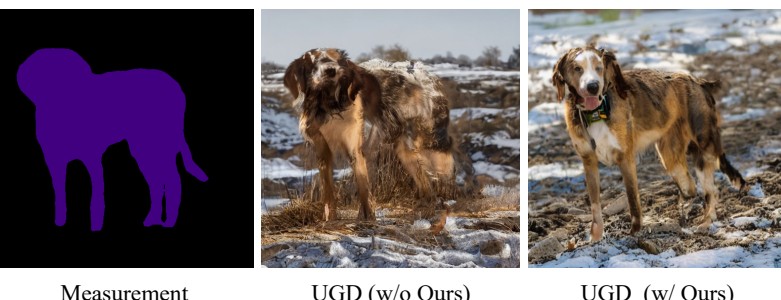

|     |     |     |
| :-: | :-: | :-: |
| Measurement | UGD (w/o Ours) | UGD (w/ Ours) |

Figure 20: Text-conditioned image generation results guided by segmentation masks using UGD (Bansal et al., 2024).

To further investigate the applicability of MCLC, we provide guided-sampling results with and without our corrector. Universal Guided Diffusion (UGD) (Bansal et al., 2024) proposes a universal guided diffusion sampling framework applicable to various guidance signals, where the guidance can be any off-the-shelf model or guiding function. We present text-conditioned image generation results guided by a segmentation mask. Figure 20 shows the extensibility of our proposed correcting mechanism to guided sampling. In this experiment, we set the inner iteration of UGD to 3.

## B.14 GUIDE FOR MCLC HYPERPARAMETER CHOICE

The step size and the number of MCLC iterations can be selected in a simple and interpretable manner. In Sec. B.6, we provide a single default configuration that already works well across a broad range of degradations. For further tuning, if the degradation becomes more severe, one may increase the corrector step size or apply the correction more frequently. According to Eq. (65), this setting may introduce a slightly larger deviation from exact measurement consistency; however, under stronger degradations, prioritizing the stabilization of the reverse dynamics becomes more beneficial.

## C   IMPLEMENTATION DETAILS

In this section, we provide details of our experimental implementation. In Sec. C.1, we describe how we measure the KL divergence with respect to the true reverse diffusion dynamics, corresponding to the experiments shown in Fig. 3. In Sec. C.2, we present an overview of our proposed method, explaining how it can be plugged into existing solvers along with the detailed algorithm. In the following sections (Sec. C.3, C.4, C.5, and C.6), we sequentially report the hyperparameter settings of the baseline latent diffusion solvers, the integration of our method into these solvers, and the configuration of our correctors, including their hyperparameter choices.

### C.1   EXPERIMENTAL DETAILS: MEASURING KL DIVERGENCE

Figure 3 reports the KL divergence at each timestep (sampled every 15 steps) between the true reverse diffusion dynamics and those produced by the LDPS solver on the Gaussian deblurring task with the FFHQ dataset. Results are shown for both the baseline solver and the solver corrected with our MCLC. To compute each KL divergence, we first collect the intermediate states of each dynamics across the dataset: $z_t^\# \sim q_t^\#$, denoting the time-evolving distribution of the latent diffusion inverse solver, and $z_t^c \sim q_t^c$, denoting that of the corrected solver. To approximate the true reverse diffusion dynamics, we employ DDIM inversion to obtain the corresponding intermediate states $z_t \sim p_t$ at each timestep. We then fit Gaussian Mixture Models (GMMs) to these sets of samples and compute the KL divergence between the resulting GMM distributions, i.e., $\mathcal{D}_{KL}(q_t^\# | p_t)$ (red line) and $\mathcal{D}_{KL}(q_t^c | p_t)$ (purple line). We fit Gaussian Mixture Models (GMMs) with 32 Gaussian components to approximate each distribution, and then compute the KL divergence between them.

### C.2   ALGORITHMS

In this section, we present the algorithm of our proposed MCLC and provide an overview of how it can be plugged into existing latent diffusion inverse solvers. More generally, latent diffusion inverse solvers follow a framework where the reverse sampling process is interleaved with a measurement-consistency step. Our MCLC step is inserted immediately after this measurement-consistency step. An overview of the pluggable algorithm is given in Algorithm 1, and the detailed procedure of MCLC is described in Algorithm 2. For efficiency, MCLC is executed every $k$ steps (e.g., $k = 3$) instead of being applied at each step.

---

**Algorithm 1** Latent diffusion inverse solver with MCLC correction

---

**Require:** Pretrained LDM $s_\theta$, VAE decoder $\mathcal{D}_\phi$, measurement $\boldsymbol{y}$ variance schedule $\{\beta(t)\}_{t=1}^T$, corrector step size hyperparameter $\lambda$

    **Init:** $\boldsymbol{z}_T \sim \mathcal{N}(\boldsymbol{0}, \mathbf{I})$
    **for** $t = T, \ldots, 1$ **do**
        $\hat{\boldsymbol{z}}_{0|t} \leftarrow \text{ApproxPosterior}\left(\boldsymbol{z}_t, \boldsymbol{s}_\theta(\boldsymbol{z}_t, t)\right)$         ▷ e.g., Tweedie's formula
        **// Measurement Consistency Step**
        $\hat{\boldsymbol{x}}_{0|t} \leftarrow \mathcal{D}_\phi(\hat{\boldsymbol{z}}_{0|t})$         ▷ Decode latent
        $(\boldsymbol{z}_{t|\boldsymbol{y}}^\#, \boldsymbol{g}_t) \leftarrow \text{LatentUpdate}\left(\hat{\boldsymbol{x}}_{0|t}, \boldsymbol{y}, \mathcal{A}\right)$     ▷ Return measurement-consistent gradient $\boldsymbol{g}_t$
        $\boldsymbol{z}_{t|\boldsymbol{y}}^0 \leftarrow \boldsymbol{z}_{t|\boldsymbol{y}}^\#$
        **// Correction Step**
        **for** $c = 1, \ldots, N_c$ **do**         ▷ Langevin update within the orthogonal complement of $\boldsymbol{g}_t$
            $\text{MCLC}(\mathbf{z}_{t|y}, \mathbf{s}_\theta, t, \boldsymbol{g}_t)$
        $\boldsymbol{z}_{t-1} \leftarrow \text{ReverseSampling}(\boldsymbol{z}_{t|\boldsymbol{y}}, \boldsymbol{s}_\theta, \beta(t))$
    **end for**
    **return** $\boldsymbol{x}_0 = \mathcal{D}_\phi(\boldsymbol{z}_0)$         ▷ final reconstruction

---

---

**Algorithm 2** Measurement-Consistent Langevin Corrector (MCLC)

---

**Parameters:** Corrector step size $\{\eta_t\}_{t=1}^T$
**Inputs:** Pre-corrected latent $z_t^\#$, score network $s_\theta$, time $t$, measurement-consistent gradient $g_t$
**Outputs:** Corrected latent $z_t^c$
**Function** MCLC($z_t^\#, s_\theta, t, g_t$) :
  $g \leftarrow g_t / \|g_t\|$          ▷ normalize measurement gradient $g_t$
  $z_t^c \leftarrow z_t^\# + \eta_t \cdot \Pi_{\perp g}\big(s_\theta(z_t^\#, t)\big) + \sqrt{2\eta_t} \cdot \Pi_{\perp g}(\epsilon)$    ▷ $\epsilon \sim \mathcal{N}(\mathbf{0}, \mathbf{I})$
  **return** $z_t^c$

---

### C.3 ALGORITHMIC DETAILS: LDPS (LATENT DPS)

We plug MCLC into Latent DPS (LDPS) and present the resulting algorithm in Algorithm 3. We used the LDPS is an extension of DPS (Chung et al., 2023) to latent diffusion models. LDPS applies a measurement-consistency step at every sampling iteration. For LDPS, we used the original PSLD implementation, with the only modifications being the removal of the PSLD regularization term and the addition of MCLC. For our experiments, we use the DDIM sampling procedure with 1000 timesteps, and apply the MCLC step at every $k$-th step during sampling. At each corrector step, we perform $N_c$ corrector iterations with step size hyperparameter $\lambda$, as defined in Eq. (57). The detailed experimental settings, corresponding to those reported in Table 1, are summarized in Table 10 across each task.

---

**Algorithm 3** MCLC-LDPS

---

**Require:** $T, y, \zeta, \{\alpha_t\}_{t=1}^T, \{\bar{\alpha}_t\}_{t=1}^T, \{\tilde{\sigma}_t\}_{t=1}^T$
**Require:** $\mathcal{E}, \mathcal{D}, \mathcal{A}, s_\theta, N_c, \lambda,$
 $z_T \sim \mathcal{N}(\mathbf{0}, \mathbf{I})$
 **for** $t = T$ **to** $1$ **do**
  $\hat{s} \leftarrow s_\theta(z_t, t)$
  $\hat{z}_0 \leftarrow \frac{1}{\sqrt{\bar{\alpha}_t}}(z_t + (1 - \bar{\alpha}_t)\hat{s})$
  $\epsilon \sim \mathcal{N}(\mathbf{0}, \mathbf{I})$
  $z_{t-1} \leftarrow \frac{\sqrt{\alpha_t}(1 - \bar{\alpha}_{t-1})}{1 - \bar{\alpha}_t} z_t + \frac{\sqrt{\bar{\alpha}_{t-1}}(1 - \alpha_t)}{1 - \bar{\alpha}_t} \hat{z}_0 + \tilde{\sigma}_t \epsilon$
  $g_t \leftarrow \zeta \nabla_{z_t} \|y - \mathcal{A}(\mathcal{D}(\hat{z}_0))\|_2^2$
  $z_{t-1}' \leftarrow z_{t-1} - g_t$
  // MCLC Correction Step
  **if** $(t \mod k) = 0$ **then**
   **for** $c = 1, \ldots, N_c$ **do**
    $z_{t-1}' \leftarrow$ MCLC($z_{t-1}', s_\theta, t-1, g_t$)   ▷ Corrector step size hyperparameter $\lambda$
 **end for**
 **return** $\mathcal{D}(\hat{z}_0)$

---

| | Inpainting (random) | Super Resolution | Gaussian Deblur | Motion Deblur |
|---|---|---|---|---|
| $k$ | 15 | 15 | 10 | 10 |
| $N_c$ | 3 | 3 | 3 | 3 |
| $\lambda$ | 0.07 | 0.15 | 0.27 | 0.27 |

Table 10: Experiment configurations for Latent DPS (LDPS).

## C.4 ALGORITHMIC DETAILS: PSLD

We plug MCLC into PSLD (Rout et al., 2023) and present the resulting algorithm in Algorithm 4. PSLD introduces an additional regularization term (gluing) and applies a measurement-consistency step with the regularized loss at every sampling iteration. We used the original PSLD implementation without any modification, except for applying MCLC. For our experiments, we use the DDIM sampling procedure with 1000 timesteps, and apply the MCLC step at every $k$-th step during sampling. At each corrector step, we perform $N_c$ corrector iterations with step size hyperparameter $\lambda$, as defined in Eq. (57). The detailed experimental settings, corresponding to those reported in Table 1, are summarized in Table 11 across each task.

---

**Algorithm 4** MCLC-PSLD

---

**Require:** $T, \boldsymbol{y}, \gamma, \zeta, \{\alpha_t\}_{t=1}^T, \{\bar{\alpha}_t\}_{t=1}^T, \{\tilde{\sigma}_t\}_{t=1}^T$

**Require:** $\mathcal{E}, \mathcal{D}, \mathcal{A}, \mathcal{A}\boldsymbol{x}_0^*, \boldsymbol{s}_\theta, N_c, \lambda,$

   $\boldsymbol{z}_T \sim \mathcal{N}(\boldsymbol{0}, \boldsymbol{I})$

   **for** $t = T$ **to** $1$ **do**

      $\hat{\boldsymbol{s}} \leftarrow \boldsymbol{s}_\theta(\boldsymbol{z}_t, t)$

      $\hat{\boldsymbol{z}}_0 \leftarrow \frac{1}{\sqrt{\bar{\alpha}_t}}(\boldsymbol{z}_t + (1 - \bar{\alpha}_t)\hat{\boldsymbol{s}})$

      $\boldsymbol{\epsilon} \sim \mathcal{N}(\boldsymbol{0}, \boldsymbol{I})$

      $\boldsymbol{z}_{t-1} \leftarrow \frac{\sqrt{\alpha_t}(1-\bar{\alpha}_{t-1})}{1-\bar{\alpha}_t}\boldsymbol{z}_t + \frac{\sqrt{\bar{\alpha}_{t-1}}(1-\alpha_t)}{1-\bar{\alpha}_t}\hat{\boldsymbol{z}}_0 + \tilde{\sigma}_t\boldsymbol{\epsilon}$

      $\boldsymbol{g}_t \leftarrow \zeta \nabla_{\boldsymbol{z}_t} \|\boldsymbol{y} - \mathcal{A}(\mathcal{D}(\hat{\boldsymbol{z}}_0))\|_2^2 + \gamma \nabla_{\boldsymbol{z}_t} \|\hat{\boldsymbol{z}}_0 - \mathcal{E}(\mathcal{A}^T \mathcal{A}\boldsymbol{x}_0^* + (\boldsymbol{I} - \mathcal{A}^T\mathcal{A})\mathcal{D}(\hat{\boldsymbol{z}}_0))\|_2^2$

      $\boldsymbol{z}'_{t-1} \leftarrow \boldsymbol{z}_{t-1} - \boldsymbol{g}_t$

      **// MCLC Correction Step**

      **if** $(t \mod k) = 0$ **then**

         **for** $c = 1, \ldots, N_c$ **do**

            $\boldsymbol{z}'_{t-1} \leftarrow \texttt{MCLC}(\boldsymbol{z}'_{t-1}, \mathbf{s}_\theta, t-1, \boldsymbol{g}_t)$         ▷ Corrector step size hyperparameter $\lambda$

   **end for**

   **return** $\mathcal{D}(\hat{z}_0)$

---

| | Inpainting (random) | Super Resolution | Gaussian Deblur | Motion Deblur |
|---|---|---|---|---|
| $k$ | 15 | 15 | 10 | 10 |
| $N_c$ | 3 | 3 | 3 | 3 |
| $\lambda$ | 0.07 | 0.15 | 0.27 | 0.27 |

Table 11: Experiment configurations for PSLD.

## C.5 ALGORITHMIC DETAILS: RESAMPLE

We plug MCLC into ReSample and present the resulting algorithm in Algorithm 5. ReSample applies the measurement-consistency step every 10 iterations, with a staged strategy: it skips the first one-third of the reverse process, performs pixel optimization in the middle one-third, and applies latent optimization with hard consistency in the final one-third. In addition, a standard DPS step is included throughout the reverse sampling process. For Resample, we used Diff-State implementation, which extends the original Resample code, with removal of the Diff-State module. To use a better-tuned setting, we adopted their implementation configuration. For our experiments, we adopt DDIM sampling with 50 steps. Following the setting in DiffStateGrad (Zirvi et al., 2025), we insert MCLC into the only latent optimization stage and DPS step. In the pixel optimization stage, we do not perform correction. Specifically, the pixel optimization stage performs 2000 ($N_{\text{pixel}}$) updates, while

the latent optimization stage performs 500 ($N_{\text{latent}}$) updates. Within the latent optimization stage, MCLC is applied every $k$ iterations, performing $N_c$ correction steps with step size $\lambda$ as defined in Eq. (57). Furthermore, we also apply MCLC to the DPS steps, where it is used at every iteration of the 50-step process. In this case, the number of corrector steps and step size are denoted by $N_c^{\text{DPS}}$ and $\lambda^{\text{DPS}}$, respectively. The detailed experimental settings, corresponding to those reported in Table 1, are summarized in Table 12 across each task.

---

**Algorithm 5** MCLC-Resample

---

**Require:** $T, \boldsymbol{y}, \zeta, \{\alpha_t\}_{t=1}^T, \{\bar{\alpha}_t\}_{t=1}^T, \{\tilde{\sigma}_t\}_{t=1}^T$

**Require:** $\mathcal{E}, \mathcal{D}, \mathcal{A}, \boldsymbol{s}_\theta, \gamma, C_{\text{pixel}}, C_{\text{latent}}, N_c, \lambda, N_c^{\text{DPS}}, \lambda^{\text{DPS}}$

   $\boldsymbol{z}_T \sim \mathcal{N}(\boldsymbol{0}, \boldsymbol{I})$             ▷ Initial noise vector

   **for** $t = T, \ldots, 1$ **do**

      $\hat{\boldsymbol{s}} \leftarrow \boldsymbol{s}_\theta(\boldsymbol{z}_t, t)$

      $\hat{\boldsymbol{z}}_0 \leftarrow \frac{1}{\sqrt{\bar{\alpha}_t}}(\boldsymbol{z}_t + (1 - \bar{\alpha}_t)\hat{\boldsymbol{s}})$

      $\boldsymbol{z}_{t-1} \leftarrow \frac{\sqrt{\alpha_t}(1-\bar{\alpha}_{t-1})}{1-\bar{\alpha}_t}\boldsymbol{z}_t + \frac{\sqrt{\bar{\alpha}_{t-1}}(1-\alpha_t)}{1-\bar{\alpha}_t}\hat{\boldsymbol{z}}_0 + \tilde{\sigma}_t\boldsymbol{\epsilon}$

      $\boldsymbol{g}_t \leftarrow \zeta\nabla_{\boldsymbol{z}_t}\|\boldsymbol{y} - \mathcal{A}(\mathcal{D}(\hat{\boldsymbol{z}}_0))\|_2^2$

      $\boldsymbol{z}'_{t-1} \leftarrow \boldsymbol{z}_{t-1} - \boldsymbol{g}_t$

      **// MCLC Correction Step**

      **for** $c = 1, \ldots, N_c^{\text{DPS}}$ **do**

         $\boldsymbol{z}'_{t-1} \leftarrow \texttt{MCLC}(\boldsymbol{z}'_{t-1}, \boldsymbol{s}_\theta, t-1, \boldsymbol{g}_t)$    ▷ Corrector step size hyperparameter $\lambda^{\text{DPS}}$

      **if** $t \in C_{\text{pixel}}$ **then**

         **// Pixel Optimization Step**

      **else if** $t \in C_{\text{latent}}$ **then**

         **// Latent Optimization Step**

         **for** $o = 1, \ldots, N_{\text{latent}}$ **do**

            $\boldsymbol{g} \leftarrow \zeta_t\nabla_{\boldsymbol{z}_t}\|\boldsymbol{y} - \mathcal{A}(\mathcal{D}(\hat{\boldsymbol{z}}_0))\|_2^2$

            Update $\boldsymbol{z}_{0|t}(\boldsymbol{y})$ using gradient $\boldsymbol{g}$

            **// MCLC Correction Step**

            **for** $c = 1, \ldots, N_c$ **do**

               $\boldsymbol{z}_{0|t}(\boldsymbol{y}) \leftarrow \texttt{MCLC}(\boldsymbol{z}_{0|t}(\boldsymbol{y}), \boldsymbol{s}_\theta, 0, \boldsymbol{g})$    ▷ Corrector step size hyperparameter $\lambda$

         **end for**

         $\boldsymbol{z}_{t-1} = \text{StochasticResample}(\hat{z}_0(y), \boldsymbol{z}'_t, \gamma)$

      **else**

         $\boldsymbol{z}_{t-1} = \boldsymbol{z}'_{t-1}$

      **end if**

   **end for**

   $\boldsymbol{x}_0 = \mathcal{D}(\boldsymbol{z}_0)$           ▷ Output reconstructed image

   **return** $\boldsymbol{x}_0$

---

| | Inpainting (random) | Super Resolution | Gaussian Deblur | Motion Deblur | HDR | Nonlinear Deblur |
|---|---|---|---|---|---|---|
| $k$ | 5 | 5 | 10 | 10 | 5 | 5 |
| $N_c$ | 3 | 3 | 5 | 5 | 3 | 3 |
| $\lambda$ | 0.15 | 0.15 | 0.15 | 0.15 | 0.15 | 0.07 |
| $N_c^{\text{DPS}}$ | 1 | 1 | 1 | 1 | 1 | 1 |
| $\lambda^{\text{DPS}}$ | 0.05 | 0.15 | 0.15 | 0.15 | 0.10 | 0.07 |

Table 12: Experiment configurations for ReSample.

## C.6 ALGORITHMIC DETAILS: LATENT DAPS

We plug MCLC into LatentDAPS (Zhang et al., 2025a) and present the resulting algorithm in Algorithm 6. LatentDAPS, decoupling consecutive steps of diffusion sampling trajectory, does not conduct the reverse sampling process. Following DiffStateGrad (Zirvi et al., 2025), we apply corrections to the log-posterior gradient $\nabla_{\boldsymbol{z}_t} \log p(\boldsymbol{z}_{0|t}|\boldsymbol{y})$. For LatentDAPS, we used the original implementation without any modifications, and followed the configuration described in the paper. For our experiments, we use 50 annealing steps and integrate the ODE with two solver steps. The MCLC step is applied every $k$ iterations during each annealing-based posterior sampling stage. At each corrector step, we perform $N_c$ updates with step size hyperparameter $\lambda$, as defined in Eq. (57). Additionally, we correct the annealed noisy latent state at every annealing step, using $N_c^{\text{int}}$ corrector updates with step size hyperparameter $\lambda^{\text{int}}$. The detailed experimental settings, corresponding to those reported in Table 1, are summarized in Table 13.

---

**Algorithm 6** MCLC-LatentDAPS

---

**Require:** annealing noise schedule $\sigma_t, \{t_i\}_{i\in\{0,\dots,N_A\}}, \mathcal{E}, \mathcal{D}, \mathcal{A}, \boldsymbol{s}_\theta, \boldsymbol{y}, \gamma_t, N_c, \lambda, N_c^{\text{int}}, \lambda^{\text{int}}$

    Sample $\boldsymbol{z}_T \sim \mathcal{N}(\boldsymbol{0}, \sigma_T^2 \boldsymbol{I})$

    **for** $i = N_A, \dots, 1$ **do**

        Compute $\hat{\boldsymbol{z}}_0^{(0)} = \hat{\boldsymbol{z}}_0(\boldsymbol{z}_{t_i})$ by solving the probability flow ODE in Eq. (39) with $\boldsymbol{s}_\theta$

        **for** $j = 0, \dots, N-1$ **do**

            $\boldsymbol{g} \leftarrow \nabla_{\hat{\boldsymbol{z}}_0} \log p(\hat{\boldsymbol{z}}_0^{(j)}|\boldsymbol{z}_{t_i}) + \nabla_{\hat{\boldsymbol{z}}_0} \log p(\boldsymbol{z}_{t_i}|\boldsymbol{y})$

            $\hat{\boldsymbol{z}}_0^{(j+1)} \leftarrow \hat{\boldsymbol{z}}_0^{(j)} + \gamma_t \boldsymbol{g}' + \sqrt{2\gamma_t}\,\boldsymbol{\epsilon}_j, \quad \boldsymbol{\epsilon}_j \sim \mathcal{N}(\boldsymbol{0}, \boldsymbol{I})$

            **// MCLC Correction Step**

            **for** $c = 1, \dots, N_c$ **do**

                $\hat{\boldsymbol{z}}_0^{(j+1)} \leftarrow \texttt{MCLC}(\hat{\boldsymbol{z}}_0^{(j+1)}, \boldsymbol{s}_\theta, 0, \boldsymbol{g})$         $\triangleright$ Corrector step size hyperparameter $\lambda$

        **end for**

        Sample $\boldsymbol{z}_{t_{i-1}} \sim \mathcal{N}(\hat{\boldsymbol{z}}_0^{(N)}, \sigma_{t_{i-1}}^2 \boldsymbol{I})$

        **// MCLC Correction Step**

        **for** $c = 1, \dots, N_c^{\text{int}}$ **do**

            $\boldsymbol{z}_{t_{i-1}} \leftarrow \texttt{MCLC}(\boldsymbol{z}_{t_{i-1}}, \boldsymbol{s}_\theta, t_{i-1}, \boldsymbol{g})$         $\triangleright$ Corrector step size hyperparameter $\lambda^{\text{int}}$

    **end for**

    **return** $\boldsymbol{z}_0$

---

| | Inpainting (random) | Super Resolution | Gaussian Deblur | Motion Deblur | HDR | Nonlinear Deblur |
|---|---|---|---|---|---|---|
| $k$ | 5 | 5 | 5 | 5 | 5 | 5 |
| $N_c$ | 3 | 3 | 3 | 3 | 3 | 1 |
| $\lambda$ | 0.10 | 0.15 | 0.10 | 0.15 | 0.10 | 0.10 |
| $N_c^{\text{int}}$ | 1 | 0 | 1 | 3 | 1 | 1 |
| $\lambda^{\text{int}}$ | 0.15 | 0 | 0.15 | 0.15 | 0.15 | 0.15 |

Table 13: Experiment configurations for LatentDAPS.

## LARGE LANGUAGE MODELS USAGE

We use large language models (LLMs) solely as general-purpose writing assistants to polish grammar and improve readability. The research ideas, technical contributions, experiments, and analyses were entirely conceived and carried out by the authors.

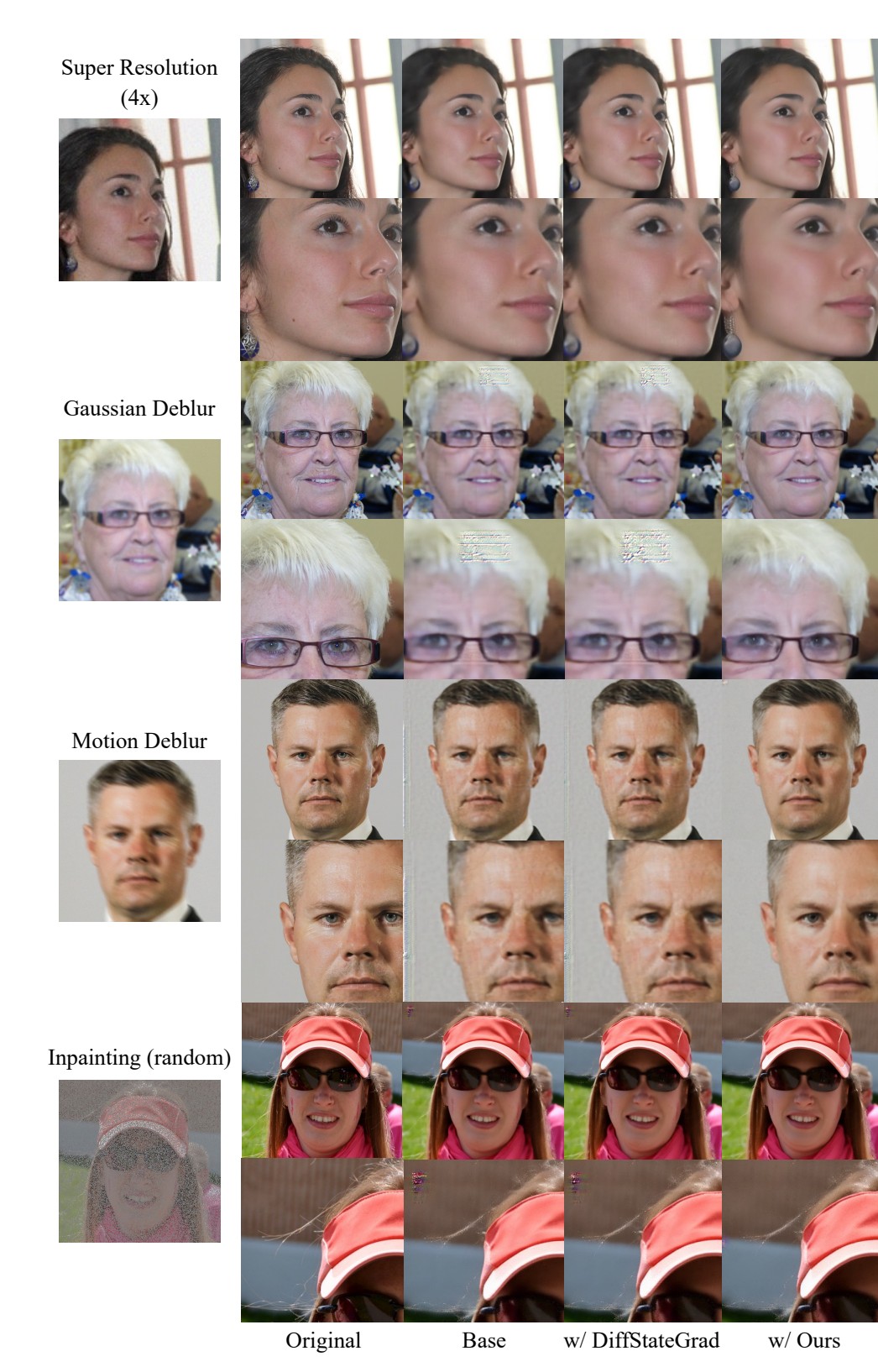

Figure 21: Qualitative comparison of LDPS, LDPS-DiffStateGrad, LDPS-MCLC on FFHQ.

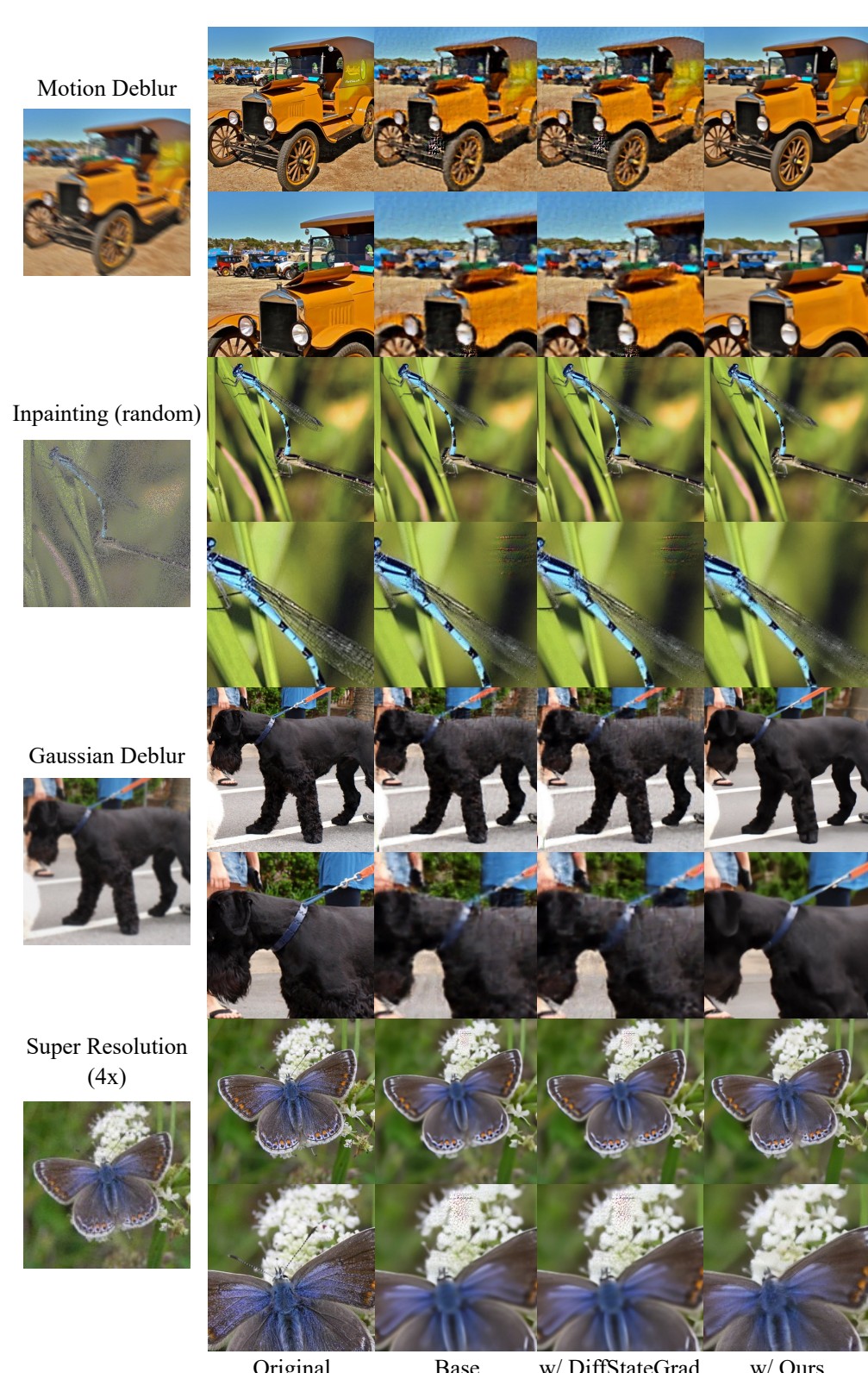

Figure 22: Qualitative comparison of LDPS, LDPS-DiffStateGrad, LDPS-MCLC on ImageNet.

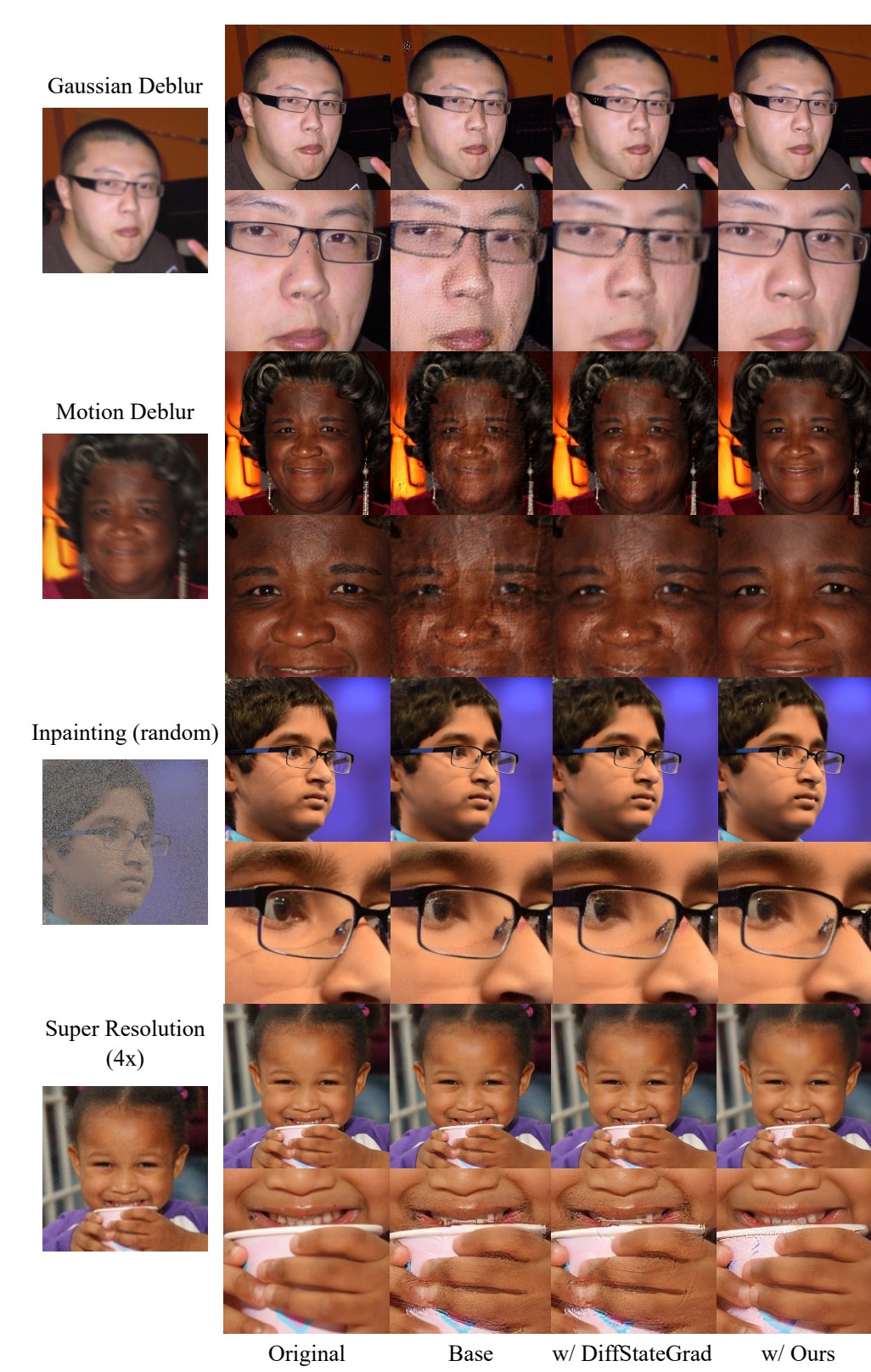

Figure 23: Qualitative comparison of PSLD, PSLD-DiffStateGrad, PSLD-MCLC on FFHQ.

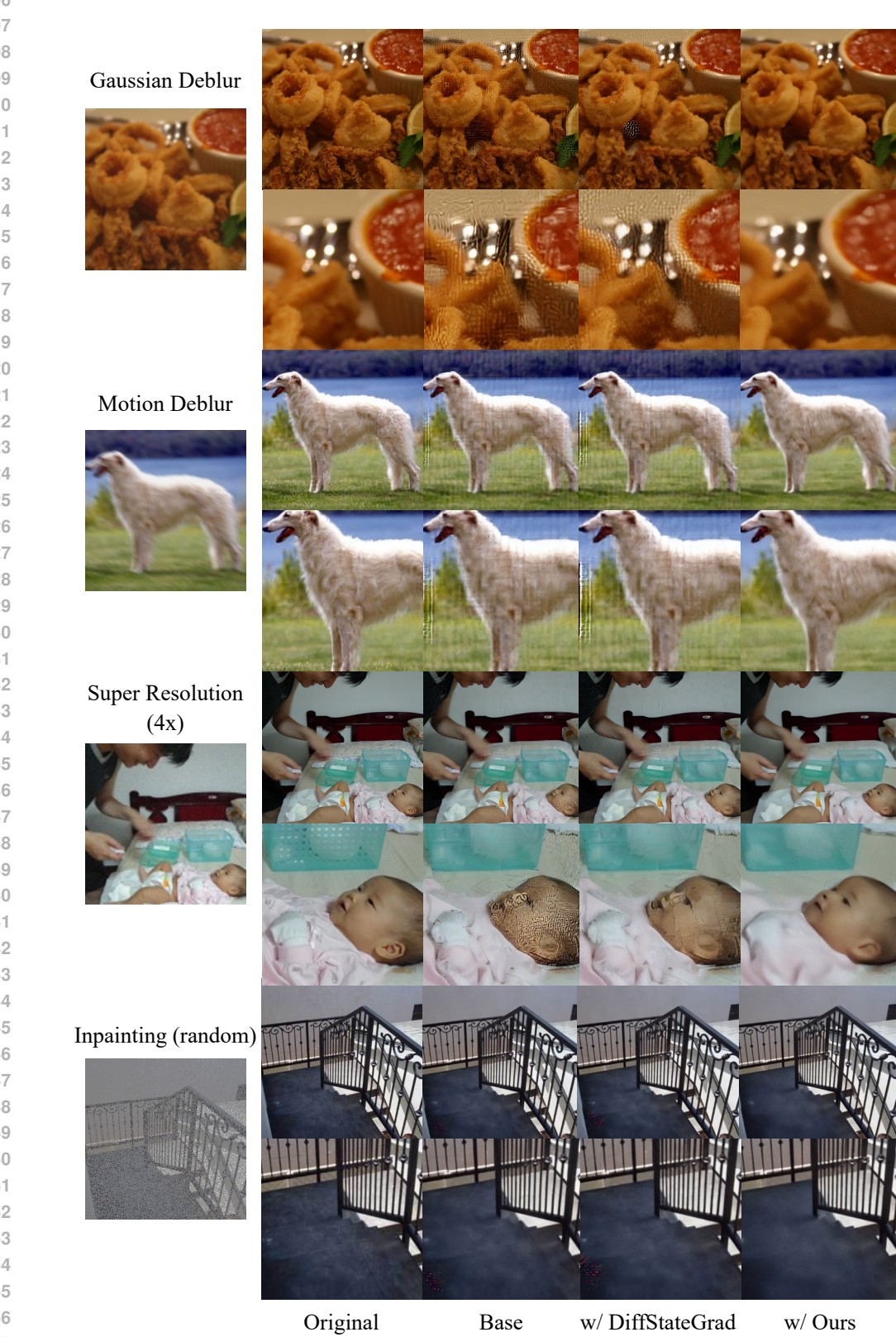

Figure 24: Qualitative comparison of PSLD, PSLD-DiffStateGrad, PSLD-MCLC on ImageNet.

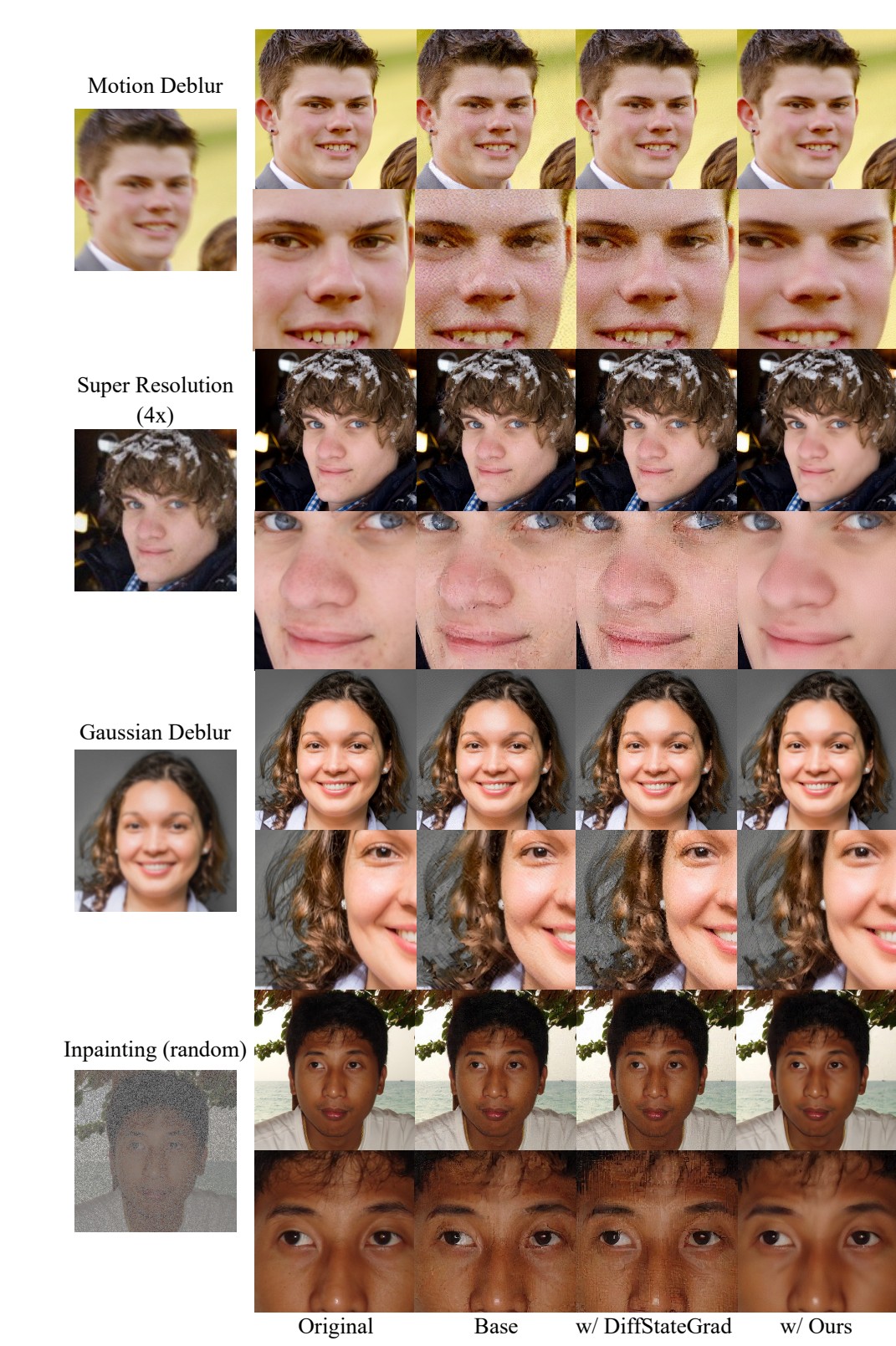

Figure 25: Qualitative comparison of Resample, Resample-DiffStateGrad, Resample-MCLC on FFHQ.

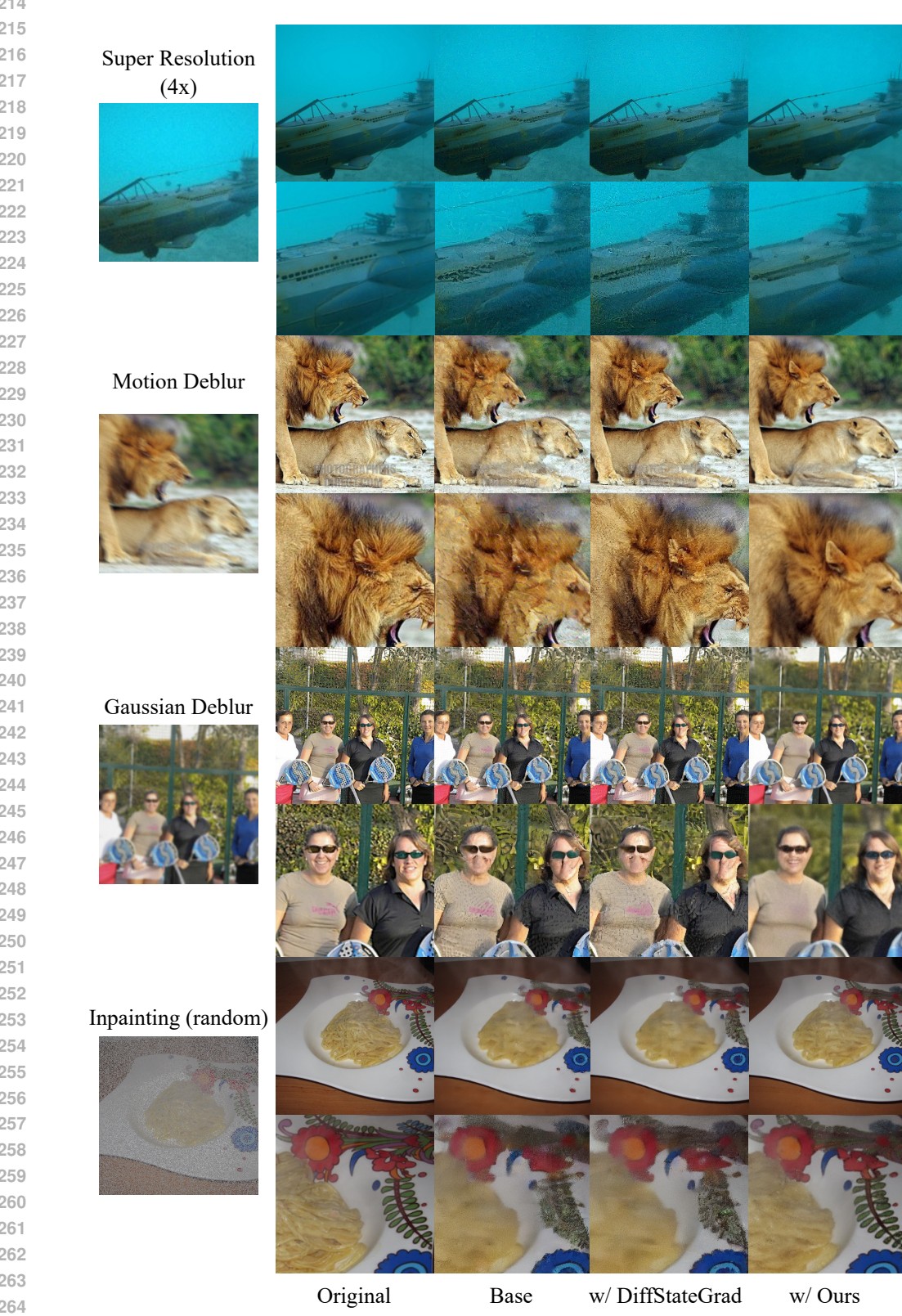

Figure 26: Qualitative comparison of Resample, Resample-DiffStateGrad, Resample-MCLC on ImageNet.

