# OpenReview forum: "Measurement-Consistent Langevin Corrector: A Remedy for Latent Diffusion Inverse Solvers"
_ICLR.cc/2026/Conference — Submitted to ICLR 2026_

### Official Review · Reviewer_Udz6 · 2025-10-28

**Soundness:** 2
**Presentation:** 2
**Contribution:** 2
**Rating:** 4
**Confidence:** 3

**Summary:**

The authors propose a Langevin corrector for inverse solvers using a latent diffusion model (LDM) as the prior. The correction step preserves measurement consistency by being orthogonal to the measurement gradient. The authors show results on many inverse problems with Stable Diffusion v1.5 as the LDM prior, adding the proposed corrector on top of many existing latent diffusion inverse solvers. They also compare to existing solvers that aren’t plug-and-play-compatible with their corrector. In many situations, they find improvement in PSNR, LPIPS, FID, and Patch FID. They also provide an experiment showing that “blob” artifacts in decoded images arise from scaled outliers in the latent space. They claim that their proposed corrector may help alleviate this type of artifact.

**Strengths:**

* The proposed corrector is a principled way to reduce artifacts that may arise with existing LDM solvers.
* The corrector is easy to plug in to existing solvers.
* The authors evaluate on a variety of inverse problems and plenty of baselines.

**Weaknesses:**

* The technical contribution is quite marginal; it is just a Langevin corrector step that follows directly from predictor-corrector sampling with Langevin dynamics, and the measurement consistency is a minor additional piece.
* The qualitative results in Figure 5 are not convincing. I have to really zoom in to see some artifacts. And I am not convinced that the baseline artifacts in Figure 1 are in fact artifacts. They look like they could be plausible artifacts (e.g., from JPEG compression) in images that the LDM was trained on.
* The proposed corrector doesn’t provide an obvious improvement to LatentDAPS.
* The blob analysis is informative, but it is unclear how and by how much the proposed corrector would alleviate such artifacts.
* Evaluating a Patch FID score with 3x3 patches does not seem meaningful to me.
* I’m confused about the fundamental assumption being demonstrated by Figure 3. I see that the KL divergence is quite bad at the beginning of reverse diffusion (without the corrector), but isn’t it fine as long as the KL divergence approaches 0, since what we care about is the distribution at time 0? The plot in Figure 3 shows that the KL divergences with and without the corrector approach the same value by time 0. I have a similar question based on Figure 2. I can see that the latents look messier with the baseline, but that doesn’t necessarily mean their decoded images are bad.

**Questions:**

* Please clarify the last weakness point. Why is it a problem for the KL divergence to be high at the beginning of reverse diffusion process, even if it approaches a small value by the time reverse diffusion is done?
* How can you be sure that what appear to be artifacts are not actually coming from the true prior?

---

> ### Author Response · Authors · 2025-11-22
> **Author response (1/3)**
>
> We thank the reviewer $\textcolor{#70B7BA}{Udz6}$ for constructive comments and feedback. We provide further clarification and additional results below. We have also included additional qualitative evidence, and we kindly ask the reviewer to look over the revised paper, especially the parts we have referred to.
>
> > **Q1 and W6. Clarification on Fig. 3. Why is a high early-stage KL divergence problematic, even though the curves converge by time 0?**
>
> **Clarification of `Fig.3.`** We would like to respectfully clarify the x-axis in `Fig.3`: the reverse sampling step=0 means pure noise at the beginning of diffusion. To clarify this, we have added the pure noise sample at reverse sampling step=0 in `Fig.3`. At the beginning of reverse diffusion (time=1000, reverse sampling step=0), both paths are close to pure noise, so the KL gap is naturally small. However, during the inverse solver dynamics in reverse diffusion, the base solver (w/o corrector, red line) diverges and ends with a large gap at time 0 (reverse sampling step=1000), whereas MCLC stabilizes the dynamics and reduces this gap. Therefore, at the end, only the corrected one avoids large off-stationary drift.
>
> **Clarification on the interpretation of Messy latents.** As the reviewer mentioned, the visually messy latent does not necessarily imply a degraded reconstruction. The intention of `Fig.2` is not to claim that every visually problematic latent produces visible artifacts, but to illustrate an instability in the reverse-diffusion dynamics. While the decoder may still map some of these latents to plausible images, such latents deviate from the data manifold and induce undesirable reverse dynamics that may lead to unstable or unreliable solutions.
>
> > **Q2 and W2. How can we be certain that the artifacts are not inherent to the true prior?**
>
> We first note that these artifacts are not unique observations from our work. Several prior studies [R1,2] on LDM-based inverse solvers have already reported these artifacts, indicating that this issue arises specifically from the structural behavior of the LDM-based inverse problem solver rather than from the generative prior itself.
>
> To provide more convincing evidence, we include two additional analyses:
>
> - (1) The same solvers exhibit artifacts even when using different priors (Stable Diffusion v2.1 and Realistic-vision v5.1) in `Appendix B.12` (`Fig.18`). This indicates that the artifacts do not stem from a specific prior but are tied to the dynamics of inverse solvers. Even with these other LDMs, MCLC still provides performance gains. (see `Table 9`).
>
> |                           | **Gaussian**                   |    **Deblur**   |        |        | **Super**         |  **Resolution**     |   (4×)     |        |
> |---------------------------|----------------------------------------|-------|--------|--------|------------------------------------|-------|--------|--------|
> |                           | PSNR (↑) | LPIPS (↓) | FID (↓) | P-FID (↓) | PSNR (↑) | LPIPS (↓) | FID (↓) | P-FID (↓) |
> | RV v5.1 (base)            | 26.69    | 0.380     | 110.38 | 78.25     | **27.96** | 0.305     | 71.55   | 54.91     |
> | RV v5.1 (ours)            | **26.75**| **0.334** | **95.46** | **56.66** | 27.82     | **0.295** | **68.2** | **50.10** |
> | SD v2.1 (base)            | 26.22    | 0.403     | 116.49 | 120.94    | **28.70**     | **0.244** | 60.34   | 49.99     |
> | SD v2.1 (ours)            | **26.69**| **0.335** | **91.13** | **52.76** | 28.64     | 0.246     | **60.33** | **48.30** |
>
>
> - (2) Pure generation from the LDM does not exhibit such artifacts, which are clearly different from JPEG-like compression artifacts (please refer to `Fig. 19`). To support this point clearly, we select a sample and compare reconstructed images through:
>   - (a) DDIM inversion [R3], and
>   - (b) performing the LDM-based inverse problem solving on a measurement obtained from the same sample.
>
> Both processes use the same LDM prior and sample, yet the artifacts appear only in the LDM-based inverse-solved output, not in the reconstruction from (a) that follows the stable generative process of the model (please refer to `Fig.19`).
>
> These results do not prove that instability of the LDM-based inverse solvers is the only cause, but they clearly rule out the pretrained LDM prior as the sole source; the artifacts arise during inverse solving, not from the prior. Furthermore, if these artifacts originated from the true prior, then enhancing prior fidelity via MCLC would exacerbate artifacts instead of mitigating them.
>
> - [R1] Chung et al., Prompt-tuning Latent Diffusion Models for Inverse Problems, ICML 2024.
> - [R2] Raphaeli et al., SILO: Solving Inverse Problems with Latent Operators, ICCV 2025.
> - [R3] Mokady et al., Null-text Inversion for Editing Real Images using Guided Diffusion Models, CVPR 2023.

---

> ### Author Response · Authors · 2025-11-22
> **Author response (2/3)**
>
> > **W1. Whether the contribution is merely a Langevin corrector step, thus technically marginal, and measurement consistency is a minor additional piece**
>
> We respectfully clarify that our contribution is not marginal, nor is the measurement-consistency component a minor addition. Our work aims to explain why latent diffusion–based inverse solvers exhibit instability and how this behavior can be corrected in a principled manner. Our contribution can be summarized as follows:
>
> **(1) First identification of instability originating from a reverse-dynamics discrepancy in LDM-based inverse solvers.**
> To our knowledge, we first characterize that the instability arises from a gap between the true reverse diffusion dynamics and the solver dynamics (`Fig. 3`). We formulate this gap using the KL divergence at each time marginal distribution and show that reducing it stabilizes the solver, improves reconstruction quality, and suppresses artifacts. This aspect has not been examined in prior works.
>
> **(2) Measurement-consistent Langevin correction**
> Although the Langevin correction is used in standard PC sampling to reduce neural score approximation error, our use of Langevin dynamics is fundamentally different. We employ it as a principled mechanism to reduce the KL divergence from the correct time-marginal distribution, which directly targets the instability. In fact, Langevin dynamics follows the steepest descent of this KL divergence, providing a theoretically grounded way [R4,5].
>
> In addition, measurement consistency is not a minor component; it is a fundamental requirement in inverse problems [R6].  In inverse problems, a naïve Langevin correction can produce visually plausible results while disturbing measurement consistency, leading to suboptimal solutions (please refer to `L455-465` and `Table 5`). In contrast, MCLC reduces the reverse-dynamics gap while preserving measurement consistency, providing an actual remedy for latent-diffusion inverse solvers. To our knowledge, such a measurement-consistent correction scheme has not been presented in prior works.
>
> **(3) Beyond the linear manifold assumption.**
> Several previous diffusion-based inverse solvers [R7,8] proposed manifold-preserving methods built on a linear manifold assumption. However, this assumption often does not hold in the latent space [R9]. Our method operates without relying on such manifold constraints and provides a more reliable solution, offering a clear conceptual advance over existing approaches.
>
> - [R4] Jordan, Kinderlehrer, and Otto, The Variational Formulation of the Fokker-Planck Equation, SIAM Journal on Mathematical Analysis 1998.
> - [R5] Vempala and Wibisono, Rapid Convergence of the Unadjusted Langevin Algorithm: Isoperimetry Suffices, NeurIPS 2018.
> - [R6] Groetsch, Inverse Problems in the Mathematical Sciences, Springer 1993.
> - [R7] He et al., Manifold Preserving Guided Diffusion, ICLR 2024.
> - [R8] Zirvi et al., Diffusion State-Guided Projected Gradient for Inverse Problems, ICLR 2025.
> - [R9] Song et al., Solving Inverse Problems with Latent Diffusion Models via Hard Data Consistency, ICLR 2024.

---

> ### Author Response · Authors · 2025-11-22
> **Author response (3/3)**
>
> > **W3. Less improvement in LatentDAPS**
>
> As discussed in `Appendix B.9`, the lower compatibility with LatentDAPS is because it has a specific design that breaks the reverse diffusion trajectory by re-initializing each iteration with annealed noise. Although this design structure is not aligned with MCLC’s role of stabilizing the reverse trajectory, it does not imply that the general applicability of MCLC is limited. `Tables 2, 3, and 6` show the results on recent advanced solvers, including TReg [R10] and FlowChef [R11], indicating the consistent improvements of MCLC. For further details and discussion, please refer to `Appendix B.1`. These results demonstrate that MCLC is well compatible with recent advanced solvers and its future applicability.
>
> - [R10] Kim et al., Regularization by Texts for Latent Diffusion Inverse Solvers, ICLR 2025.
> - [R11] Patel et al., Steering Rectified Flow Models in the Vector Field for Controlled Image Generation, ICCV 2025.
>
> > **W4. How and how much MCLC alleviates blob artifacts**
>
> Thanks for recognizing the value of our blob-artifact analysis. As discussed in the main paper, blob artifacts arise when latent values are amplified during reverse dynamics and pushed far outside the feasible range. Although MCLC attempts to pull such values back, this is relatively weaker than the amplification, which explains why some artifacts cannot be fully corrected.
>
> To clarify how much MCLC mitigates this issue, we have provided additional qualitative results in `Appendix B.8` (`Fig. 15`). These results show that MCLC substantially reduces the magnitude of the out-of-range latent values, leading to noticeable suppression of blob artifacts in decoded images.
>
> > **W5. Patch FID with 3x3 patches meaningful?**
>
> We would like to note that the Patch FID has already been adopted in prior works [R12-14] as a metric for evaluating more detailed image quality and localized degradations.
>
> To address a potential concern that 3×3 patches may be too coarse to be meaningful, we additionally report Patch FID results using finer patch grids: 5×5 patches (patch size = 103×103) and 8×8 patches (patch size = 64×64). Across all patch granularities, MCLC consistently achieves a lower Patch FID.
>
>
> |          | Gaussian        |   Deblur     |        | Motion          |   Deblur     |        |
> |----------|-------------------------|--------|--------|-------------------------|--------|--------|
> |          | 3×3                    | 5×5    | 8×8    | 3×3                    | 5×5    | 8×8    |
> | Base     | 90.54                  | 85.20  | 77.17  | 102.60                 | 85.20  | 82.15  |
> | Ours     | **59.13**              | **50.37** | **58.41** | **60.05**             | **50.37** | **59.88** |

---

### Official Review · Reviewer_D5xV · 2025-10-30

**Soundness:** 4
**Presentation:** 3
**Contribution:** 3
**Rating:** 6
**Confidence:** 3

**Summary:**

This paper argues that existing LDM-based inverse solvers often suffer from artifacts and degraded quality. The authors attribute this to a divergence between the solver's dynamics and the true reverse diffusion process. They propose a plug-and-play module that applies a projected Langevin dynamics step to "correct" the latent state at each iteration, pulling it back toward the true data manifold. The evaluations are comprehensive.

**Strengths:**

1. The proposed MCLC is a clever solution. The use of Langevin dynamics to reduce KL divergence and the projection to maintain measurement consistency are both well-justified.

2. The method is designed as a modular add-on that can improve multiple existing solvers (LDPS, PSLD, ReSample), which is a plus.

3. The method achieves significant and consistent improvements in perceptual metrics (LPIPS, FID, and the proposed P-FID) across a wide range of tasks and datasets.

**Weaknesses:**

1. The appendix reveals that MCLC requires its own set of hyperparameters that must be individually tuned for every base solver and every restoration task. This is a massive tuning effort that directly contradicts the paper's claims of simplicity and generality.

2. The method fails to provide significant improvements for the more advanced LatentDAPS solver. If the method only works on older/simpler solvers and is incompatible with newer SOTA methods, its contribution is limited to being a "patch" for those specific methods rather than a general-purpose solution.

3. The authors concede that improvements in PSNR are "modest". This indicates the method is primarily an artifact-removal and perceptual enhancement technique, which may be less valuable for scientific or high-fidelity applications where accuracy (PSNR) is more important than visual pleasantness (LPIPS/FID).

4. The paper repeatedly critiques prior work (e.g., DiffStateGrad, MPGD) for relying on a strong linear manifold assumption that fails to hold in latent space. However, MCLC's own projection is based on the first-order gradient, which is itself a linear approximation of the measurement-consistent manifold. The paper even admits its consistency guarantee is only "up to the first-order Taylor expansion", effectively trading one linear approximation for another.

**Questions:**

1. Given the extensive task-specific tuning in Tables 4-7, how can the authors justify the zero-shot and plug-and-play claims? Is there a single, default set of MCLC hyperparameters that provides a reasonable improvement across all tasks?

2. Can the authors elaborate on why their first-order gradient projection is fundamentally superior to the linear manifold assumption they critique? Both appear to be linear approximations of a nonlinear reality.

3. The paper states MCLC is less compatible with LatentDAPS because LatentDAPS does not inherit reverse diffusion dynamics. Does this imply MCLC is fundamentally incompatible with any solver that breaks from the standard DDIM/DDPM reverse process, limiting its future applicability?

---

> ### Author Response · Authors · 2025-11-22
> **Author response (1/4)**
>
> We sincerely thank the reviewer $\textcolor{#A9CF54}{D5xV}$ for invaluable comments and appreciate the recognition of our method justification and comprehensive results. We address the raised points and questions below.
>
> > **W1 and Q1. If MCLC requires task-specific hyperparameter tuning, doesn’t this contradict the claims of simplicity and generality?**
>
> We appreciate the valuable feedback. We respectfully would like to note that task-specific parameter adjustment is standard practice for most inverse-problem solvers [R1-8], not unique to MCLC. Even within the same task, the solver parameters naturally depend on the degree of degradation (e.g., blur kernel size and intensity, inpainting ratio, noise level), i.e., existing methods also require hyperparameter tuning. What MCLC contributes is a plug-and-play mechanism that directly attaches to existing solvers and enhances their zero-shot stability, supporting our claims of simplicity and generality.
>
> **A single default setting provides reasonable improvements across tasks.** To directly address Q1, we have newly included a table showing that a single default MCLC hyperparameter set provides reasonable improvements across linear tasks in `Appendix B.6` (`Table 8`). The partial version of this table is provided below. This demonstrates the generality of MCLC and is consistent with the theoretical bound in `Eq.65`, which indicates that an appropriate step size together with a sufficient number of correction iterations yields a simple and reliable default configuration. Our hyperparameter search was conducted to further balance efficiency and performance under different degradation levels, and we have also provided a guideline for choosing MCLC hyperparameters in `Appendix B.14`.
>
> We agree that making the hyperparameter choice more adaptive to task or degradation levels would be a promising direction, and we included this point in the Conclusion (`Sec.6`) as future discussion.
>
> - [R1] Daubechies, Defrise, and De Mol, An iterative thresholding algorithm for linear inverse problems with a sparsity constraint, Communications on Pure and Applied Mathematics 2004.
> - [R2] Beck and Teboulle, A Fast Iterative Shrinkage-Thresholding Algorithm for Linear Inverse Problems, SIAM Journal on Imaging Sciences 2009.
> - [R3] Hu and Jacob, Higher Degree Total Variation (HDTV) Regularization for Image Recovery, IEEE TIP 2011.
> - [R4] Chen et al., Learning Blind Denoising Network for Noisy Image Deblurring, ICASSP 2020.
> - [R5] Hurault et al., Convergent Bregman Plug-and-Play Image Restoration for Poisson Inverse Problems, NeurIPS 2023
> - [R6] Fabian et al., DiracDiffusion: Denoising and Incremental Reconstruction with Assured Data-Consistency, ICML 2024.
> - [R7] Pandey and Yang et al., Fast Samplers for Inverse Problems in Iterative Refinement Models, NeurIPS 2024.
> - [R8] Shen et al., Understanding and Improving Training-free Loss-based Diffusion Guidance, NeurIPS 2024.

---

> ### Author Response · Authors · 2025-11-22
> **Author response (2/4)**
>
> | Task            | Base      | Method               | PSNR ↑  | LPIPS ↓ | FID ↓   | PFID ↓  |
> |-----------------|-----------|----------------------|---------|---------|---------|---------|
> | Gaussian Deblur | LDPS      | Base                 | 27.61   | 0.349   | 100.10  | 93.55   |
> |                 |           | **Ours (single default)** | **27.99** | **0.334** | **85.84** | **71.62** |
> |                 |           | Ours (tuned)         | 28.14   | 0.303   | 80.83   | 54.74   |
> |                 | PSLD      | Base                 | 27.84   | **0.314** | 89.18 | 90.54   |
> |                 |           | **Ours (single default)** | **27.94** | 0.317 | **79.81** | **69.39** |
> |                 |           | Ours (tuned)         | 27.97   | 0.286   | 66.28   | 59.13   |
> |                 | ReSample  | Base                 | 26.44   | 0.368   | **75.17** | 148.11  |
> |                 |           | **Ours (single default)** | **27.33** | **0.355** | 80.90 | **96.17** |
> |                 |           | Ours (tuned)         | 27.25   | 0.353   | 78.38   | 106.16  |
> | Motion Deblur  | LDPS      | Base                 | 26.54   | 0.390   | 118.77  | 112.74  |
> |                |           | **Ours (single default)** | **27.03** | **0.363** | **99.71** | **81.82** |
> |                |           | Ours (tuned)         | 27.45   | 0.318   | 82.94   | 55.55   |
> |                | PSLD      | Base                 | 26.87   | **0.343** | 106.34 | 102.60 |
> |                |           | **Ours (single default)** | **26.92** | 0.348 | **90.92** | **72.30** |
> |                |           | Ours (tuned)         | 26.86   | 0.308   | 74.64   | 60.05   |
> |                | ReSample  | Base                 | 22.45   | 0.635   | 108.14  | 174.52  |
> |                |           | **Ours (single default)** | **24.19** | **0.599** | **103.70** | **114.85** |
> |                |           | Ours (tuned)         | 24.24   | 0.588   | 102.02  | 118.87  |
> | Super Resolution (4×)| LDPS      | Base             | **28.47** | **0.301** | 78.08 | 69.66 |
> |                      |           | **Ours (single default)**| 28.19   | 0.307   | **74.33** | **57.48** |
> |                      |           | Ours (tuned)         | 28.34   | 0.283   | 74.78  | 58.55  |
> |                      | PSLD      | Base             | **27.69** | 0.265 | 63.95 | 63.47 |
> |                      |           | **Ours (single default)**| 27.44   | **0.261** | **61.09** | **52.43** |
> |                      |           | Ours (tuned)         | 27.33   | 0.267   | 62.12   | 58.13   |
> |                      | ReSample  | Base                 | 26.40   | 0.347   | 70.16   | 133.15  |
> |                      |           | **Ours (single default)** | **27.73** | **0.264** | **55.38** | **68.55** |
> |                      |           | Ours (tuned)         | 28.32   | 0.236   | 53.85   | 78.08   |
> | Inpainting (Random) | LDPS      | Base                 | 31.22   | 0.171   | 48.88   | 83.30   |
> |                     |           | **Ours (single default)** | **31.31** | **0.167** | **47.76** | **79.23** |
> |                     |           | Ours (tuned)         | 31.28   | 0.169   | 48.05   | 81.68   |
> |                     | PSLD      | Base                 | 30.14   | 0.222   | 58.84   | **79.59** |
> |                     |           | **Ours (single default)** | **31.30** | **0.167** | **48.04** | 79.74 |
> |                     |           | Ours (tuned)         | 30.73   | 0.185   | 49.80   | 72.69   |
> |                     | ReSample  | Base                 | 27.27   | 0.374   | 103.17  | 133.80  |
> |                     |           | **Ours (single default)** | **28.75** | **0.296** | **85.90** | **103.25** |
> |                     |           | Ours (tuned)         | 29.35   | 0.235   | 75.65   | 108.27  |

---

> ### Author Response · Authors · 2025-11-22
> **Author response (3/4)**
>
> > **W2 and Q3. MCLC’s compatibility with recent advanced LDM-based inverse problem solver and its applicability beyond the standard DDIM/DDPM reverse process.**
>
> We thank the reviewer for the insightful comment, which helps make our work more convincing.
> We would like to clarify that lower compatibility with LatentDPAS does not imply that MCLC has limited applicability to recent advanced inverse solvers. As discussed in `Appendix B.9`, LatentDAPS is less compatible with MCLC due to its particular design: it does not follow the reverse trajectory but instead decouples consecutive steps by re-initializing each iteration, i.e., performing annealed initializations for transitions. This specific mechanism is not well aligned with MCLC, whose role is to stabilize the reverse trajectory.
> To further demonstrate MCLC’s applicability, we have newly added drop-in improvements in the recent advanced solver (TReg [R9]) and flow-based model (FlowChef [R10]), showing consistent gains (see `Tables 2, 3, and 6,` and `Appendix B.1`). These results confirm that (1) MCLC remains effective and compatible with newer methods providing stabilized solutions (`Table 2`) and (2) MCLC can also be applied to flow-based approaches beyond standard DDPM/DDIM-style solvers (`Tables 3 and 6`). This highlights MCLC’s future applicability and potential.
>
> | Method         | Gaussian |    Deblur      |          | Super  |   Resolution      |     (16×)       |
> |----------------|------------------|----------|----------|--------------------------|----------|----------|
> |                | PSNR (↑)         | LPIPS (↓)| FID (↓)  | PSNR (↑)                 | LPIPS (↓)| FID (↓)  |
> | TReg           | 20.84            | 0.476    | 37.12    | 18.39                    | **0.633**| 44.91    |
> | TReg w/ Ours   | **21.33**        | **0.456**| **27.62**| **19.15**                | 0.646    | **33.86**|
>
>
>
> | Method           | Gaussian |      Deblur    |          | Motion |     Deblur     |          |
> |------------------|------------------|----------|----------|----------------|----------|----------|
> |                  | PSNR (↑)         | LPIPS (↓)| FID (↓)  | PSNR (↑)       | LPIPS (↓)| FID (↓)  |
> | FlowChef         | 23.76            | 0.364    | 106.76   | 22.58          | 0.519    | 185.44   |
> | FlowChef w/ Ours | **28.52**        | **0.288**| **77.36**| **26.01**      | **0.353**| **100.40** |
>
>
> | Method           | Super Resolution  |     12× (Avgpool)     |          | Super Resolution |   12x  (Bicubic)       |         |
> |------------------|-------------------|----------|----------|-------------------|----------|----------|
> |                  | PSNR (↑)          | LPIPS (↓)| FID (↓)  | PSNR (↑)          | LPIPS (↓)| FID (↓)  |
> | FlowChef         | **25.26**         | 0.480    | 181.15   | **25.30**         | 0.501    | 174.51   |
> | FlowChef w/ Ours | 24.81             | **0.393**| **125.57**| 24.85            | **0.424**| **130.70** |
>
>
> In addition, these experiments use the same MCLC hyperparameter setting, which may be helpful for addressing W1 and Q1.
>
> - [R9] Kim et al., Regularization by Texts for Latent Diffusion Inverse Solvers, ICLR 2025.
> - [R10] Patel et al., Steering Rectified Flow Models in the Vector Field for Controlled Image Generation, ICCV 2025.

---

> ### Author Response · Authors · 2025-11-22
> **Author response (4/4)**
>
> > **W4 and Q2. Can the authors elaborate on why MCLC is superior to the linear manifold assumption they critique?**
>
> We would like to clarify that the two notions of linearity being compared here are entirely different. The linear manifold assumption is a geometric assumption about the structure of the diffusion data manifold at timestep t, illustrated as the bottom region in the zoomed box of `Fig.4`. It has been labeled as “diffusion manifold” in the figure. This assumption is known not to hold in latent space, as discussed in [R13]. In contrast, MCLC does not assume any manifold structure. The Taylor expansion we use is not a geometric approximation of the data manifold, but an approximation of the loss deviation that is used solely to prove that the correction step preserves measurement consistency. It is a theoretical tool to ensure that the update stays within the orthogonal complement of the measurement constraint, illustrated as the “orthogonal complement” in `Fig.4`. As indicated in `Fig.4`, the role and purpose of these two notions are totally different, and MCLC does not impose any structure on the data manifold.
>
> Thus, the two notions of linearity differ fundamentally:
>
> - **Linear manifold assumption**: assuming that the manifold is locally linear in order to handle the gradient update direction.
> - **Taylor expansion in MCLC**: an analytical tool used to ensure that the correction step preserves measurement consistency without any manifold assumptions.
>
> Our point is simply that prior latent diffusion inverse solvers [R11,12] rely on a linear manifold assumption, which may not hold in latent space [R13]. In contrast, MCLC operates without this assumption, leading to more stable and reliable behavior in latent space. In addition, although our derivation only ensures the measurement consistency perfectly up to first-order terms, we have shown that the higher-order deviation remains bounded when the step size is chosen properly (please refer to `Appendix A`). Thus, the Taylor expansion is not meant to be interpreted as linearity.
>
> Finally, we note that our discussion of the linear manifold assumption is not intended as a criticism of prior works. If clarification is needed, we will adjust the tone down accordingly in the paper. The seminal works have provided an important and valuable ground for subsequent research, and we fully acknowledge their contributions.
>
> - [R11] He et al., Manifold Preserving Guided Diffusion, ICLR 2024.
> - [R12] Zirvi et al., Diffusion State-Guided Projected Gradient for Inverse Problems, ICLR 2025.
> - [R13] Song et al., Solving Inverse Problems with Latent Diffusion Models via Hard Data Consistency, ICLR 2024.
>
>
> > **W3. The experimental results of PSNR may be less valuable for applications where accuracy (PSNR) is more important than visual pleasantness (LPIPS/FID)**
>
> We would like to clarify that the role of MCLC is to stabilize the solving process and improve the solution reliability of the ill-posed problem, not to act merely as a perceptual enhancement module. Accordingly, in more challenging degradations, MCLC yields clear PSNR improvements in `Table 1` (e.g., Motion Deblur and Gaussian Deblur, with blur kernel size=128), demonstrating the benefits in both perceptual quality and in reconstruction fidelity. This is because improved stability guides the solver toward more reliable solutions. As reported in `Appendix B.2` (`Fig.10`), the PSNR histograms consistently shift toward higher values across all linear tasks, indicating improved stability and overall effectiveness. `Figure 13` is also helpful, as it shows that MCLC raises the performance ceiling of existing solvers.
>
> For several solvers and tasks, the improvement may appear relatively small because the task is less ill-posed or the base solver already achieves high PSNR due to its strong data-consistency design. Nonetheless, the reconstruction can still suffer from instability-induced artifacts, and MCLC directly addresses this instability.

---

### Official Review · Reviewer_JP5B · 2025-10-31

**Soundness:** 3
**Presentation:** 3
**Contribution:** 2
**Rating:** 4
**Confidence:** 4

**Summary:**

The paper proposes the Measurement-Consistent Langevin Corrector (MCLC), a plug-and-play method that augments latent diffusion inverse solvers with an orthogonally projected Langevin corrector step to preserve measurement consistency. The authors claim this corrector mitigates artifacts, narrows the KL gap between solver dynamics and the true reverse diffusion, and improves perceptual metrics across various inverse problems.

**Strengths:**

1. The paper is clearly written and well-structured, with consistent notation and clear transitions between sections.
2. The authors provide both theoretical and empirical analyses, including derivations that connect the proposed correction step to reduced KL divergence, which helps ground the method conceptually.
3. The evaluation covers a diverse set of inverse problems (super-resolution, deblurring, inpainting, HDR, nonlinear deblurring) and includes comparisons with multiple baseline solvers, demonstrating good experimental thoroughness.

**Weaknesses:**

1. The core idea of MCLC, adding a Langevin correction step with an orthogonal projection, appears very similar to established approaches such as predictor-corrector sampling in Song et al., 2021. The paper acknowledges this but presents the orthogonal projection for “measurement consistency” as its main novelty. This modification feels incremental, raising the question of whether constraining the Langevin step geometrically constitutes a substantial conceptual advance.
2. While the paper attributes artifacts and “blob” patterns to inherent issues in latent diffusion inverse solvers, it does not rule out simpler causes such as suboptimal hyperparameter tuning. Competing methods may not exhibit these artifacts simply because their measurement-consistency step sizes or regularization terms are better tuned.
3. Using only 100 images from each dataset is a bit unconvincing. Standard validation splits are publicly available for both datasets (1k images) and are commonly used in related work. The paper should report results on these standard sets or, at minimum, provide a clear justification and description of how the 100-image subsets were selected.
4. The paper does not provide any analysis or discussion of the additional computational burden introduced by MCLC. Since the method applies an extra Langevin and projection step at each diffusion iteration, it likely increases runtime and memory usage, but no quantitative results or comparisons are given.

**Minor Comments:**
- The effectiveness of MCLC appears sensitive to the choice of the corrector step size $\eta_t$, which the authors note remains non-trivial to select. Since the method’s stability and measurement consistency likely depend on this tuning, it would be helpful to discuss how $\eta_t$  was chosen in practice and whether results are robust to this parameter.

**Questions:**

1. *Regarding weakness 2:* Could the observed artifacts instead stem from under-tuned baselines rather than a fundamental flaw in their dynamics? Have the authors verified that measurement-consistency step sizes (e.g., $\zeta$ and $\gamma$ in Alg. 4)  and solver parameters were optimized fairly for all methods?
2. *Regarding weakness 4:* Can the authors report the actual runtime and memory overhead of MCLC relative to the base solvers, and clarify whether the added cost is negligible or significant in practice? Did you select k=3 heuristically (in Appendix C2)?

---

> ### Author Response · Authors · 2025-11-22
> **Author response (1/3)**
>
> We sincerely thank the reviewer $\textcolor{#E7B500}{JP5B}$ for valuable and detailed feedback. We address these points and provide more clarification below.
>
> > **W1: Whether the geometric constraint on the Langevin step provides a meaningful conceptual advance?**
>
> We respectfully clarify that our contribution is not simply adding a projection step. The conceptual advances of MCLC are threefold:
>
> **(1) The first identification of instability as a reverse diffusion dynamics discrepancy.**
> From empirical observations of unstable reverse-diffusion dynamics (`Fig.2`), we identify that the instability in LDM-based inverse solvers arises from a discrepancy between the solver’s induced distributions and the true time-marginal distributions (`Fig.3`). We evaluate this discrepancy using the KL divergence and confirm that reducing this gap makes the LDM-based solvers more stable. To our knowledge, the relationship between this instability and the time-marginal discrepancy has not been examined and addressed in prior works.
>
> **(2) MCLC ensures measurement-consistent Langevin update.**
> In inverse problems, measurement consistency is an essential and fundamental requirement [R6]. However, a direct application of PC sampling to inverse solvers can lead the dynamics to move along many suboptimal directions that pull samples toward high-probability regions, but may disturb measurement consistency (please refer to `L455-465` and `Table 5`). Therefore, the measurement-consistent correction scheme is not merely incremental; it provides a clear and effective solution for the inverse problem, which has not been approached previously.
>
> **(3) Conceptual advances beyond the linear manifold assumption**
> Existing diffusion-based inverse solvers [R1-4] assume a linear manifold assumption. While this assumption provides a convenient theoretical tool, it may not hold in latent space, as noted in [R5]. MCLC does not rely on this manifold assumption. Instead, we take a more general perspective without any manifold constraints. This enables more reliable latent space inverse problem solving and provides a clear conceptual advance.
>
> - [R1] Chung et al., Improving Diffusion Models for Inverse Problems using Manifold Constraints, NeurIPS 2022.
> - [R2] Chung et al., Diffusion Posterior Sampling for General Noisy Inverse Problems, ICLR 2023.
> - [R3] He et al., Manifold Preserving Guided Diffusion, ICLR 2024.
> - [R4] Zirvi et al., Diffusion State-Guided Projected Gradient for Inverse Problems, ICLR 2025.
> - [R5] Song et al., Solving Inverse Problems with Latent Diffusion Models via Hard Data Consistency, ICLR 2024.
> - [R6] Groetsch, Inverse Problems in the Mathematical Sciences, Springer 1993.

---

> ### Author Response · Authors · 2025-11-22
> **Author response (2/3)**
>
> > **W2 and Q1: Could the observed artifacts stem from under-tuned base methods rather than a fundamental flaw in their dynamics?**
>
> We respectfully would like to note that these artifacts have already been reported in prior works [R7,8] as inherent issues of latent diffusion-based inverse solvers even under well-tuned settings, particularly when using domain-agnostic general priors (e.g., Stable Diffusion). Thus, these artifacts are not unique to our implementation nor attributable to under-tuning. For more clarification, we address this point below.
>
> **We follow the tuned configurations of the original paper and implementation.** For clarity, we have detailed our baseline settings for each algorithm in `Appendix C`. All baselines were evaluated using their official or faithfully reproduced configurations: PSLD with its reported parameters, LDPS with the same settings as PSLD following [R8], and Resample with the official DiffState-Grad code and configurations.
> We have added additional results on recent advanced inverse solvers by strictly following the tuned configurations from the papers and public implementations in `Tables 2, 3, and 6 `(more details in `Appendix B.1`). This supports the effectiveness of MCLC under tuned settings. We hope this clarification will make our evaluation more transparent and convincing.
>
> **The baseline configuration we used is reasonably well-tuned.** To further address the reviewer’s concern, we have provided additional results by sweeping the key parameter, gradient step size, by scaling it with {`0.1x`, `0.5x`, `1x`, `2x`, `4x`} (please refer to `Appendix B.7` and `Figs.13-14`.) Across these settings, artifacts consistently appeared in base solvers (`Fig. 14`), and their instability resulted in worse FID than MCLC (`Fig.13` and table below). As shown in the table, the baseline configurations we used are nearly optimal in many cases. Nevertheless, `Fig.13` shows a practical tuning bound, where it is still difficult to obtain a perfectly tuned setting that achieves both stability and high fidelity. MCLC addresses this limitation by stabilizing the LDM-based inverse solver without sacrificing measurement fidelity, achieving both beyond the tuning bound.
>
> |          |      LDPS       |   (Motion Deblur)         |      PSLD       |      (Gaussian Deblur)      |     ReSample     |    (Super Resolution 4x)       |
> |-------|------------------|------------|------------------|------------|-------------------|-----------|
> |       | **PSNR (↑)**     | **FID (↓)**| **PSNR (↑)**     | **FID (↓)**| **PSNR (↑)**      | **FID (↓)** |
> | Base 0.1x  | 25.67            | 92.65      | 26.55            | 85.91      | 27.55             | 81.15     |
> | Base 0.5x  | 26.69            | 102.41     | 27.44            | 82.38      | 27.40             | 68.00     |
> | Base 1x     | 26.54            | 118.77     | 27.84            | 89.18      | 26.40             | 70.16     |
> | Base 2x     | 25.31            | 142.82     | **28.40**        | 89.27      | 24.76             | 79.85     |
> | Base 4x     | 17.54            | 255.61     | 27.75            | 97.09      | 23.47             | 86.90     |
> | Ours 1x | **27.45**        | **82.94**  | 27.97            | **66.28**  | **28.32**         | **53.85** |
>
> - [R7] Chung et al., Prompt-tuning Latent Diffusion Models for Inverse Problems, ICML 2024.
> - [R8] Raphaeli et al., SILO: Solving Inverse Problems with Latent Operators, ICCV 2025.

---

> ### Author Response · Authors · 2025-11-22
> **Author response (3/3)**
>
> > **W3. The 100-image subset selection is unconvincing and needs justification**
>
> As noted in `L394`, we strictly follow the data selection protocol suggested by DAPS [R9]: we use the same 100-image subset provided in their official implementation (the first 100 images of the validation set, e.g, FFHQ dataset 49000–49099.png), without any additional or preferential selection on our side. This 100-image validation protocol has also been recently adopted [R4,5,9,10] for extensive experiments across datasets and tasks. We have clarified this setting in the revised version.
>
> In addition, to further validate the effectiveness of MCLC, we present a result on the full 1k validation set for the super-resolution (4x) task using ReSample. This experiment demonstrates that our conclusions are consistently drawn from the 100-image evaluation.
>
> |       | **Super** | **Resolution**       |  **(4x)**     |       |
> |-------|---------------------------|-------|-------|-------|
> | Method| PSNR | LPIPS | FID | P-FID |
> | Resample | 26.26 | 0.348 | 36.62 | 93.82 |
> | Resample w/ Ours | **27.74** | **0.270** | **29.39** | **42.79** |
>
> - [R9] Zhang et al., Improving Diffusion Inverse Problem Solving with Decoupled Noise Annealing, CVPR 2025.
> - [R10] Wu et al., Principled Probabilistic Imaging using Diffusion Models as Plug-and-Play Priors, NeurIPS 2024.
>
>
> > **W4 and Q2: The paper did not report the additional computational burden introduced by MCLC**
>
> Thanks for the constructive feedback. To clarify the additional cost of MCLC, we have included the runtime and memory analyses in the main paper (`L437-453` and `Fig.6`). Across all solvers, the additional wall-clock time is still within a reasonable range (typically 3–4%), and MCLC introduces no additional memory overhead, as confirmed by peak-memory usage. Since MCLC requires only the forward pass through LDM and simple algebraic operations, the computational burden is manageable. The number of correction iterations k=3 was determined heuristically.
>
> > **Minor Comment: Guide for tuning of the MCLC step size.**
>
> Thanks for the helpful suggestion. As derived in `Eq.65` (`L1007`), the MCLC step size can be guided by the theoretical bound; for a 64x64x4 latent, a value around $\lambda \approx 0.01$ works reliably.  In practice, for efficiency, increasing the step size and reducing the correction interval also work well. For severe degradations, slightly larger step sizes and more correction iterations can be beneficial, even if they introduce a small measurement-consistency deviation.
>
> We summarize these practical guidelines in `Appendix B.14`, and the provided tuned configurations serve as empirical guidance. We have also included a table showing that a single hyperparameter setting performs consistently across all linear tasks (see `Table 8`), suggesting it is a reasonable starting point for tuning.

---

### Official Review · Reviewer_5qUu · 2025-11-01

**Soundness:** 2
**Presentation:** 2
**Contribution:** 2
**Rating:** 4
**Confidence:** 4

**Summary:**

This paper introduces the Measurement-Consistent Langevin Corrector (MCLC), a novel, plug-and-play module designed to remedy artifacts and degraded quality commonly seen in existing zero-shot inverse solvers based on Latent Diffusion Models (LDMs).

**Strengths:**

1. The plug-and-play nature is highly significant, showing that simple, post-update correction can elevate the performance of various base solvers (LDPS, PSLD, ReSample) to be comparable to, or even surpass, more complex, recent methods (LatentDAPS). The method is even shown to be applicable to Pixel Diffusion Models (PDMs)

**Weaknesses:**

1. Langevin correction is a well-known technique in diffusion models that can be used with any diffusion solver, which is not novel to me.

2. Measurement Consistency correction requires additional forward and backward propagation, lacking relevant analysis of computation time and consumption.


3. To my knowledge, we only need a simple Restart [1][2] to alleviate accumulated errors and adverse artifacts, which is simpler and more intuitive for me.

[1] RePaint: Inpainting using Denoising Diffusion Probabilistic Models

[2] Restart Sampling for Improving Generative Processes

**Questions:**

Why did the author only conduct experiments on latent  diffusion, while their baseline method DiffState also performed experiments in the pixel domain.

What are the advantages of the proposed method compared to Restart method.

---

> ### Author Response · Authors · 2025-11-22
> **Author response (1/2)**
>
> We thank the reviewer $\textcolor{#F1433F}{5qUu}$ for insightful comments and suggestions. We address these points below.
>
> > **W1. Langevin correction is not novel**
>
> We would like to clarify that our novelty does not lie in using Langevin correction itself, but 1) in identifying the instability in latent diffusion-based inverse problem solvers as a gap between solver-induced distribution and true time-marginal diffusion distribution at timestep, and 2) in introducing a measurement-consistent correction scheme that ensures the Langevin update operates correctly in the inverse problem, where measurement consistency is a fundamental objective.
>
> Specifically, to our knowledge, we are the first to characterize the instability of LDM-based inverse problem solvers as a reverse-dynamics discrepancy. By formulating this issue as a gap from the true diffusion time-marginal distribution (please refer to `Fig.3` and `Sec.3`), we show that reducing this gap stabilizes the LDM-based inverse problem solvers. Building on this, we first propose a principled correction scheme to remedy the instability of LDM-based inverse problem solvers.
>
> Furthermore, while the standard Langevin correction drives samples toward high-probability regions of the time-marginal distribution at each timestep t, applying it directly in inverse solvers can disturb measurement consistency (please refer to `L455-465` and `Table 5`). This is because there are many suboptimal directions in the high-dimensional latent space that may increase prior fidelity but deviate from the measurement. Our proposed measurement-preserving correction mechanism has not been explored in previous works, and it provides a theoretically grounded and well-justified solution to the reverse dynamics discrepancy while remaining faithful to the inverse problem objective, measurement consistency [R1].
>
> - [R1] Groetsch, Inverse Problems in the Mathematical Sciences, Springer 1993.
>
> > **Q1. Why did the authors only conduct experiments on latent diffusion?**
>
> We would like to note that experiments on PDM are already included in `Appendix B.5` (`Table 7` and `Fig.12`). Additionally, since our main focus (as outlined in `Sec.1`) is on general learned priors rather than task-specific PDMs/LDMs (e.g., models trained only on FFHQ), we primarily report results using LDM-based inverse solvers built on large pretrained diffusion models such as Stable Diffusion.
>
>
> > **W2. Extra forward/backward cost in MCLC and missing computation analysis**
>
> We would like to clarify that MCLC does not introduce any additional backward propagation, because it reuses the gradient computed in the measurement-consistency step of the base solver. Thus, the extra computation is a lightweight feed-forward through LDM. In our implementation, each forward and backward operation takes: forward 0.045s, backward 0.161s.
>
> Thanks for the helpful suggestion regarding computation. We have newly added computation-time and memory analyses in the main paper (`L437-453` and `Fig.6`). The additional wall-clock time is modest (3-4%). For ReSample, the increase is more noticeable due to its heavy inner gradient-descent loops, but the overhead remains manageable, and MCLC provides substantial performance gains (see `Table 1`). In addition, since MCLC requires only model forward and simple algebraic operations, it introduces no additional memory overhead, as confirmed by the peak memory usage.

---

> ### Author Response · Authors · 2025-11-22
> **Author response (2/2)**
>
> > **W3 and Q2. Advantages of the proposed method over the simple Restart scheme**
>
> We would like to respond from a theoretical and an empirical perspective.
>
> **Theoretical perspective.** As identified in our analysis, the quantity that be reduced to resolve the instability is the KL divergence between the solver’s induced distribution $q_t$ and the true diffusion time-marginal $p_t$:  $\mathrm{KL}(q_t \| p_t).$ The restart scheme, which perturbs the latent with a small forward diffusion noise step and then applies a reverse sampling step again, does not provide any guarantee of decreasing this KL divergence gap.
>
> In contrast, at each diffusion time $t$, Langevin dynamics correspond to the steepest descent flow of $\mathrm{KL}(q_t \| p_t)$ in Wasserstein space [R2,3], which guarantees monotonic KL decrease. Therefore, Langevin correction provides a principled direction for reducing the reverse-dynamics discrepancy, whereas restarting schemes have no such guarantee. Although restart may suppress artifacts in some cases, the root cause of instability is this KL gap, and MCLC has a clear advantage because it directly reduces this gap while preserving measurement fidelity.
>
> **Empirical perspective.** Additionally, following the reviewer’s suggestion, we have implemented two restart variants on the LDM-based inverse problem solver, PSLD. At intermediate sampling steps, we applied iterative restarting using
> - (1) Adding a forward diffusion step and then running standard reverse sampling (similar to [R4]), and
> - (2) performing reverse sampling with gradient-based posterior sampling (similar to [R5,6]), which even requires additional backpropagation costs.
>
> |             |            |       Gaussian     |     Deblur        |            |
> |---------------------|------------|------------|------------|------------|
> | Method              | PSNR (↑)   | LPIPS (↓) | FID (↓)    | P-FID (↓)  |
> | Base                | 27.84      | 0.314     | 89.18      | 90.54      |
> | Restart (1)         | 27.71      | 0.316     | 87.34      | 73.83      |
> | Restart (2)         | 27.90      | 0.318     | 89.44      | 76.12      |
> | Ours                | **27.97**      | **0.286**     | **66.28**      | **59.13**      |
>
>
> Both variants indicate that simply reintroducing noise and reversing does not effectively stabilize the reverse diffusion dynamics in LDM-based inverse solvers, as appeared in FID scores. These empirical observations align with the theoretical arguments above and further highlight the advantage of MCLC.
>
> - [R2] Jordan, Kinderlehrer, and Otto, The Variational Formulation of the Fokker-Planck Equation, SIAM Journal on Mathematical Analysis 1998.
> - [R3] Vempala and Wibisono, Rapid Convergence of the Unadjusted Langevin Algorithm: Isoperimetry Suffices, NeurIPS 2018.
> - [R4] Xu et al., Restart Sampling for Improving Generative Processes, NeurIPS 2023.
> - [R5] Lugmayr et al., Repaint: Inpainting using Denoising Diffusion Probabilistic Models, CVPR 2022.
> - [R6] Grechka et al., GradPaint: Gradient-guided Inpainting with Diffusion Models, Computer Vision and Image Understanding 2024.

---

### Author Response · Authors · 2025-11-22
**Official Comment by Authors**

$\textcolor{#F1433F}{Reviewer \ 5qUu}$, $\textcolor{#E7B500}{Reviewer \ JP5B}$, $\textcolor{#A9CF54}{Reviewer \ D5xV}$, $\textcolor{#70B7BA}{Reviewer \ Udz6}$

**We sincerely thank all reviewers for their constructive and thoughtful feedback.
We appreciate the reviewers’ recognition of our work, including:**

- Informative theoretical and empirical analyses ($\textcolor{#E7B500}{JP5B}$, $\textcolor{#70B7BA}{Udz6}$)
- A principled and well-justified solution ($\textcolor{#E7B500}{JP5B}$, $\textcolor{#A9CF54}{D5xV}$, $\textcolor{#70B7BA}{Udz6}$)
- Plug-and-play nature with broad applicability ($\textcolor{#F1433F}{5qUu}$, $\textcolor{#A9CF54}{D5xV}$, $\textcolor{#70B7BA}{Udz6}$)
- Comprehensive evaluations across diverse tasks and solvers ($\textcolor{#E7B500}{JP5B}$, $\textcolor{#A9CF54}{D5xV}$, $\textcolor{#70B7BA}{Udz6}$)
- Well-organized and clearly structured presentation ($\textcolor{#E7B500}{JP5B}$, $\textcolor{#A9CF54}{D5xV}$)

**We have carefully addressed all comments in the responses below and revised the paper.  Newly added or modified parts in the revised paper are marked in green.**

**Our revisions are summarized as follows:**

- Clarified the evaluation dataset → Section 4, L394–395 ($\textcolor{#E7B500}{JP5B}$)
- Added experiments validating broader applicability of MCLC→ Section 4, p. 8; Appendix B.1, pp. 19–20 ($\textcolor{#A9CF54}{D5xV}$, $\textcolor{#70B7BA}{Udz6}$)
- Added computational cost analysis→ Section 4, p. 9 ($\textcolor{#F1433F}{5qUu}$, $\textcolor{#E7B500}{JP5B}$)
- Expanded discussion on the measurement-consistent correction → Section 4, p. 9 ($\textcolor{#F1433F}{5qUu}$, $\textcolor{#E7B500}{JP5B}$, $\textcolor{#70B7BA}{Udz6}$)
- Added experiments on baseline hyperparameter tuning → Appendix B.7, pp. 24–25 ($\textcolor{#E7B500}{JP5B}$)
- Added qualitative results on blob-artifact mitigation → Appendix B.8, p. 26 ($\textcolor{#70B7BA}{Udz6}$)
- Added analysis on the cause of artifacts → Appendix B.12, pp. 29–30 ($\textcolor{#70B7BA}{Udz6}$)
- Added discussion and experiments on MCLC hyperparameters, including a single default setting and tuning guidelines→ Appendix B.6, p. 22-23 & Appendix B.14, p. 31 ($\textcolor{#A9CF54}{D5xV}$, $\textcolor{#E7B500}{JP5B}$)

**We sincerely appreciate the reviewers’ time, constructive suggestions, and valuable insights, which have significantly strengthened our submission and made it more convincing.**

---

> ### Author Response · Authors · 2025-11-28
> **Clarification and Summary of Our Contributions**
>
> **We summarize our contributions to clearly convey the novelty, insights, and conceptual advancements of our work:**
>
> - We identify instability in latent diffusion inverse solvers as a divergence between the solver’s reverse dynamics and the true diffusion dynamics, and show that reducing this divergence stabilizes the solvers.
> - We propose a novel, theoretically justified plug-and-play measurement-consistent correction scheme that remedies latent-diffusion inverse solvers while preserving measurement consistency, enabling proper correction when solving inverse problems.
> - Our divergence-reduction correction is designed without the linear manifold assumption, which may hold in latent space, offering more stable and reliable solutions for latent diffusion inverse solvers.
> - We present comprehensive experiments across solvers and tasks, demonstrating the broad applicability and effectiveness of MCLC.
> - We provide an analysis of the origin of blob-like artifacts, revealing their connection to the latent space and the decoder.
>
> **We hope this clarification provides a clear understanding of the value and distinct contributions of our work. We will also incorporate these clarified points into the final version to ensure that our contributions are presented clearly.**
>
> **We sincerely appreciate the area chair and reviewers for their time and careful attention devoted to our submission.**

---

### Author Response · Authors · 2025-12-03
**Summary of Our Rebuttal for Area Chair**

Dear Area Chair,

We sincerely appreciate your time and effort in handling our submission, especially given the unusual circumstances from the OpenReview incident.

We have carefully addressed all questions, concerns, and suggestions by the reviewers and posted detailed responses on **Nov 23**. Since further discussion with the reviewers is no longer possible, we provide a rebuttal summary of the primary concerns to help your understanding and decision.

> **1. Clarification of the Problem in LDM-based Inverse Problem Solvers**

- *Initial Concerns:* Instability may stem from (i) under-tuned baselines or (ii) the sole prior, rather than being inherent to LDM-based inverse solvers, and (iii) KL curves in `Fig. 3` may suggest no issue ($\textcolor{#E7B500}{JP5B}$, $\textcolor{#70B7BA}{Udz6}$).
- **Clarification and Revision**: Prior works have already reported similar issues in LDM-based inverse solvers. We further address the reviewers’ concerns:
    - (i) Not caused by tuning: Included hyperparameter-sweep results (`Figs. 13–14`, `App. B.7`) show that the baselines are already well-tuned, and tuning alone has a clear performance ceiling. MCLC consistently surpasses this ceiling.
    - (ii) Not caused solely by the prior: Instability persists across different priors when the solver structure is fixed (`Tab. 9`, `Fig. 18` in `App. B.12`). Moreover, naive generation sampling produces no such artifacts (`Fig. 19`).
    - (iii) Misinterpretation of the KL-divergence: To prevent potential misinterpretation of `Fig. 3`, we added a clearer illustrative example.
- **Impact:** These revisions clarify the core problem setting, address the reviewers’ concerns, and further strengthen motivation of MCLC.

> **2. Clarification of the Novelty**

- *Initial Concerns:* The novelty and contribution appear to be marginal. ($\textcolor{#F1433F}{5qUu}$, $\textcolor{#E7B500}{JP5B}$, $\textcolor{#70B7BA}{Udz6}$)
- **Clarification:** Our contribution does not lie in merely applying Langevin correction. Instead, it comes from:
    - First identification of instability as a reverse-dynamics discrepancy, with evidence that reducing this gap stabilizes the solver.
    - A theoretically justified, measurement-consistent correction scheme usable in a plug-and-play manner.
    - A general stabilization approach that does not rely on the linear-manifold assumption, which may not hold in latent space.
    - Analysis of blob artifacts, explaining their origin.
- **Revision:** Revised paper to highlight these contributions.
- **Impact:** These demonstrate that our work provides both meaningful conceptual advances and practical improvements in LDM-based inverse solvers.

> **3. Future Applicability of MCLC**

- *Initial Concerns:* MCLC appears to show relatively lower compatibility with LatentDAPS, which may imply limited applicability. ($\textcolor{#A9CF54}{D5xV}$, $\textcolor{#70B7BA}{Udz6}$)
- **Clarification:** As discussed in `App. B.9`, this lower compatibility is due to LatentDAPS’s particular design rather than a limitation of MCLC.
- **Revision:** We added quantitative results and discussions on recent advanced LDM-based and flow-based inverse solvers (`Tabs. 2,3,6` and `App. B.1`).
- **Impact:** These results show MCLC’s high practical utility and future applicability.

> **4. Clarification of Computational Burden**

- *Initial Concerns*: Need to provide an analysis of the computational burden by MCLC. ($\textcolor{#F1433F}{5qUu}$, $\textcolor{#E7B500}{JP5B}$)
- **Clarification:** Since MCLC requires only a forward pass and simple algebra, it introduces no memory overhead, and additional runtime remains within a reasonable range (mostly 3–4%).
- **Revision:** We added a computational cost analysis in `Fig.6` and `L436-453`.
- **Impact:** MCLC achieves meaningful performance gains without high additional computational costs.

> **5. Clarification of the Hyperparameter Tuning of MCLC**

- *Initial Concerns:* It seems that MCLC needs task-specific tuning. Is there one default configuration that improves performance across tasks? ($\textcolor{#A9CF54}{D5xV}$, $\textcolor{#E7B500}{JP5B}$)
- **Clarification:** Our theoretically derived bound (`App. A`, `Eq. 65`) provides clear step-size choices. Moreover, task-specific tuning is a standard practice in inverse problem solvers.
- **Revision:** We included a single default setting that yields reasonable improvements across diverse tasks in practical settings (`Tab. 8`, `App. B.6`), and provided practical tuning guidelines to help users when degradations vary (`App. B.14`).
- **Impact:** With theoretical justification, default configurations, and practical guidelines, MCLC becomes principled and easy to apply, enabling practical use and further extensions.

We believe that our clarifications and revisions have substantially strengthened our submission. We sincerely appreciate your time and consideration during this exceptional review process, and thank you again for evaluating our submission.

---

### Meta-Review · Area_Chair_yySr · 2026-01-06

**Summary:**

This paper proposes Measurement-Consistent Langevin Corrector (MCLC), a plug-in corrector for latent diffusion inverse solvers (e.g., LDPS/PSLD/ReSample). The central claim is that a main failure mode is instability in the reverse dynamics (a mismatch between the solver-induced distribution and the diffusion time-marginal), and that adding a measurement-consistent Langevin-style correction can stabilize sampling and reduce artifacts across inverse problems. Reviewers generally agree the problem is real and the empirical coverage is decent, but the decision is driven by concerns about novelty (how far this goes beyond standard corrector ideas) and by questions about practical trade-offs (tuning burden, overhead, and consistency of gains on strong solvers).

**Reviewer Concerns:**

Concerns addressed by the rebuttal:
1) Transparency of overhead: the authors add runtime/memory discussion and clarify the additional compute is modest relative to base solvers.
2) “Is this just under-tuned baselines?”: the response reports additional sweeps and argues the observed instability is not simply a tuning artifact.
3) Evaluation protocol questions (e.g., small subsets): the response clarifies the subset choice and adds larger-scale checks to support the same trend.
4) “Why not just restart?”: the response adds a restart comparison and a rationale for why restart does not reliably close the gap.
5) Clarity points (e.g., interpretation of KL curves / “messy latents”): the response provides clearer explanations and figure clarifications.

Concerns still outstanding:
1) Novelty/impact remains the main sticking point. Even with the authors’ framing, the method still reads close to a standard corrector recipe with an added measurement-preserving constraint, and reviewers may reasonably differ on whether that clears the conceptual bar.
2) Usability and tuning burden: the rebuttal provides defaults and guidance, but it still seems likely that strong performance across tasks/solvers requires nontrivial tuning, which weakens the “simple plug-and-play” message.
3) Strength of evidence on the strongest / most modern solver variants: the authors explain limited compatibility with some methods, but that leaves some uncertainty about how future-proof the contribution is.

**Reviewer Scores:**

1) Reviewer (rating 4): likely unchanged at 4, or a small increase to 5, since the rebuttal directly answers the latent-vs-pixel and restart questions but does not fundamentally change the novelty concern.
2) Reviewer (rating 4): likely increase to 5 after the added overhead analysis and fairness/tuning clarifications.
3) Reviewer (rating 6): likely unchanged at 6. The rebuttal helps on tuning/defaults and applicability, but the reviewer’s overall tone is already “borderline accept / fine either way.”
4) Reviewer (rating 4): likely increase to 5 after the clarifications on the KL interpretation and what instability is meant to capture.

---

### Decision · Program_Chairs · 2026-01-26

Reject